# Resonant inelastic X-ray scattering tools to count $5f$ electrons of actinides and probe bond covalency

Bianca Schacherl [1,2,10], Michelangelo Tagliavini [3,10],
Hanna Kaufmann-Heimeshoff[1], Jörg Göttlicher[4], Marinella Mazzanti [5],
Karin Popa [6], Olaf Walter [6], Tim Pruessmann [1], Christian Vollmer [1],
Aaron Beck[1], Ruwini S. K. Ekanayake [1], Jacob A. Branson [2], Thomas Neill [1,9],
David Fellhauer[1], Cedric Reitz [1], Dieter Schild [1], Dominique Brager [7],
Christopher Cahill[7], Cory Windorff [8], Thomas Sittel [1], Harry Ramanantoanina[1],
Maurits W. Haverkort [3] ✉ & Tonya Vitova [1] ✉

The actinides possess a complex electronic structure, making their chemical and physical properties among the least understood in the periodic table. Advanced spectroscopic tools, able to obtain deep insights into the electronic structure and binding properties of the actinides, are highly desirable. Here, we introduce two sensitive spectroscopic tools: one determines the number of localized $5f$ electrons on an actinide atom, and another assesses the covalent character of actinide-ligand bonding. Both tools are based on the multiplet structure present in actinide $M_4$ edge core-to-core resonant inelastic X-ray scattering (CC-RIXS) maps. The spectral intensity of different many-body final-state multiplets directly depends on the local many-electron ground-state symmetry including the local $5f$ spin configuration. By comparing U $M_4$ edge CC-RIXS data for 21 U, Np, Pu and Am compounds, we demonstrate the ability to compare the number of localized $5f$ electrons and bond covalency across the actinide series.

The actinide elements have fascinating electronic structure and chemical binding properties. Actinide chemistry is, however, still less understood than the chemistry of many other elements in the periodic table[1–9], primarily due to the important role of relativistic effects and the increasing number of electrons leading to strong electron correlation effects and complex chemistry and physics[10,11]. Predictive understanding of the actinide electronic structure and bonding properties is the baseline for deep understanding of their environmental behavior, for tuning their magnetic and superconducting properties in new materials or for optimizing properties of

[1]Karlsruhe Institute of Technology (KIT), Institute for Nuclear Waste Disposal (INE), P.O. Box 3640, 76021 Karlsruhe, Germany. [2]Lawrence Berkeley National Laboratory (LBNL), Chemical Sciences Division (CSD), 1 Cyclotron Road, Berkeley, CA 94720, USA. [3]Heidelberg University, Institute for Theoretical Physics (ITP), Philosophenweg 19, 69120 Heidelberg, Germany. [4]Karlsruhe Institute of Technology (KIT), Institute for Photon Science and Synchrotron Radiation (IPS), P.O. Box 3640, 76021 Karlsruhe, Germany. [5]Institut des Sciences et Ingénierie Chimiques, Ecole Polytechnique Fédérale de Lausanne (EPFL), CH-1015 Lausanne, Switzerland. [6]European Commission, Joint Research Centre Karlsruhe (JRC), Karlsruhe, Germany. [7]Department of Chemistry, The George Washington University, 800 22nd Street, NW, Washington, DC 20052, USA. [8]Department of Chemistry and Biochemistry, New Mexico State University, MSC 3 C, P.O. Box 30001 Las Cruces, NM 88003, USA. [9]Present address: Radioactive Waste Disposal and Environmental Remediation (RADER) National Nuclear User Facility and Williamson Research Centre, Department of Earth & Environmental Sciences, The University of Manchester, Oxford Road, Manchester M13 9PL, UK. [10]These authors contributed equally: Bianca Schacherl, Michelangelo Tagliavini. ✉e-mail: M.W.Haverkort@thphys.uni-heidelberg.de; tonya.vitova@kit.edu

radiopharmaceuticals[12–15]. The development of experimental and theoretical spectroscopic tools, able to reliably probe the complex electronic structure of the actinide elements is the key[15]. Contemporary theory can describe electron density distributions, localized on the actinide atom or delocalized and partners in a chemical bond[9,11]. The ability to apply experimental techniques sensitive to the localized or delocalized $5f$ electron densities for all actinides, in both model and applied systems, to answer fundamental questions as well as those relevant to societal challenges, is much needed. Various advanced experimental techniques, such as Nuclear Magnetic Resonance (NMR)[16], electron paramagnetic resonance (EPR)[17], actinide $M_{4,5}$ absorption edge ($M_{4,5}$) X-ray magnetic circular dichroism (XMCD)[18], high energy resolution X-ray absorption near edge structure (HR-XANES) or valence band resonant inelastic X-ray scattering (VB-RIXS) are sensitive to different levels of localization of the $5f$ electron density on the actinide atoms and provide complementary information[15,19].

However, these methods have specific limitations. NMR is limited to specific actinide elements/isotopes[20], actinide $M_{4,5}$ edge HR-XANES is sensitive to changes of $5f$ electron density near the metal but not strictly to electrons localized on the actinide atom, leading to ambiguous determinations of local electron configurations[15,19,21,22]. Actinide $M_{4,5}$ edge VB-RIXS can probe both localized and delocalized electron density, but the probability for the electron transitions is low making it very challenging to obtain high quality data, especially with the high energy resolution needed to probe localized electrons[15,23]. Actinide $O_{4,5}$ edge non-resonant inelastic X-ray scattering (NIXS) at high momentum transfer is powerful for characterizing ground state electron configurations for actinides. However, it requires samples with a high concentration of the element of interest, a very high photon flux and multi-analyzer crystal spectrometers, making it very challenging and hardly suitable for general applications[24–26]. Actinide $3d4f$ core-to-core resonant inelastic X-ray scattering measured at the $M_4$ absorption edge (actinide $M_4$ edge CC-RIXS) was explored by us and others as an advanced tool for probing the electronic structure and bonding properties of the actinide elements[15,27–30]. It can be applied to all actinide elements and is now accessible at several synchrotrons all over the world[15,31]. However, there are many aspects of the actinide $M_{4,5}$ edge CC-RIXS map not yet well understood.

In this work, we will demonstrate that a satellite peak in actinide $M_4$ edge CC-RIXS can be used to uniquely count the $5f$ electrons localized on the early actinide atoms (U-Am) and, when recorded in a different experimental geometry, is sensitive to the level of covalency of the actinide-ligand chemical bonds. This spectroscopic probe is accessible and will be beneficial for characterizing the magnetic properties of actinide materials. It will underpin our understanding of the bonding properties of actinide molecular and solid-state materials, such as Pu in a colloidal solution, which is of high relevance for obtaining new insights into the environmental mobility behavior of plutonium. Our findings show that this tool contributes significantly to the high-resolution X-ray spectroscopic toolkit, which can provide a holistic understanding of the level of participation of the actinide $5f$, $6d$ and $6p$ orbitals in covalent binding in any actinide material.

## Results and discussion
### The origin of the satellite peak in actinide $M_4$ edge CC-RIXS maps

Core-to-core RIXS is a spectral measurement based on the X-ray scattering from atoms, resulting in the emission of X-rays of lower energy. The energy of the incident photon is transferred to the atom triggering excitations, these are followed by electronic de-excitations, which are determined by selection rules. In this general description of the RIXS process, the atom will have different ground, intermediate and final state electronic structures (cf. Fig. 1a)[32]. The RIXS process can also be qualitatively described for $M_4$ edge CC-RIXS ($3d4f$) as follows; the actinide $3d$ electrons are excited to $5f$ states ($3d \rightarrow 5f$) by incident

photons, followed by relaxation of $4f$ electrons to fill the created $3d$ core-holes with high probability ($4f \rightarrow 3d$, cf. Fig. 1b) resulting in photon emission. The selection rules for electron transitions imply that the angular momentum $l$ can change by $\pm 1$ ($\Delta l = \pm 1$) and the total angular momentum $J$ can change by $0, \pm 1$ ($\Delta J = 0, \pm 1$)[32]. In this work, we focus on the actinide $M_4$ edge CC-RIXS and the most intense emitted characteristic fluorescence, i.e., the actinide $M_\beta$ fluorescence line (cf. Fig. 1c, d). Note, the experimental data for all figures in the paper and SI and all input files for the calculations are available as source data under https://doi.org/10.35097/5qqht0vfqrktvccj. The U $M_4$ edge CC-RIXS depicts the intensity of the $M_\beta$ fluorescence across a series of emission energies as a function of the incident excitation energy scanned across the actinide $M_4$ absorption edge. Instead of emission energy, energy transfer can be depicted on the abscissa (cf. Fig. 1d); the energy transfer is equal to excitation minus emission energy.

The satellite peak of interest in this study is observed at 6 to 8 eV higher emission energy compared to the most intense (resonant) peak in the $M_4$ edge CC-RIXS map (white line, WL, cf. Fig. 1c, d). The cross-section through the maximum intensity of the satellite peak at fixed emission energies is marked with cyan, whereas green is used for the cross-section at constant excitation energies. Blue and orange lines are used for the sections at constant energy transfer through the maxima of the WL and the satellite peaks, respectively (cf. Fig. 1c, d).

### Theoretical description of the origin of the satellite peak in the actinide $M_4$ edge CC-RIXS maps

The origin of the satellite feature is related to the exchange interaction between the $4f$ core hole and $5f$ electrons ($4f–5f$) in the multi-electron final state of the RIXS process[28]. It is therefore a characteristic of the $4f^{13}5f^n$ multiplets and it fundamentally exists independently of the chosen scattering geometry. However, the intensity of the peak obtained experimentally is related to the many-body matrix elements involving the ground state, final state, and light polarization. The latter depends on the experimental geometry. Satellite peaks with similar origin were discussed for $4f$ X-ray photoelectron spectroscopy (XPS) where the final state configuration also has an open $5f$ and $4f$ shell[33–35]. However, those satellite peaks strongly overlap with peaks present because of excitations from the occupied valence band (shake-up or shake-off), caused by charge transfer processes, limiting their useful application as a characterization tool[36].

In order to better understand the effect of Coulomb interactions on bound states, it is convenient to expand the Coulomb interaction to spherical harmonics (cf. Methods, Supplementary Note 1, and Supplementary Fig. 1). This leads to Slater integrals describing the transfer of angular momentum between the electrons. It is important to note that these types of expansions are convenient for atoms in spherical symmetry. For atoms in solids or molecules, it works sufficiently well if the chosen basis is atomic like (cf. Methods section Calculations). As a result, the strength of the $4f–5f$ exchange interaction responsible for the satellite peak is given by the Slater coefficients $G^k_{4f5f}$, with k = 0, 2, 4, 6 specifying the exchange of angular momentum between the electrons[28]. The monopole part of the exchange interaction, given by the Slater integral $G^{k=0}_{4f5f}$, dominates the final state multiplet splitting and leads, following Hund's rule, to an increase of the energy of the states with smaller total spin.

An example is presented in Fig. 2a, where the energy level diagram for an atomic $4f^{13}5f^1$ system as a function of $G^{k=0}_{4f5f}$ is displayed. The spin (S = ½) of the $4f^{13}$ configuration can couple with the spin of the $5f^1$ configuration (S = ½) to form a singlet (one state) or triplet (195 possible different states). The energy difference between the spin singlet and triplet states is equal to 14 times $G^{k=0}_{4f5f}$. The large size of this effect results from the high multiplicity of the $G^0_{4f5f}$ scattering process, which is particularly large for two open $f$ shells. Besides the monopole exchange interaction, we included the full atomic Coulomb interactions and relativistic spin-orbit coupling. As can be seen in Fig. 2a, the

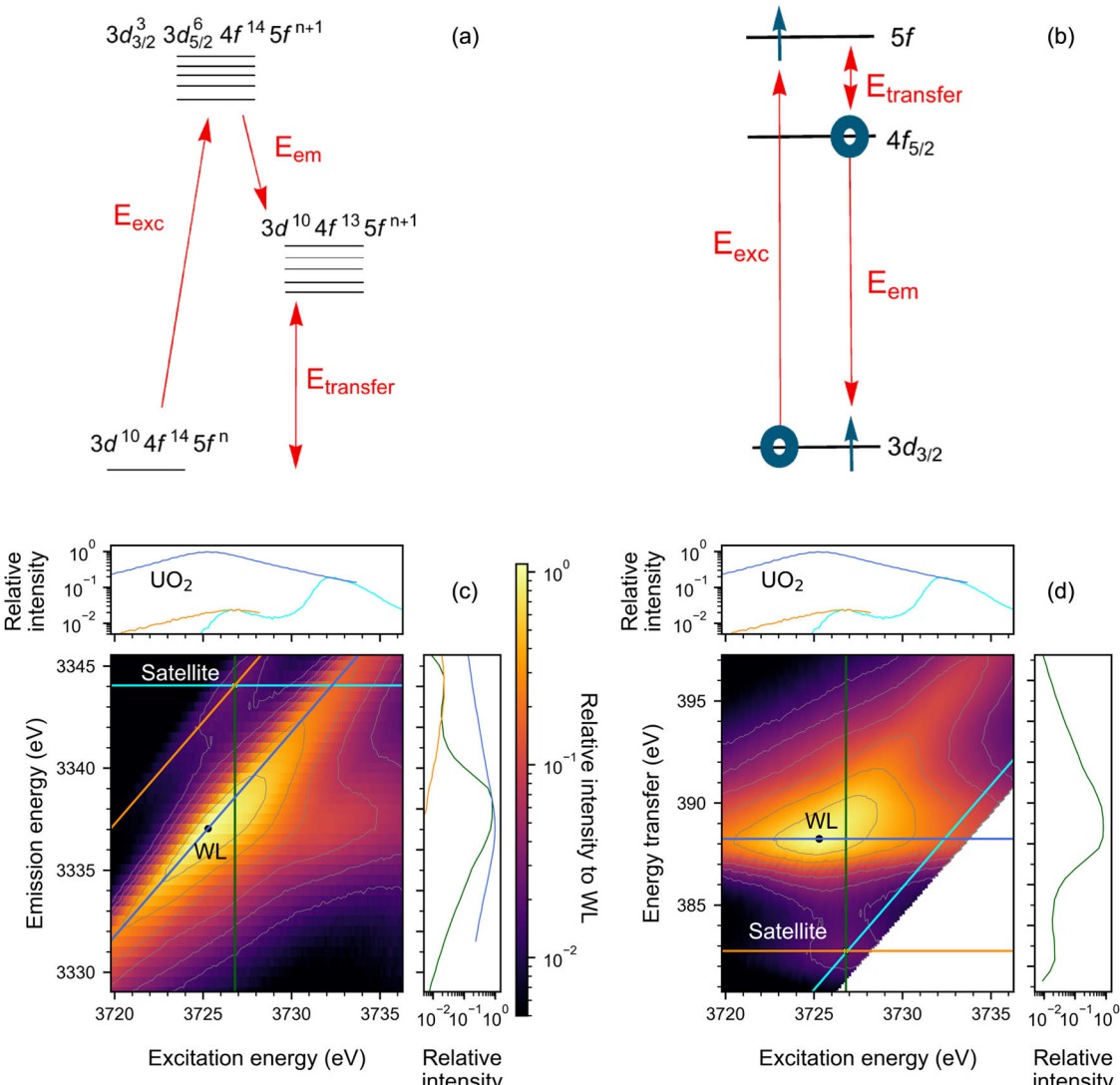

**Fig. 1 | Schematics of the RIXS process and $M_4$ edge CC-RIXS maps of $UO_2$.**
**a** Exemplary energy level diagram of the many electron system for ground ($3d^{10}4f^{14}5f^n$), intermediate ($3d_{3/2}^3\,3d_{5/2}^6\,4f^{14}5f^{n+1}$) and final state ($3d^{10}4f^{13}5f^{n+1}$) describing the CC-RIXS process at the actinide $M_4$ absorption edge. The states define the total energy of the atom. The red arrows define the typical energy quantities used to describe the process. **b** Single particle picture for the CC-RIXS process at the actinide $M_4$ absorption edge. The blue arrows represent electrons and the blue circles electron holes after the excitation/deexcitation. **c** Experimental

$UO_2$ $M_4$ edge CC-RIXS map depicting intensity changes of the $M_\beta$ characteristic fluorescence as a function of the excitation energy across the U $M_4$ absorption edge. **d** The same RIXS plot (with the same color scale) as in **c** shown on the energy transfer scale. The cyan and green colored lines in **c** and **d** visualize the cross sections of the RIXS map for constant emission energy and excitation energy at the maximum intensity of the satellite peak. Blue and orange are used for the cuts at constant energy transfer at the maximum intensity of the main resonance called white line (WL) and the satellite peaks, respectively.

two interactions are both on the order of 10 eV. Note that $4f^{13}5f^1$ is the final configuration for a RIXS process with $5f^0$ ground state, for example for $U^{VI}$.

The intensity of the singlet and triplet final states depend on the optical selection rules. These determine how, for a particular geometry, a ground-state singlet is excited into a final-state singlet or triplet. The multiplets are modified by the additional interactions within an atom, molecule or solid, but the monopole exchange interaction remains one of the dominant interaction terms for a $4f^{13}5f^1$ configuration. For $5f^n$ ground state configurations, the final state configuration will be $4f^{13}5f^{n+1}$ and the $4f$ and $5f$ spin can either couple parallel or anti-parallel. The energy splitting is given by the monopole part of the exchange interaction and determines the multiplet splitting (cf. Supplementary Figs. 2 and 3 for $4f^{13}5f^2$, $4f^{13}5f^3$ energy level diagrams). The selection rules, which determine the energy of the high-spin main peak and low-spin satellite are similar to the $5f^0$ case, albeit

modified due to different Clebsch-Gordan coefficients in the angular momentum coupling involved. As a result, there is a clear intensity dependence of the satellite peak depending on the local electron configuration of the ground-state $5f$ shell. The latter can be used to experimentally determine the electronic structure properties of actinide compounds.

### Trends in experimental and calculated actinide $M_4$ edge CC-RIXS for different scattering geometries

We explored the satellite peak for two different experimental configurations shown in Fig. 2b. The actinide $M_4$ edge CC-RIXS experiments were performed at the ACT station of the CAT-ACT beamline (Fig. 2e) and the SUL-X beamline (Fig. 2f) at the KIT Light Source, Karlsruhe Germany[31,37–39]. An X-ray emission spectrometer in vertical (ACT) or horizontal (SUL-X) Rowland circle geometry was employed to diffract the actinide $M_\beta$ fluorescence and focus it into the detector. The

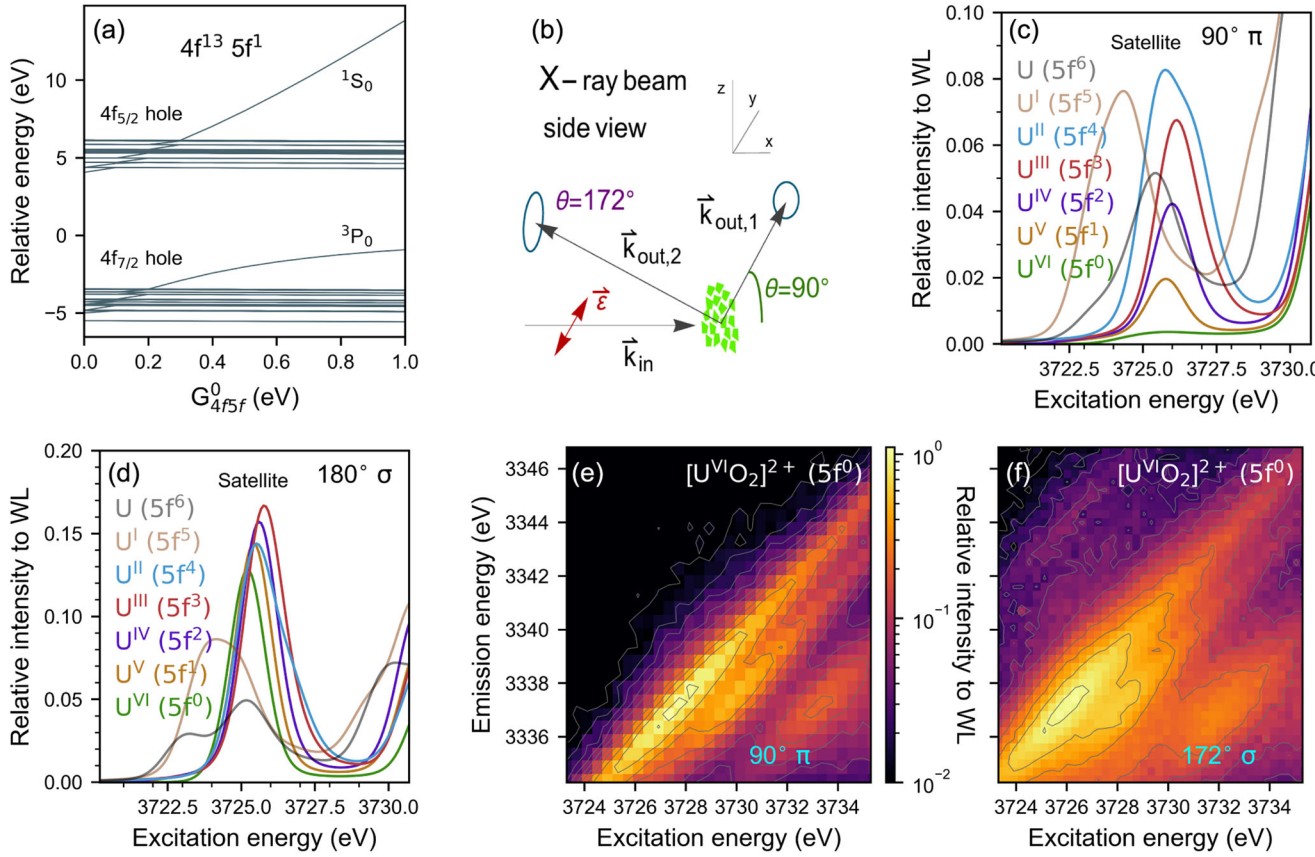

**Fig. 2 | The origin of the satellite peak and its intensity dependence on the scattering geometry. a** Calculated energy level diagram for the 4f states for an atomic $4f^{13}5f^1$ system, where only one single atom with local Coulomb interaction and spin-orbit coupling is considered. **b** Side view of the experimental setup with denoted linear polarization of the incident X-ray beam in the plane of the synchrotron (ε). The considered scattering angles θ defined between the incoming and scattered X-rays are here 90° and 172° in the horizontal (the plane of the synchrotron) or vertical scattering plane with respect to the incident X-ray polarization vector (ε). The two different experimental geometries are denoted "90° π" and "172° σ". **c, d** Cross section at constant emission energy at the maximum of the satellite peak of calculated U M₄ edge CC-RIXS map (cf. Fig. 1c, d). Portrayed are atomic calculations for U in different oxidation states/5f ground state electron configurations for **c** 90° π and **d** 180° σ scattering geometry; **e** experimentally recorded CC-RIXS maps for U^VI in ([U^VIO₂(Mesaldien)][40]) at 90° π scattering geometry and **f** the same U^VI compound measured in 172° σ scattering geometry[38].

analyzer crystals at scattering angle of 90° π and 172° σ (close to backscattering) are displayed on the scheme in Fig. 2b. The incident polarization is linear or close to linear and lies in the synchrotron plane and in the scattering plane (π geometry) or is perpendicular to the scattering plane (σ geometry). For simplicity, 180° σ is used instead of 172° σ for the calculations. Actinide M₄ edge CC-RIXS maps were calculated for the two different scattering geometries. The computed satellite peaks at fixed emission energy are plotted for the 90° π and 180° σ geometries in Fig. 2c and d. We applied atomic multiplet theory, crystal field theory, ligand field theory and DFT-based ligand field theory for the AnO₂ configurations (cf. Methods section Calculations) to investigate the origin of the intensity of the satellite peak and its relation to ground-state material properties. These levels of theory all include the full local coulomb interaction and successively include a more realistic description of the solid environment and chemical bonding. We found that basic understanding of the spectral features can be obtained from Atomic Theory (AT) combined with the material-dependent 5f electron occupation. In Fig. 2c and d we employed atomic calculations for a U atom with different number of 5 f electrons in the ground state. To be able to compare them better amongst each other, we kept for these calculations all interaction parameters constant (cf. values in Supplementary Table 4). In Fig. 2c, the relative intensity of the satellite peak compared to the WL in the M₄ edge CC-RIXS map (cf. Fig. 1c) grows from low intensity for U^VI ($5f^0$) to well

visible peak for U^II ($5f^4$). The shape of the satellite peak changes more notably for $n_{5f} > 4$. The energy position of the maximum intensity is relatively stable; it shifts non-monotonically over a range smaller than 1 eV.

The relationship between peak intensity and 5f electron occupation depends on the experimental geometry (light polarization dependent selection rules) and, as can be seen in Fig. 2d, is minor for the 180° σ scattering geometry. The satellite peak intensity increases as the scattering angle deviates from 90° π. Figure 2e and f display experimental measurement of the satellite peak for U^VI ($5f^0$ electron configuration). Here, the CC-RIXS maps in 90° π geometry (e) and 172° σ geometry (f) of solid [U^VIO₂(Mesaldien)][40], denoted as [U^VIO₂]²⁺, are compared. In Supplementary Figs. 4 and 5 this comparison can be found for slighter differences in angle changing from 90° (crystal 3) to 103.9° (crystal 1). The absolute change in intensity here is small compared to the switch to the 172° σ geometry, where the **Δ**J = 0 contribution is 16 times larger than the intensity for crystal 3. The intensity of the satellite peak is very weak for the 90° π scattering geometry (crystal 3) and there is well visible intensity for the 103.9° π scattering geometry (crystal 1) and quite prominent double peak appears for 172° σ scattering geometry (cf. Fig. 2e and f, Supplementary Figs. 4, 25). There is a slight discrepancy between the intensity and broadenings of the calculated and measured satellite peaks, which arise from an underestimation of the experimental broadening in the theoretical

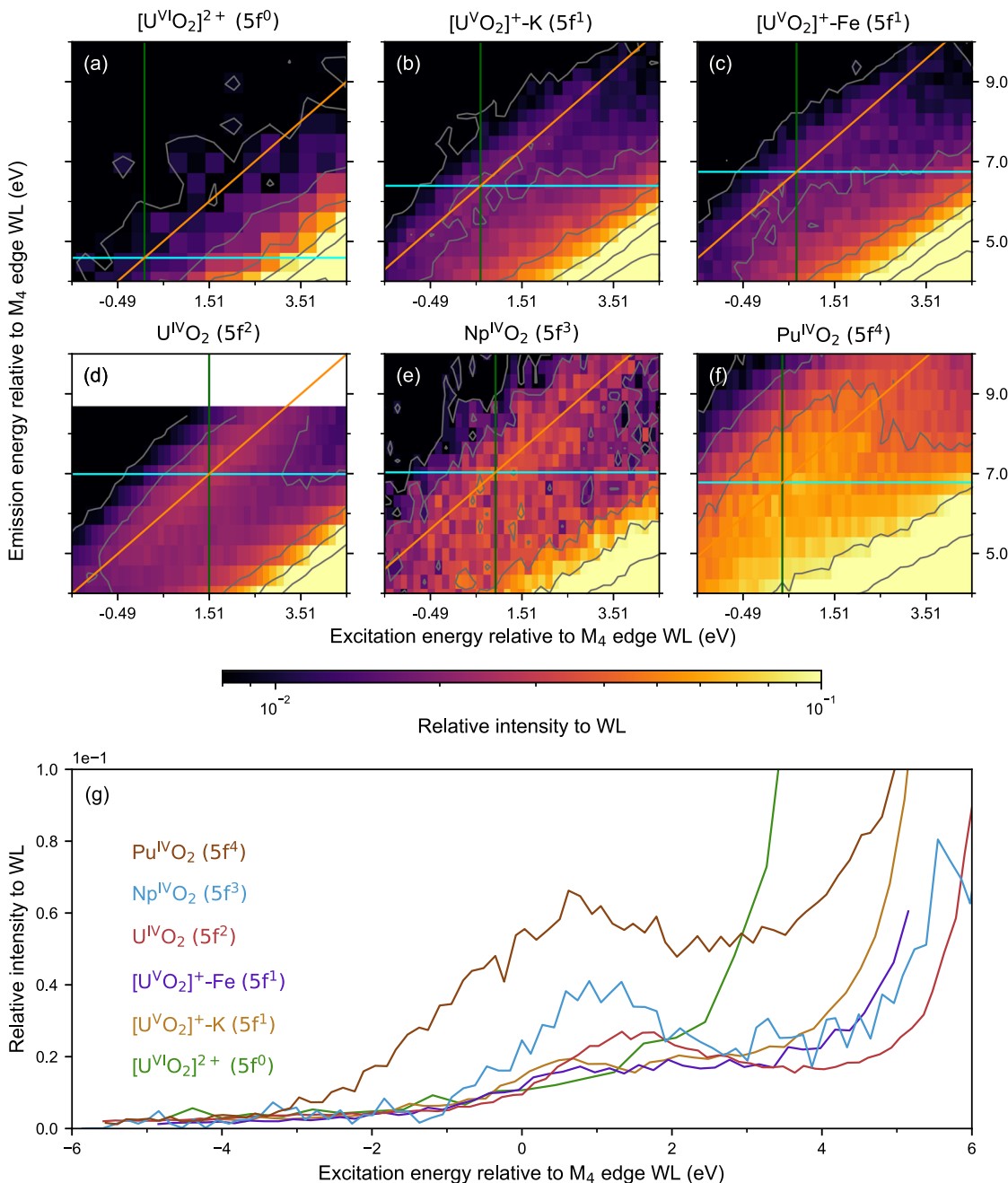

**Fig. 3 | Experimental U, Np and Pu M₄ edge CC-RIXS maps recorded at 90° π scattering geometry.** The $M_4$ edge CC-RIXS maps of **a** $[U^{VI}O_2(\text{Mesaldien})]$[40] ($[U^{VI}O_2]^{2+}$), **b** $\{[U^{V}O_2(\text{Mesaldien})]K\}_n$ ($[U^{V}O_2]^+$-K), **c** $[Fe(TPA)(Py)U^{V}O_2(\text{Mesaldien})]$ ($[U^{V}O_2]^+$-Fe), **d** $U^{IV}O_2$, **e** $Np^{IV}O_2$ and **f** $Pu^{IV}O_2$ are depicted. The color-coded intensities are plotted on the same logarithmic scale. Isointensity contour lines for the same intensity levels in **a**–**f** are visible in grey. The cyan and green colored lines visualize the cross sections of the RIXS map for constant emission energy and excitation energy at the maximum intensity of the satellite peak. Orange is used for the cuts at constant energy transfer at the maximum intensity of the satellite peaks. **g** Experimental cross sections for the samples from **a**-**f** through $M_4$ edge CC-RIXS maps recorded in a 90° π scattering geometry. The cross-sections are performed at constant emission energy at the maximum intensity of the satellite peak (cyan line in a-f).

spectra or from the simplistic nature of the pure atomic theory, which disregards possible effects caused by spherical symmetry breaking and/or bond covalency effects.

**The satellite peak in the actinide M₄ edge CC-RIXS as a tool for measuring 5f electron configurations of the actinide elements**

Figure 3a–f depicts the actinide $M_4$ edge CC-RIXS spectrum of the satellite peak for $[U^{VI}O_2(\text{Mesaldien})]$ denoted as $[U^{VI}O_2]^{2+}$, $\{[U^{V}O_2(\text{Mesaldien})]K\}_n$ denoted as $[U^{V}O_2]^+$-K[40] and $[Fe(TPA)(Py)U^{V}O_2(\text{Mesaldien})]$ denoted as $[U^{V}O_2]^+$-Fe[19], $U^{IV}O_2$, $Np^{IV}O_2$ and $Pu^{IV}O_2$.

Those compounds have variable 5f electron occupations, $n_{5f} = 0$ ($U^{VI}$), 1 ($[U^{V}O_2]^+$-K[40] and $[U^{V}O_2]^+$-Fe), 2 ($U^{IV}O_2$), 3 ($Np^{IV}O_2$) or 4 ($Pu^{IV}O_2$). The spectra were measured at 90° π geometry as displayed in Fig. 2b. It is clearly visible that the intensity of the satellite peak, relative to the WL maximum intensity, increases going from $n_{5f} = 0$, 1 to 2 for the uranium system. Increasing intensity is also observable for $n_{5f}$ changing from 2 to 4 for the An$^{IV}O_2$ system. Cross sections of the satellite peak at fixed emission energy allow for a better comparison of intensity and energy position (cf. Fig. 3g). The intensity of the peak rises as the number of f electrons increases.

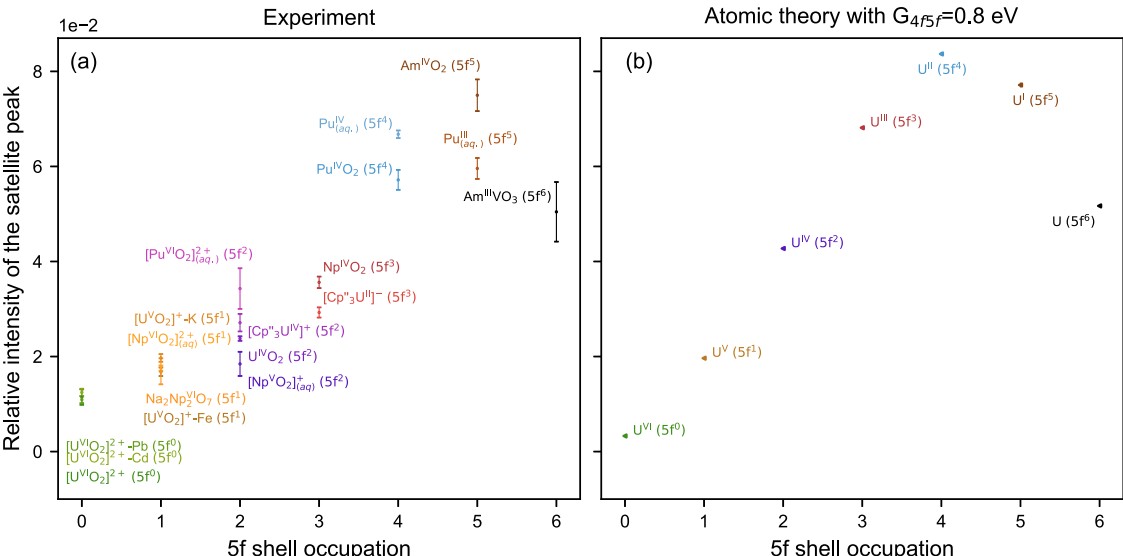

**Fig. 4 | Quantitative evaluation of the satellite intensity in a 90° π scattering geometry. a** Experimentally determined relative intensities of satellite peaks of actinide = U, Np, Pu or Am $M_4$-edge CC-RIXS maps of 18 actinide compounds sorted by 5f shell occupation. **b** Relative intensities of satellite peaks of U $M_4$ edge CC-RIXS maps obtained by performing atomic calculations of $U^{VI}$ to $U^0$ corresponding to $5f^0 - 5f^6$ occupancy. The parameters are here the same as for the calculations in Fig. 2c–d. The values for the satellite peak intensities in **a** and **b** are determined as the ratio between the areas of the curves fitted to the satellite and WL constant energy transfer cross sections (cf. Supplementary Fig. 7). The intensity error bars are calculated using a Monte Carlo like approach which is described in the methods section in detail.

To verify whether this trend is generally valid, a multitude of actinide compounds with different $5f$ electron occupations, chemical environments, and physical forms were investigated in a 90° π scattering geometry. The actinide $M_4$ edge CC-RIXS maps are normalized to the maximum of a Lorentzian function fitted to the constant energy transfer cross-section through the WL. The energy positions of the WLs and satellite peaks are displayed in Supplementary Fig. 6. These positions were determined by comparing cross sections obtained at different constant excitation, emission, or energy transfer values. The energy transfer cross sections (orange and blue traces in Fig. 1) through the maxima of the WLs and the satellite peaks were collected and modeled with Lorentzian functions (see Supplementary Fig. 7). The ratio of the areas (A) of these Lorentzian functions ($A_{sat}/A_{WL}$) were plotted for different $n_{5f}$ values in Fig. 4a. The uncertainties were computed by performing error propagation as explained in the Methods section. These also include a Monte Carlo statistical approach for the determination of uncertainties related to determination of satellite and main resonant peaks positions. A different approach using the emission energy cross section (cyan trace in Fig. 1) was also performed, it yielded similar results and can be found in Supplementary Note 5 and Supplementary Figs. 8 and 9.

**The satellite peak for $5f^0$ configuration**
We find that the chemical environment does not influence the intensity of the satellite peak for three different $U^{VI}$ compounds, despite differences in their coordination environments. $U^{VI}$ and $U^V$ form trans-dioxo bonds often referred to as uranyl ($UO_2^+ = U^V$-yl and $UO_2^{2+} = U^{VI}$-yl) with characteristic covalent axial binding of U with O ($U-O_{ax}$). These axial oxo groups are open to interaction with Lewis acids such as metal cations, which have been shown to affect the U = O bond covalency. The average $U-O_{ax}$ bond length is 1.779 Å and 1.773 Å in $[U^{VI}O_2]^{2+}$-Pb (close $Pb-O_{ax}$)(Supplementary Fig. 26a, f, h), where Pb has close interaction with $O_{ax}$, and $[U^{VI}O_2]^{2+}$-Cd (no $Cd-O_{ax}$ contact) (Supplementary Fig. 26e, g, i), where Cd has no contact with $O_{ax}$, respectively. $U^{VI}$-yl is fivefold coordinated by O's in the equatorial plane for both compounds, except in $[U^{VI}O_2]^{2+}$-Cd (no $Cd-O_{ax}$ contact) where one O is replaced by N[41,42]. The small difference in bond lengths indicates a

similar degree of covalency of the $U^{VI}$-yl bond for both compounds[43]. However, Brager et al. found through quantum theory of atoms in molecules (QTAIM) analyses that there is a small decrease in the bond covalency of the $U-O_{ax}$ bond for $[U^{VI}O_2]^{2+}$-Pb (close $Pb-O_{ax}$ contact), corroborated by distinctly different luminescence properties. This is based on the close $Pb-O_{ax}$ bond distance (2.887 Å), which leads to substantial interaction between Pb and $O_{ax}$ not present in the Cd compound ($Cd-O_{ax} = 5.681$ Å). This result is consistent with the U $M_4$ edge HR-XANES spectra (cf. Fig. 5b, Supplementary Fig. 10) where a slightly smaller ($-0.2 \pm 0.05$ eV) A-C shift between the first and third spectral peaks were observed for the $[U^{VI}O_2]^{2+}$-Pb (close $Pb-O_{ax}$ contact) and $[U^{VI}O_2]^{2+}$-Cd (no $Cd-O_{ax}$ contact) compounds[29]. This spectral shift shows sensitivity to the $U^{VI}-O_{ax}$ bond length and bond covalency[29]. In $[U^{VI}O_2]^{2+}$, the uranium atom is coordinated by 3 N and 2 O donors in the equatorial plane and the average $U^{VI}-O_{ax}$ bond length is 1.782 Å (Supplementary Fig. 12a). The spectrum of $[U^{VI}O_2]^{2+}$ has a smaller A-C shift ($5.5 \pm 0.05$ eV) suggesting a smaller $U^{VI}-O_{ax}$ bond covalency compared to $[U^{VI}O_2]^{2+}$-Pb (close $Pb-O_{ax}$ contact) ($6 \pm 0.05$ eV)/Cd (no $Cd-O_{ax}$ contact) ($6.2 \pm 0.05$ eV) (Fig. 5b, Supplementary Fig. 11b). Despite small or large modifications in the covalent character of the $U^{VI}-O_{ax}$ binding, the satellite peak remains similar, suggesting no sensitivity to the delocalized electron density binding the atoms.

**The satellite peak for $5f^1$ configuration**
For the $5f^1$ configuration the satellite peak becomes more intense. It is possible to compare the satellite peak intensity for two compounds $[U^VO_2]^+$-K and $[U^VO_2]^+$-Fe, that have a uranyl structure with the same equatorial ligand, with different Lewis acids, $K^I$ or $Fe^{II}$, axially bound to the $U^V$-yl[19]. We showed recently that the axial $U^V$-yl bond becomes less covalent, whereas the equatorial bond covalency increases upon binding of $Fe^{II}$ to the $U^V$-yl oxygen atom[19]. Despite this change in the $U^V$ binding, well visible in the U $M_4$ edge HR-XANES spectra in Supplementary Fig. 11a, the satellite peak in CC-RIXS maintains a similar intensity. Note that the U $M_4$ edge HR-XANES spectrum of $[U^VO_2]^+$-Fe is slightly shifted to higher energy compared to the $[U^VO_2]^+$-K spectrum, whereas there is only very small intensity difference for the satellite peak. This finding suggests that the satellite peak is more sensitive to

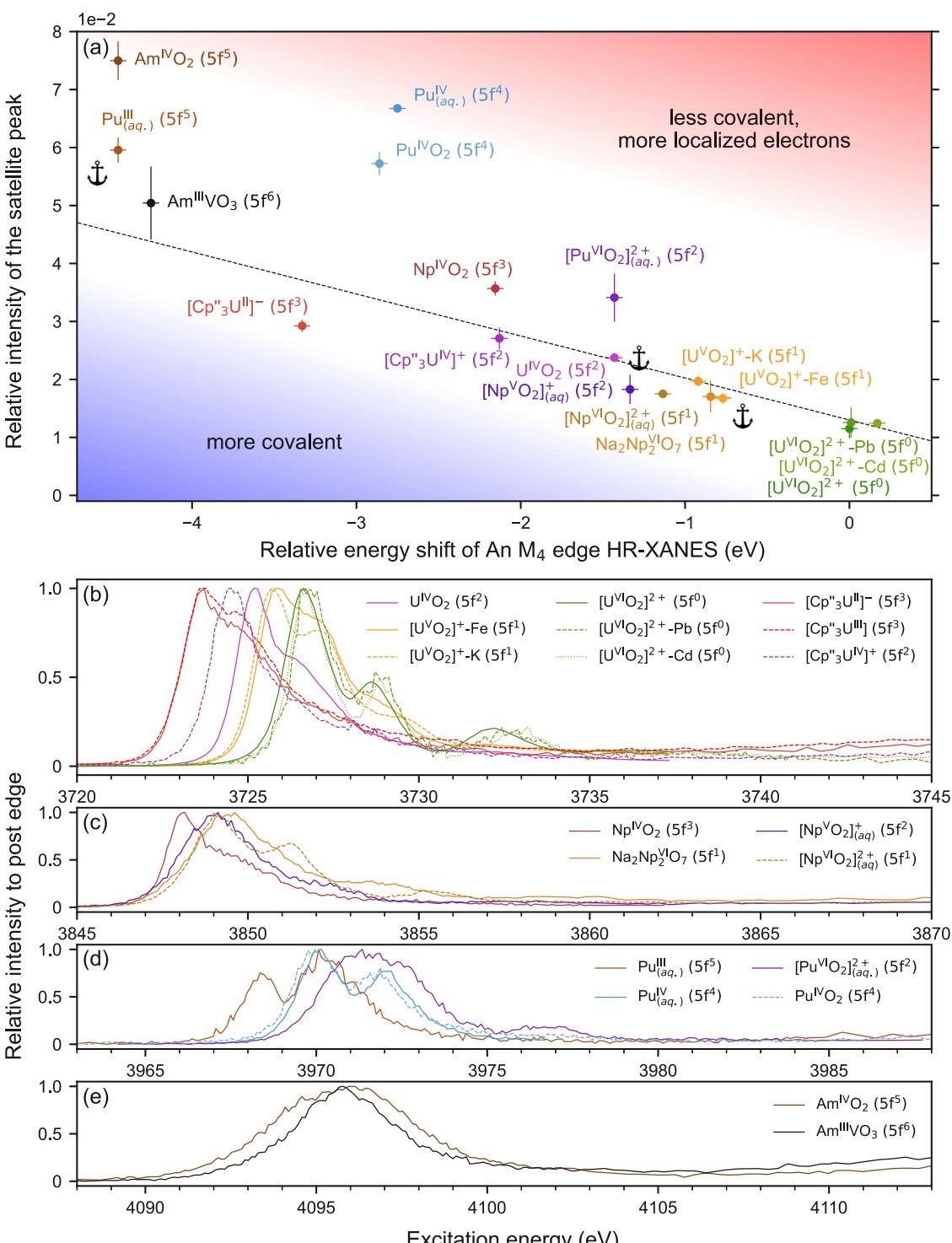

**Fig. 5 | Correlation between M₄ edge HR-XANES energy position and satellite peak intensity. a** Correlation between relative satellite peak intensity and energy shift of actinide (An=U, Np, Pu or Am) M₄ edge HR-XANES of 18 actinide compounds collected at 90° π scattering geometry. The dashed line shows a fit of the U values, excluding the [Cp"₃U$^{IV}$]⁺ and [Cp"₃U$^{II}$]⁻ compounds, which was determined to be y = (−0.72468)x + 1.30024. Similar colors indicate formal 5f shell occupation, as in Fig. 4. Relative energy shifts were applied for Np, Pu or Am according to the electron configurations determined of one representative compound of each element. [PuO₂]²⁺ (aq) (5f²) was matched to UO₂ (5f²) and this was marked with the anchor symbol in the figure. Na₂Np$^{VI}$₂O₇ (5f¹) was shifted by the average of the shift deduced from [U$^V$O₂]⁺-K and [U$^V$O₂]⁺-Fe (5f¹) denoted by the second anchor, and the Am$^{IV}$O₂ (5f⁵) was shifted by the Pu$^{III}$ (aq) (5f⁵) value denoted by the third anchor. The other Np/Pu/Am compounds were shifted relatively to the anchored values of Na₂Np$^{VI}$₂O₇ (5f¹), [PuO₂]²⁺ (aq) (5f²) and Am$^{IV}$O₂ (5f⁵), respectively. **b–e** Actinide M₄ edge HR-XANES of all compounds studied at 90° π geometry containing **b** U, **c** Np, **d** Pu, **e** Am, color-coded according to **a**. The error bars for the relative intensity are calculated using a Monte Carlo like approach which is described in the methods section in detail. The excitation energy error bars correspond to 0.5 eV.

localization of the actinide $5f$ electrons compared to the energy shift of the HR-XANES spectrum. Even more impactful is the comparison of compounds with different actinide elements but the same $5f$ occupation. $[Np^{VI}O_2]^{2+}$ (aq) and $Na_2Np^{VI}_2O_7$ also have a formal $5f^1$ configuration and essentially the same relative satellite peak intensity compared to the $[U^VO_2]^+$-K and $[U^VO_2]^+$-Fe compounds.

A comparison of the $Np^{VI}$ coordination environment of $[Np^{VI}O_2]^{2+}$ (aq) and $Na_2Np^{VI}_2O_7$ illustrates: The $Np^{VI}$-yl is equatorially coordinated by five water molecules and six oxo atoms in $[Np^{VI}O_2]^{2+}$ (aq) and $Na_2Np^{VI}_2O_7$, respectively. The $Np-O_{ax}$ and $Np-O_{eq}$ bond lengths for the two compounds ($[Np^{VI}O_2]^{2+}$ (aq) vs. $Na_2Np^{VI}_2O_7$) are $Np-O_{ax} = 1.75$ vs. $1.91\,\text{Å}$ and $Np-O_{eq} = 2.4$ vs. $2.35\,\text{Å}$[44]. In Fig. 5c, we compare the Np $M_4$ edge HR-XANES spectra for $[Np^{VI}O_2]^{2+}$ (aq) and $Na_2Np^{VI}_2O_7$ (Supplementary Note 10, Supplementary Figs. 16–18). There is a substantial change in the A-C shift from $5.2\,\text{eV}$ ($Na_2Np^{VI}_2O_7$) to $6.2\,\text{eV}$ ($[Np^{VI}O_2]^{2+}$ (aq)) that suggests increased $Np-O_{ax}$ bond covalency in $[Np^{VI}O_2]^{2+}$ (aq) and agrees with the much shorter $Np^{VI}-O_{ax}$ bond length for $[Np^{VI}O_2]^{2+}$ (aq). The small energy shift to lower energies of the Np $M_4$ edge HR-XANES spectrum for the solid compound signifies that the electron density on Np does not change notably, likely due to stronger $Np^{VI}-O_{eq}$ interactions in the solid compound, as suggested by the smaller $Np-O_{eq}$ average bond length. The similar satellite intensity can be explained by comparable localized 5 f electron density on $Np^{VI}$ in solid $Na_2Np^{VI}_2O_7$ and $[Np^{VI}O_2]^{2+}$ (aq).

### The satellite peak for $5f^2$ configuration

Here, the relation of the satellite peak intensity to the chemical environment can be demonstrated by comparing a organometallic $[Cp''_3U^{IV}]^+ = [C_5H_3(SiMe_3)_2]_3U^{IV}Cl$ coordination compound[45] to the bulk material $U^{IV}O_2$, which have very different electronic and geometric structures. The drastic $0.9 \pm 0.05\,\text{eV}$ shift to lower energy in the U $M_4$ edge HR-XANES WL position (Fig. 5b and Supplementary Fig. 13) for the spectrum of $[Cp''_3U^{IV}]^+$ is confronted with a very similar relative satellite intensity for the two compounds (Figs. 4b and 5a). Combining these two experimental results, we can conclude that there is greater, delocalized 5 f electron density for U in $[Cp''_3U^{IV}]^+$ compared to $U^{IV}O_2$. It was shown for example, in uranocene ($U[C_8H_8]_2$), that uranium bonds to aromatic systems can be rather covalent[46]. The relationship between bond covalency differences and the absorption edge energy shift can be examined in the case of Ce. The absorption edge of the Ce $L_3$ edge XANES of cerocene ($Ce[C_8H_8]_2$)[47] is several eV lower than that of a Ce(IV) standard or $Ce^{IV}O_2$ suggesting a similar effect due to the more covalent $Ce-[C_8H_8]^{2-}$ bonds in cerocene compared to Ce-O bonds in $Ce^{IV}O_2$[48]. The $[Pu^{VI}O_2]^{2+}$ (aq) and $[Np^{VI}O_2]^+$ (aq) satellite peak intensity values deviate from each other (Fig. 4a). Both $Np^V/Pu^{VI}-O_{ax}$ bonds have substantial covalent contributions. However, due to the increased nuclear charge, the $5f$ electrons become more localized for $Pu^{VI}$ and the overlap driven bond covalency is likely decreasing in $Pu^{VI}$-yl as previously suggested[29]. This is reflected in the larger satellite peak intensity revealing a larger relative number of 5 f electrons localized on $Pu^{VI}$ compared to $Np^V$.

### The satellite peak for $5f^3$ configuration

The satellite peak intensity can help to shed light on long standing discussions, such as the heavily discussed formal oxidation state of low valent actinide compounds, i.e. $An^I$, $An^{II}$ or $An^{III}$ [49–54]. Here, we measured the satellite intensity of $[Cp''_3U^{II}]^-$, which has been assigned a $5f^36d^1$ configuration based on the UV−visible spectroscopy, magnetic data, single crystal XRD and their analogy to the lanthanide compounds[55]. DFT calculations confirmed this assignment[56]. This $[Cp''_3U^{II}]^-$ is compared to the isostructural $[Cp''_3U^{III}]$ ($n_{5f} = 5f^3$) compound in Fig. 5b and Supplementary Fig. 13. The resulting spectra yielded very similar WL positions, corresponding to similar electron density on U. Additional VB-RIXS experiments would need to be performed to identify the 6d portion of the electron occupation in $[Cp''_3U^{II}]^-$, however, an a $^1H$ NMR

spectrum of the compound recorded after the shipment is very similar to a pristine sample from the literature (Supplementary Fig. 14), confirming the retention of the $U^{II}$ compound. We find a $1.0\,\text{eV}$ energy shift to lower energies of the U $M_4$ edge HR-XANES spectrum for $[Cp''_3U^{III}]$ and $[Cp''_3U^{II}]^-$ compared to the spectrum of the structurally similar $[Cp''_3U^{IV}]^+$ (cf. Fig. 5b and Supplementary Fig. 13). Interestingly, the satellite peak intensities are similar for both $[Cp''_3U^{II}]^-$ and $[Cp''_3U^{IV}]^+$ compounds, suggesting rather delocalized part of the larger $5f$ electron density on U in $[Cp''_3U^{II}]^-$. The satellite peak intensity of the organometallic compounds are slightly smaller than that of $Np^{IV}O_2$ ($n_{5f} = 5f^3$).

### The satellite peak for $5f^4$ and $5f^5$ configurations

Another important example is the comparison of $Pu^{IV}$(aq) and $Pu^{IV}O_2$. The satellite peak intensity is larger for $Pu^{IV}$(aq) suggesting more localized $5f$ electron density on Pu compared to $Pu^{IV}O_2$. The Pu $M_4$ edge HR-XANES spectra shown in Fig. 5d and Supplementary Fig. 20 exhibit a small energy shift to higher energies for $Pu^{IV}$ (aq) and a small, similar shift is present for the Pu $M_5$ edge HR-XANES data, revealing that the $5f$ electron density on Pu is similar but slightly lower for $Pu^{IV}$ (aq)[57]. Quantum chemical calculations have reported $Pu^{IV}O_2$ is a solid compound with larger Pu-O bond covalency opposed to the more ionic $Pu^{IV}-OH_2$ chemical bond in $Pu^{IV}$ (aq). Compared to other compounds in this study, the level of bond covalency in $Pu^{IV}O_2$ is small, which agrees well with our previous experimental results[57]. For the $5f^5$ case, we see the largest discrepancies from the theoretical predictions where $Am^{IV}O_2$ displays a higher intensity than the $5f^4$ systems. However, this can be partially rationalized since the non-ideal geometry and conditions of the experiment for this absorption edge introduced larger uncertainties in the determination of the satellite peak area, by increasing overlap between WL and satellite. The satellite peak area for $Pu^{III}$ (aq) has lower intensity than for $Pu^{IV}$ (aq) and thereby, it is consistent with the theoretical prediction in Fig. 4b.

### The satellite peak for $5f^6$ configuration

For $5f^6$ electron configurations, the theory predicts a break in the upwards trend and a much lower intensity than the previous $5f$ occupations The intensity of the satellite peak is given by Clebsch-Gordan coefficients related to the angular momentum coupling of the valence and core states. The values of $<L^2>$ and $<S^2>$ for the core states are "fixed" as it is a single hole in the $4f$ shell. Thus the intensity depends on the values of $<L^2>$ and $<S^2>$ in the $5f$ shell, which scale with the number of electrons. For a half-filled shell the expectation value of $<L^2>$ becomes zero and the angle between L and S changes. This leads to a saturation in the trend that one starts to see for the $5f^5$ compounds. The experimentally determined relative satellite intensity of $Am^{III}V^{III}O_3$ amounts to $5.0 \pm 0.6\%$ of the WL intensity, quantitatively matching the theoretically predicted value of $5.2 \pm 0.2\%$ for $5f^6$. The Pu and Am $M_5$ and/or $M_4$ edge HR-XANES data for the $5f^4$, $5f^5$ and $5f^6$ compounds are depicted in Fig. 5d and e, Supplementary Figs. 20–22. It is remarkable that the Am $M_4$ edge HR-XANES exhibits only a very small energy shift of about $0.2 \pm 0.05\,\text{eV}$ for $Am^{III}$ and $Am^{IV}$ reported here for the first time. More specifically, the broadening of the $Am^{IV}$ WL is large and appears shifted to lower energy. A clear energy shift of $-1.35 \pm 0.05\,\text{eV}$ in the Am $M_5$ edge spectra of $Am^{III}V^{III}O_3$ compared to the $Am^{IV}O_2$ spectrum is visible in Supplementary Fig. 22b. The Pu $M_4$ edge HR-XANES spectrum of $Pu^{III}$ (aq) is reported here also for the first time.

### Comparison of satellite peak intensity in the actinide $M_4$ edge CC-RIXS and HR-XANES energy shift

Our results show that the satellite peak measured at $90°$ $\pi$ scattering geometry is sensitive to the number of $f$ electrons localized on the actinide atom, and its integral intensity (area of the cross-section of the peak) can be used to obtain their relative count (cf. Fig. 4). We have plotted the relation between the relative satellite peak intensity for the

90° π geometry and the relative absorption edge shift of the actinide $M_4$ edge HR-XANES for the U, Np, Pu, and Am compounds (Fig. 5a). The dashed line shows a fit of the U values, excluding the $[Cp''_3U^{IV}]^+$ and $[Cp'''_3U^{II}]^-$ compounds, which deviate from the linear trend for U. We observe that the values for some compounds of Np and Pu deviate from this trend and are shifted to higher satellite peak intensities (above the dashed line). In contrast, the values for $[Cp''_3U^{IV}]^+$ and $[Cp'''_3U^{II}]^-$ are strongly shifted to larger, relative energy shifts of the $M_4$ edge HR-XANES (below the dashed line).

Considering that the energy position of the HR-XANES absorption edge and the satellite peak intensity probe electron density with different distributions between the actinide and the ligands, we can use the relationship between the relative shift of the HR-XANES to the satellite peak intensity as a tool to compare the level of localization on the actinide atom or the involvement of the actinide $5f$ electrons in bond covalency. Specifically, points above the dashed line represent compounds with a stronger level of localization of the $5f$ electrons, such as in Pu compounds. Compounds trending toward the lower left corner with strong relative $M_4$ edge HR-XANES shifts, such as $[Cp''_3U^{IV}]^+$ and $[Cp'''_3U^{II}]^-$, are characterized by a larger delocalization of the $5f$ electrons participating in covalent bonds. The $5f^5$ and the $5f^6$ compounds are above the line; however, it should be noted that they have a decreasing satellite peak intensity compared to $5f^4$, as suggested by experiment and theory shown in Fig. 4.

## Actinide $M_4$ edge CC-RIXS as a tool for benchmarking quantum chemical calculations

The CC-RIXS maps at the $M_4$ edge are valuable to benchmark different calculations due to their sensitivity to different interactions. In the previous sections, it is shown that the spherical $4f$–$5f$ exchange strength parameter $G^0_{4f5f}$ influences the energy position of the satellite peak and that the $5f$ occupation in the ground state defines the intensity of the peak in a 90° π scattering geometry. The first conclusion is further confirmed in Fig. 6, which presents simulated $M_4$ edge CC-RIXS maps for $Pu^{IV}O_2$ powder. Here, multiplet crystal field theory (CFT, cf. Methods section Calculations) was adopted. In the four panels, all interactions were kept constant and only $G^0_{4f5f}$ was changed from 0 eV to 0.9 eV. For $G^0_{4f5f} = 0$ eV, no satellite is visible. With increasing $G^0_{4f5f}$ the feature gains in intensity and moves to lower emission energies. The same behavior can be observed in 180° σ scattering geometry (cf. Supplementary Fig. 24). Analogously to Fig. 2a, a second bunch of satellite states, this time at lower emission energies than the WL, detach from the $M_4$ WL ($4f_{5/2}$ hole character) complex and affect the shape of its shoulder.

In multiplet models for correlated materials, ab initio values for the Coulomb parameters have to be reduced to agree with experimentally observed values known as the Nephelauxetic effect. This reduction is partly due to screening of the Coulomb interaction due to the charges not included in the active basis[58]. The screening of the monopole part of the Coulomb interaction, i.e., $G^0_{4f5f}$ and $F^0_{4f5f}$ is normally larger than the screening of the higher multipole interactions. This yields an unexpected possibility for the practical use of $M_4$ edge CC-RIXS spectra, normally only partially available from core level XPS. The energy position of the satellite can be used to identify the correct value for $G^0_{4f5f}$ for a known actinide oxidation state. It is a more versatile tool compared to XPS since it can be performed on materials in any state of matter without the need for ultra-high vacuum conditions often modifying the chemical state of surface species. Note that XPS is a surface-sensitive technique, whereas U $M_4$ edge RIXS mainly probes the bulk of the material.

In Fig. 6e the $G^0_{4f5f}$ dependency on the satellite peak position and intensity for $U^{IV}O_2$ is investigated and compared with CFT calculations. The cuts of the $M_4$ edge CC-RIXS map are at constant excitation energy (cf. the green cut in Fig. 1c). In agreement with Fig. 2a, increasing $G^0_{4f5f}$ shifts the satellite position to lower emission energy. In addition, the

intensity of the feature increases with $G^0_{4f5f}$ and offers a second measure for the validation of $G^0_{4f5f}$ values in calculations. The theory reveals the presence of a second satellite region located in the WL high-energy tail, which is not observed in the experimental spectra at the 90° π scattering geometry. This could be due to overestimation of the calculated distance between the satellites or to additional effects not considered in the theory. However, the predicted feature is visible in 172° σ and for the first time shown here (cf. Fig. 7). In Fig. 6e this second satellite has a factor 4-5 larger intensity than the satellite peak and strongly affects the shape of the calculated WL. The same applies to the measurement in backscattering geometry (cf. Fig. 7 and for Supplementary Fig. 25).

Both the energy position and intensity of the satellite peak are sensitive to the $4f$–$5f$ exchange strength, $G^0_{4f5f}$. For the $An^{IV}O_2$ compounds, this can be illustrated by plotting the experimentally determined and calculated values for the relative energy versus relative intensity. This is presented for different elements and different $G^0_{4f5f}$ values in Fig. 6f. CFT simulations systematically overestimate distances and intensities, but the general trend is obvious. The highest deviation is observed for $Pu^{IV}O_2$, where a significant experimental broadening of the data may bias the comparison. Ligand Field Theory (LFT) calculations did not improve this comparison much for $Pu^{IV}O_2$. Keeping this overestimation in mind, the plot can be used to estimate effective $G^0_{4f5f}$ values for the different elements. At the CFT level of theory, these are rounded to $G^0_{4f5f} = 0.7$ eV for U, $G^0_{4f5f} = 0.6$ eV for Np, and $G^0_{4f5f} = 0.9$ eV for Pu. Note, $G^0_{4f5f}$ was calculated ab initio to be between 1.1 eV for U and 1.2 eV for Pu (cf. Supplementary Table 4). It is necessary to mention that different level of theories generally need different parameters to describe physical reality. LFT partially includes screening mechanisms by explicit consideration of charge transfer configurations in the model. The larger active basis in LFT compared to the CFT set results in a smaller screening of ab initio values. We propose to use the energy position of the satellite feature and in general, the line shape of the spectra as an experimental tool to determine the amount of scaling needed for $G^0_{4f5f}$ in computations.

## Actinide $M_4$ edge CC-RIXS as a tool for probing bond covalency of the actinide-ligand bond

In the case of a covalent bond leading to a larger delocalized electron density between the actinide atom and the ligand, the $G^k_{4f5f}$ parameters are screened[32]. The result is that for a more covalent bond, the intensity of the satellite peak is expected to decrease at 172° σ for which the satellite peak intensity is 2-20 times higher than in the 90° π scattering geometry. The calculations illustrate the effect since the intensity of the satellite peak for equal $G^k_{4f5f}$ decreases at a Ligand Field Theory level for $Pu^{IV}O_2$ (cf. Fig. 6f). The tendency for 90° π is strongly dependent on the number of $5f$ elecron in the ground state, while it is not the case in backscattering geometry. There it is clearly visible that it is at his maximal relative intensities for all simulated $5f$ occupations (cf Fig. 2b). We can conclude that the intensity of the satellite peak at backscattering geometry appears sensitive to the electron density transferred from the ligand to the actinide metal. Therefore, it could be used potentially as a tool to probe bond covalency.

To verify this hypothesis, we recorded U $M_4$ edge CC-RIXS maps at a 172° σ scattering geometry for a series of $U^{VI}$ compounds. In addition to the $[U^{VI}O_2]^{2+}$-Pb (close Pb-$O_{ax}$) and $[U^{VI}O_2]^{2+}$-Cd (no Cd-$O_{ax}$ contact) compounds discussed above, $[U^{VI}O_2]^{2+}$-M compounds with or without direct $O_{ax}$ interaction (M = Pb, Ag or Cd) and coordinated by five or six equatorial O donors were studied (Supplementary Fig. 25). Details on the coordination structures are given in Supplementary Fig. 26 [41–43]. Furthermore, we performed ligand field density function theory (LFDFT) calculations of the U $M_4$ edge CC-RIXS maps. This theoretical approach uses DFT and the average of configuration-type calculations that enables the evaluation of ligand-field parameters without scaling factors or empirical corrections. The experimental and calculated U $M_4$

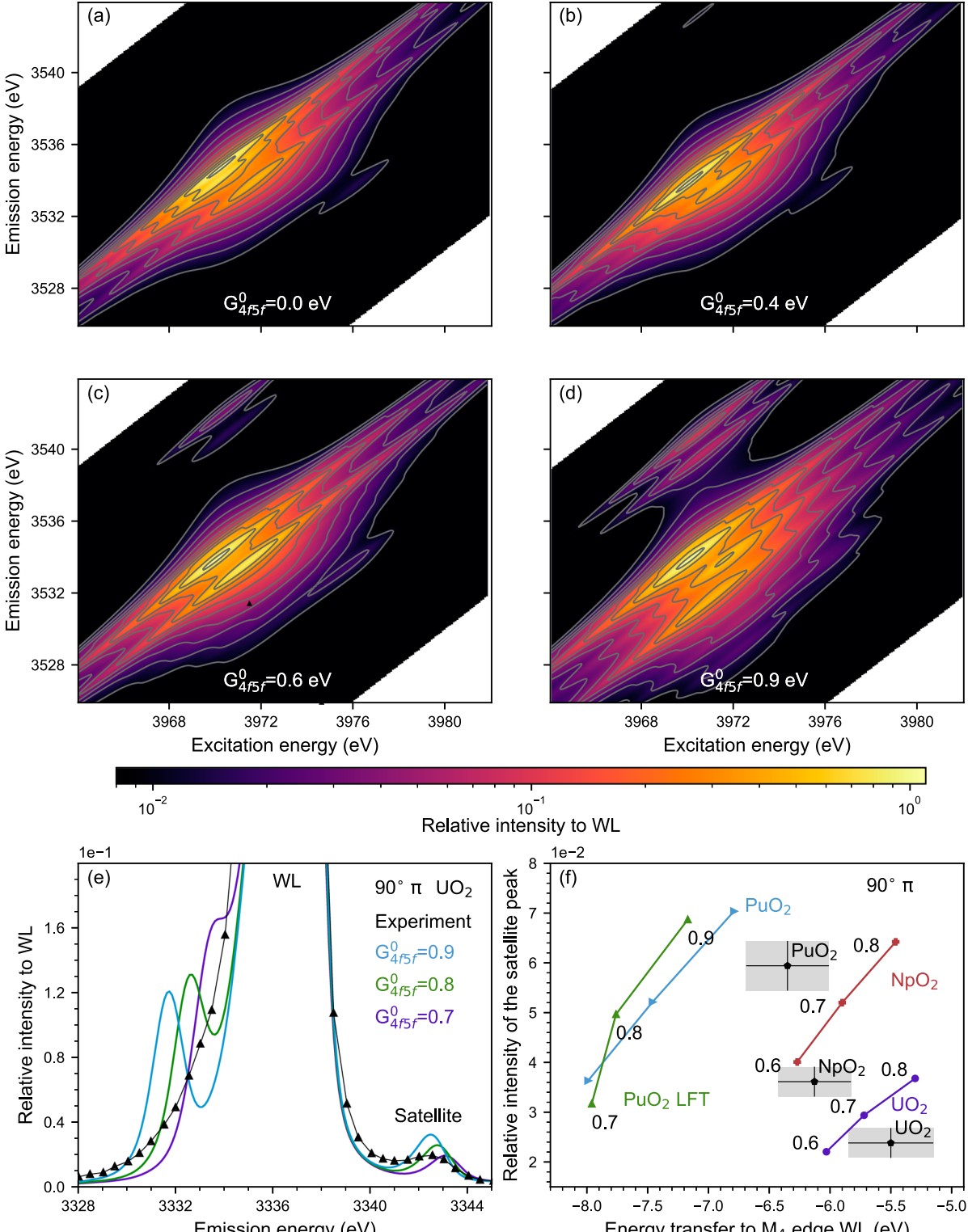

**Fig. 6 | The influence of $G^0_{4f5f}$ for M$_4$ edge CC-RIXS maps. a-d** Simulated M$_4$ edge CC-RIXS maps for Pu$^{IV}$O$_2$ for a 90° π scattering geometry. The CFT calculations only differ in the chosen value of $G^0_{4f5f}$ = {0.0, 0.4, 0.6 or 0.9} eV. The spectral intensities are plotted using one logarithmic scale and isointensity curves are drawn at fixed values. In these plots only lifetime and no experimental broadening was accounted for. **e** Experimental and theoretical cross sections at constant excitation energy at the maximum of the satellite peak (cf. Fig. 1 c, d) for U$^{IV}$O$_2$. The calculations are performed for different $G^0_{4f5f}$ parameters. **f** Relative intensity, determined as in Fig. 4, and energy position of the satellite peaks in U$^{IV}$O$_2$, Np$^{IV}$O$_2$, and Pu$^{IV}$O$_2$ M$_4$ edge CC-RIXS maps experimentally recorded in comparison to crystal field theory (CFT) calculations for various $G^0_{4f5f}$ parameters for U$^{IV}$, Np$^{IV}$ and Pu$^{IV}$ and to ligand field theory (LFT) calculations for Pu$^{IV}$O$_2$.

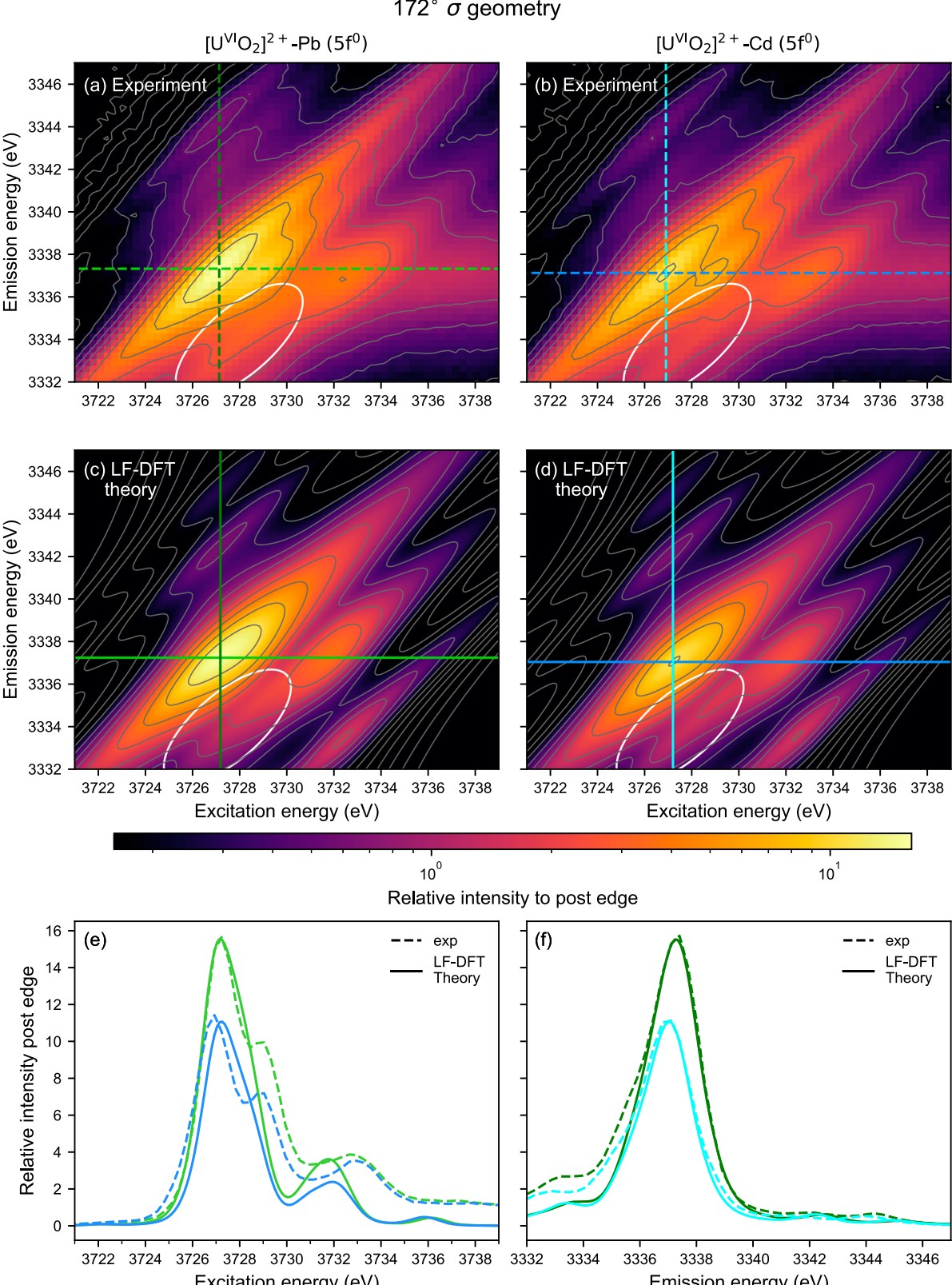

**Fig. 7 | U M$_4$ edge CC-RIXS maps at a 172° σ scattering geometry for U$^{VI}$ compounds. a–d** U M$_4$ edge CC-RIXS maps of **a, c** [U$^{VI}$O$_2$]$^{2+}$-Pb (close Pb-O$_{ax}$) and **b, d** [U$^{VI}$O$_2$]$^{2+}$-Cd (no Cd-O$_{ax}$ contact), **a, b** are obtained experimentally whereas **c, d** are calculated using ligand field density function theory (LFDFT). All CC-RIXS spectra are plotted on a logarithmic scale and the ellipsoids mark the emerging satellite intensity at low emission energy. **e, f** Cross sections at the **e** maximum intensity of the main resonance peak at fixed emission (light green lines in **a, c** and dark blue lines in **b, d**) or **f** excitation energy (dark green lines in **a, c** and cyan lines in **b, d**). Dashed and solid lines are used for the experimental (**a, b**) and calculated (**c, d**) CC-RIXS spectra, respectively. The calculated intensities of the main resonance peaks in **c** and **d** are scaled to match the intensities of the corresponding experiments in **a** and **b**.

edge CC-RIXS maps and cross sections show variations for the $[U^{VI}O_2]^{2+}$-Pb (close Pb-O$_{ax}$) and $[U^{VI}O_2]^{2+}$-Cd (no Cd-O$_{ax}$ contact) compounds displayed in Fig. 7 (cf. Supplementary Figs. 25, 28). In Fig. 7f, the cross sections at constant excitation energy through the maxima of the main resonance peaks are shown. They are similar for experiment and theory and reveal two satellites on the top and one satellite on the bottom (circled) of the CC-RIXS maps. There is one satellite at the bottom of the calculated CC-RIXS maps that is only faintly observed in the experiment.

To decipher the influence of the inter-electron repulsion on the satellite peaks, we performed two different types of calculations (1) omitting both $3d-5f$ and $4f-5f$ inter-electron interaction parameters (Supplementary Fig. 29e-h) or (2) implementing 3d-5f but still not considering $4f-5f$ interactions (Supplementary Fig. 29i-l). The results depicted in Supplementary Fig. 29 illustrate that the two satellite peaks equally appear but only when the $4f-5f$ interaction is considered. This result confirms that the integral intensity of the satellite peaks measured in a 172° σ scattering geometry is sensitive to the size of the $4f-5f$ inter-electron repulsion interaction. In turn, the size of the $4f-5f$ interaction depends on the $5f$ electron density distribution between U and the ligands. The larger the U-ligand bond covalency, the smaller the $4f-5f$ repulsion interaction as predicted by the Nephelauxetic effect[59], which is correlated with smaller integral intensity of the satellite peaks. It states that for a free atom, the inter-electron repulsion parameters are the largest possible. When the atom is coordinated by ligands, the value of interelectron repulsion parameter ($4f-5f$ in our case) ultimately becomes smaller than in the free atom case since the $4f-5f$ repulsion decreases; the extent of this reduction is related to the level of delocalization of electron density between the metal center and the ligand.

The integral intensity of the two satellite peaks is 20.5% smaller for $[U^{VI}O_2]^{2+}$-Cd (no Cd-O$_{ax}$ contact) if compared to $[U^{VI}O_2]^{2+}$-Pb (close Pb-O$_{ax}$ contact) (Supplementary Fig. 27), which suggests larger U-ligand bond covalency in $[U^{VI}O_2]^{2+}$-Cd. The experimental U $M_4$ edge CC-RIXS maps were normalized to the post-edge intensity for these analyses

(cf. Methods). Figure 8 shows comparison between the relative intensity of the satellite peaks for the 172° σ scattering geometry and reported QTAIM[60] metric values for the $[U^{VI}O_2]^{2+}$ system (see Supplementary Table 8). The electron density (ρ) and delocalization index (δ) metrics were chosen for the analysis. They were determined for the U-O$_{ax}$ and U-equatorial ligands bonding interactions at their respective bond critical points. The bonding interaction is essentially dominated by the U-O$_{ax}$ interaction (larger ρ and δ values for the five $[U^{VI}O_2]^{2+}$ complexes). But the contribution of the U-equatorial ligand interaction is also not negligible and it presents the biggest disparity between the different complexes. Compounds with five-coordinated equatorial ligands $[U^{VI}O_2]^{2+}$-Cd (no Cd-O$_{ax}$ contact) and $[U^{VI}O_2]^{2+}$-Pb close Pb-O$_{ax}$) have strong U-equatorial ligand bonds and at the same time slightly weaker U-O axial bonds. In turn, these compounds are mainly associated with smaller satellite peak intensity (yellow fields in Fig. 8). On the other hand, compounds with six-coordinated equatorial ligands $[U^{VI}O_2]^{2+}$-Ag (close Ag-O$_{ax}$), $[U^{VI}O_2]^{2+}$-Ag (long Ag-O$_{ax}$) and $[U^{VI}O_2]^{2+}$-Pb (no Pb-O$_{ax}$) have weak U-equatorial ligand bonds and slightly stronger U-O axial bonds, which are mainly associated with larger satellite peak intensity. Note that $[U^{VI}O_2]^{2+}$-Ag (close Ag-O$_{ax}$) slightly deviates from this trend, which probably arises from the effect of the Ag$^+$ ion on the overall electronic structure. The variations of the intensity of the satellite peak at the 172° σ scattering geometry compared to the lack of intensity at the 90° π scattering geometry corroborates the theoretical prediction that the peak at backscattering geometry will be sensitive to actinide-ligand bonding situation. It remains the question if the decrease in satellite peak intensity is potentially more sensitive to the metal-equatorial ligands bond covalency specifically for actinyls as Fig. 8 suggests; smaller satellite peak intensity for larger electron density (ρ) and delocalization index (δ) for the compounds in the yellow field compared to the compounds in the green field in Fig. 8b.

## Discussion

The experimental and computational results show that the complex electronic structure and binding properties of the actinide elements

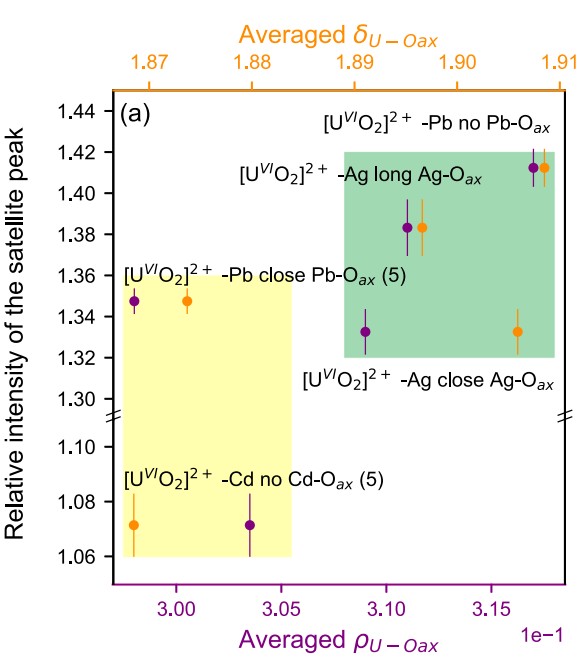
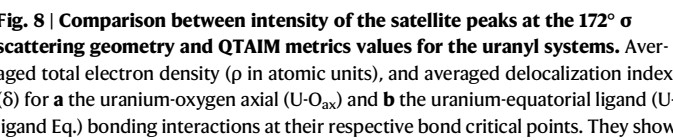
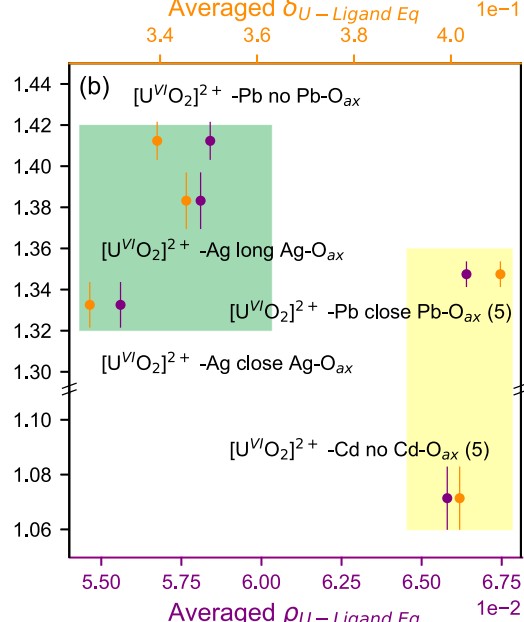

**Fig. 8 | Comparison between intensity of the satellite peaks at the 172° σ scattering geometry and QTAIM metrics values for the uranyl systems.** Averaged total electron density (ρ in atomic units), and averaged delocalization index (δ) for **a** the uranium-oxygen axial (U-O$_{ax}$) and **b** the uranium-equatorial ligand (U-ligand Eq.) bonding interactions at their respective bond critical points. They show overall bonding strength as function of the compounds, i.e., large ρ and δ values indicate stronger uranium-ligand bond. The yellow and green rectangles depict the coordination number of the U center in the equatorial plane, i.e. 5 and 6, respectively. Note that (5) signifies that in the equatorial plane of $[U^{VI}O_2]^{2+}$-Cd (no Cd-O$_{ax}$ contact) and $[U^{VI}O_2]^{2+}$-Pb (close Pb-O$_{ax}$), the U center is penta-coordinated.

can be revealed by advanced spectroscopic tools and computations. We discussed the origin and use of the satellite peak in the actinide $M_4$ edge CC-RIXS maps shifted to higher emission energy with respect to the main resonance. We showed that the satellite peak results from the competition between a j-j coupled state favored by the core-level 4 $f$ spin-orbit coupling and an LS coupled state favored by Coulomb interaction, induced by the spherical part of the $4f$–$5f$ exchange interaction $G^0_{4f5f}$. It was also demonstrated that the intensity of the satellite peak is very sensitive to the scattering geometry. It increases with the increasing number of $5f$ electrons localized on the actinide element for the 90° $\pi$ scattering geometry. Very small intensity was found for $n_{5f} = 5f^0$ ($U^{VI}$) and steady intensity increase for $n_{5f} = 5f^{1-4}$ was illustrated. The upward trend was shown to be broken and followed by decreasing intensity for $n_{5f} = 5f^5$ and $n_{5f} = 5f^6$, which is a direct consequence of Hund's rule where we find the expectation value of $L^2$ to be zero for a half-filed shell. The experimental and theoretically predicted experimental trends generally agree (cf. Fig. 4). Very small change of intensity for variations of $U^V$-$O_{ax}$ (uranyl) bond covalency was found by comparing spectra for two $U^V$ compounds with systematic modification of the coordination sphere and small differences in binding covalent interactions. Various additional examples, totaling eighteen U, Np, Pu and Am compounds, provide strong evidence that the satellite peak is sensitive mainly to the electrons localized on the actinide element for the 90° $\pi$ scattering geometry. This sensitivity can be used to determine actinide electron configurations or formal oxidation states.

The classical tool used for oxidation state determination is the chemical shift of the absorption edge of an X-ray absorption near edge structure spectrum[29,32,61]. However, it is influenced by both localized electrons on the metal and partially delocalized electron density distributed between the bonding partners. This makes the satellite peak a refined tool for counting the $5f$ electrons compared to the absorption edge position of conventional or high-energy resolution XANES (HR-XANES). We showed that the relation between the relative satellite peak intensity and the relative actinide $M_4$ edge HR-XANES absorption edge shift can help to identify the level of localization or participation of the $5f$ electrons in covalent bonds. Recently, we illustrated that U $M_4$ edge valence band RIXS (VB-RIXS) is sensitive to the $5f$ electron density in the occupied valence band, which is only the delocalized 5f electron density (strictly correct for the early actinides up to Pu), thus it is a complementary tool to the satellite peak studies we present here[19]. Measuring the satellite peak can also be used for low valent compounds to unambiguously determine the formal oxidation state of the actinide element[19,30,32,62,63].

Furthermore, we illustrated that the intensity of the satellite peak increases at non-90° $\pi$ scattering geometry, with the largest peak intensity observed at 172° $\sigma$ and calculated for 180° $\sigma$ scattering geometry. In this backscattering geometry, we observed the presence of two satellite peaks for $U^{VI}$-yl compounds. LFDFT calculations of U $M_4$ edge RIXS maps manifested that the two satellite peaks originate from $4f$–$5f$ interactions. Larger delocalization of electron density between U and the ligand, leads to weaker $4f$–$5f$ interaction and lower satellite peak intensity as resulting from the Nephelauxetic effect. This is valid for any experimental geometry. Showing the largest satellite intensities and no dependency on the formal $5f$ occupation, $\sigma$/$\pi$ backscattering geometries (or similarly 90° $\sigma$) offer a suitable way to explore sensitivity on the screening of $4f$–$5f$ interactions.

By comparing the satellite peak intensities for five $U^{VI}$-yl compounds, where $U^{VI}$-yl has similar coordination environments, we found pronounced differences. We argued that those are likely related to differences of the U-ligand bond covalency. It was shown by Brager et al.[43] and QTAIM results discussed here that there is stronger interaction for U-$O_{eq}$ bonds in $[U^{VI}O_2]^{2+}$-M (M = Pb (close Pb-$O_{ax}$) or Cd (no Cd-$O_{ax}$)); this U-$O_{eq}$ interaction increases for five- versus six-fold $O_{eq}$

coordination of U. Considering this, the increase of U-$O_{eq}$ covalency appears associated with the decreased intensity of the satellite peak. This is specifically visible by comparing QTAIM metrics for $[U^{VI}O_2]^{2+}$-Cd with no Cd-$O_{ax}$ bonding exhibiting small integral satellite peak intensity, whereas it is large for $[U^{VI}O_2]^{2+}$-Pb with Pb-$O_{ax}$ interaction. In both compounds the U is 5-fold coordinated in the equatorial plane.

Neidig et al.[2] described the bond covalency between metal and ligand with the mixing coefficient $\lambda_{ML}$. It is dependent on the orbital overlap ($S_{ML}$) and the energy degeneracy of metal and ligand valence orbitals ($E^0_M - E^0_L)^2$.

$$\lambda_{ML} \propto \frac{S_{ML}}{E^0_M - E^0_L} \qquad (1)$$

We anticipate that the satelite peak intensity measured in back-scattering geometry will be valuable for probing the overlap part of the electron density in the actinide-ligand chemical bonds and thus is a valuable contribution to the toolbox of spectroscopic instruments for independent measurements of the mixing coefficient, orbital overlap and energy match of metal and ligand valence orbitals influencing the bond covalency[2,29,64]. It is intensively discussed how to measure these different contributions to bond covalency[2,4,5,29,65,66].

The satellite peak can be also effectively used to benchmark the $4f$–$5f$ exchange interaction $G^0_{4f5f}$ parameter in quantum chemical computations. Precise $5f$ configurations are needed to understand and tune physical properties like magnetism of single molecular magnets or transition energies of superconductors based on actinides. There is currently a huge effort from synthetic chemists and spectroscopists towards developing experimental strategies and tools for probing the binding behavior of the actinides[66–75]. One of the aims is to answer the still open question: what is the actual relation between bond covalency and bond stability/reactivity for the actinide elements? The undiscovered or not well-understood properties of the actinide elements are driving force and motivation to further develop modern spectroscopic tools. We anticipate that the herein proposed new tool at different scattering geometries, probing both $5f$ electron configurations and bond covalency, will have an important impact in the modern efforts to advance our understanding of the electronic structure, binding and ultimately the physical and chemical properties of the actinide elements.

## Methods
### Materials and synthesis of compounds
Deionized water (18.2 MΩ·cm; Milli-Q Plus, Merck Millipore, Germany) and chemicals of analytical grade or higher were utilized.

$U^{IV}O_2$, $Np^{IV}O_2$, and $Pu^{IV}O_2$. The $^{nat}U^{IV}O_2$ sample consisted of a quarter of a sintered pellet originally reduced under Ar/H according to Kelm et al.[76]. It was encapsulated by two layers of 12.5 µm Kapton[77]. The $U^{IV}O_2$ sample was stored for several years under ambient conditions. $^{237}Np^{IV}O_2$ was synthesized via the oxalate decomposition route described in Popa et al.[78,79]. Np(IV)-oxalate was precipitated by adding stochiometric amounts of 1 M oxalic acid to 0.6 M Np(IV) in 2 M $HNO_3$. The precipitate was washed with distilled water several times to remove nitrate traces and transferred to a hydrothermal synthesis autoclave reactor with 5 mL of water. The mixture was heated for 18 h at 200 °C resulting in the formation of nano-$NpO_2$. The product was washed with water, ethanol and acetone and the purity was confirmed by pXRD and Raman spectroscopy[78,79]. The reaction yielded nano-crystalline $Np^{IV}O_2$ powder, which was glued on a sample holder and encapsulated by two layers of 12.5 µm Kapton. $^{239}Pu^{IV}O_2$ in the form of bulk power was encapsulated by covering it with two Kapton layers with 12.5 µm and 8 µm thickness. The $U^{IV}O_2$ and the $Pu^{IV}O_2$ were also used as reference materials in the following publications[19,29,80].

**Np$^V$ reference.** was obtained by precipitating U$^{VI}$ and Np$^V$ from an aqueous potassium–sodium-containing carbonate-rich solution. Its synthesis and detailed characterization can be found in Vitova et al.[30] and was used as a reference for the Np experiments. Np$^V$ consists of K$_3$[Np$^V$O$_2$(CO$_3$)$_2$]$_{(cr)}$ and K[Np$^V$O$_2$CO$_3$]$_{(cr)}$. U was characterized as K$_3$Na[U$^{VI}$O$_2$(CO$_3$)$_3$](cr).

**[U$^{VI}$O$_2$]$^{2+}$ and [U$^V$O$_2$]$^+$-K/Fe.** The {[U$^V$O$_2$(Mesaldien)]K}$_n$ compound ([U$^V$O$_2$]$^+$-K) was synthesized by adding [(U$^V$O$_2$Py$_5$)(KI$_2$Py$_2$)] to one equivalent of MesaldienK$_2$ in pyridine. The [U$^{VI}$O$_2$(Mesaldien)] (= [U$^{VI}$O$_2$]$^{2+}$) complex is formed by disproportionation of [U$^V$O$_2$]$^+$-K with [UI$_4$(Et$_2$O)$_2$] in the presence of (Mesaldien)K$_2$ as described by Mougel et al.[19] The [Fe(TPA)(Py)U$^V$O$_2$(Mesaldien)]I (= [U$^V$O$_2$]$^+$-Fe) sample was prepared by reacting [U$^V$O$_2$]$^+$-K with [Fe(TPA)I$_2$] in pyridine as described in Vitova et al.[19] Structural information can be found in Supplementary Fig. 12 and HR-XANES in Supplementary Fig. 11.

**[U$^{VI}$O$_2$]$^{2+}$-M (M = Pb, Ag or Cd).** The [UO$_2$Pb(C$_{15}$H$_{11}$N$_3$)(C$_9$H$_6$O$_6$)(NO$_3$)] ([U$^{VI}$O$_2$]$^{2+}$-Pb (close Pb-O$_{ax}$)), [UO$_2$Pb(C$_{15}$H$_{11}$N$_3$)(C$_{10}$H$_{10}$O$_4$)$_4$] ([U$^{VI}$O$_2$]$^{2+}$-Pb (no Pb-O$_{ax}$)), [(UO$_2$)$_{1.5}$Ag$_2$(C$_8$H$_3$N$_2$)NO$_3$][UO$_2$(C$_7$H$_3$Cl$_2$O$_2$)$_3$] ([U$^{VI}$O$_2$]$^{2+}$-Ag (close Ag-O$_{ax}$)), [Ag$_2$(C$_8$H$_3$N$_2$)][UO$_2$(C$_7$H$_5$ClO$_2$)$_3$] ([U$^{VI}$O$_2$]$^{2+}$-Ag (long Ag-O$_{ax}$)), and [UO$_2$Cd$_{0.5}$(C$_{10}$H$_8$N$_2$)(C$_7$H$_2$NO$_5$)] ([U$^{VI}$O$_2$]$^{2+}$-Cd (no Cd-O$_{ax}$ contact)) solid samples were synthesized via solvothermal and hydrothermal methods and characterized by single crystal XRD (cf. CCDC no 2064214, 2064218, 2160089, 2160096, and 2310856. Structural details are in Supplementary note 15, Supplementary Fig. 26[41–43]. HR-XANES of close ([U$^{VI}$O$_2$]$^{2+}$-Pb (close Pb-O$_{ax}$) ([U$^{VI}$O$_2$]$^{2+}$-Cd (no Cd-O$_{ax}$ contact) can be found in Supplementary Fig. 10.

**[Cp$_3$″U$^{II}$]$^-$, [Cp$_3$″U$^{III}$], and [Cp$_3$″U$^{IV}$]$^+$.** KCp″ [Cp″ = C$_5$H$_3$(SiMe$_3$)$_2$] was synthesized analogously to KCp′ in Peterson et al.[81] The Cp″$_3$U$^{III}$ = [Cp$_3$″U$^{III}$] was synthesized according to Windorff and Evans (2014) except UI$_3$(DIOX)$_{1.5}$ was used in place of UI$_3$ for this sample[82]. Only HR-XANES was measured for this sample no CC-RIXS maps. The [K(crypt)][Cp″$_3$U$^{II}$] = [Cp$_3$″U$^{II}$]$^-$ sample was synthesized as described in Windorff et al.[55] by stirring diluted [Cp″$_3$U$^{III}$] and 2,2,2-Kryptofix in THF under a dinitrogen atmosphere at ambient temperature and adding KC$_8$. The resulting product was extracted into Et$_2$O and washed with pentane. The Cp$_3$″U$^{IV}$Cl = [Cp$_3$″U$^{IV}$]$^+$ compound was synthesized in a manner similar to Blake et al.[45] by adding three equivalents of KCp″ to UCl$_4$ in diethyl ether, removing the solvent and extracting the residue into pentane. The extract was concentrated and cooled down yielding light orange-brown crystals. The crystalline samples of U(II), U(III) and U(IV) were drop casted onto a cryostat sample holder with dry Et$_2$O.

Following the shipment a 1H NMR spectrum in dry $d_8$-THF was recorded to check for retention of the +2 oxidation state in [Cp$_3$″U$^{II}$]$^-$. The spectrum in Supplementary Fig. 14d shows excellent agreement with a pristine sample with nearly identical shifts. Several resonances in the +8 to 0 ppm region are present that are consistent with diamagnetic organic compounds such as solvent (C$_4$H$_8$O), chelate (2,2,2-kryptofix), and those associated with ligand (HCp″ or KCp″), or ligand-like decomposition products. There are some small unidentified resonances$^{(*)}$ at ca. δ −3 and −6 ppm, these are relatively minor (< 5%) and could be due to lack of rigorously deoxygenated solvent, or other brief exposure to air/water. The spectrum suggests the sample displays high oxidation state purity as [Cp″$_3$U$^{II}$]$^{1-}$ (δ 20.5 Cp″-$H$, −5.75 Cp″$H$, −9.5 Cp″-$TMS$ ppm in C$_4$D$_8$O) exhibit line broadening consistent with fast exchange[55]. A longer study of the sample suggested the resonances shift and broaden into the baseline, consistent with the original study[55]. The structural information and the NMR data can be found in Supplementary Fig. 14. HR-XANES of these compounds can be found in Supplementary Fig. 13.

**Additional Np and Pu samples.** All Np and Pu experiments were conducted under controlled Ar atmosphere in a glove box. $^{237}$Np and

$^{242}$Pu concentrations were routinely determined by liquid scintillation counting (Tri-Carb 3110 TR, Perkin Elmer), gamma-spectroscopy (Ge detector AL-30, Canberra), and alpha-spectroscopy (Alpha Analyst spectrometer, Canberra). PTFE vials were cleaned before use according to ASTM C1285-02[83]. UV-Vis/NIR absorption spectra of liquid Np samples were recorded on an Agilent Cary 6000i spectrophotometer (double beam mode, wavelength range 200–1400 nm, data interval 0.2 nm, integration time 0.1 s per step, quartz cuvettes with 2 mm pathlength from Hellma). Vis/NIR absorption spectra of liquid Pu samples were recorded on a diode array spectrometer (MCS 501 system equipped with a halogen lamp source system CLH 500 from Zeiss, single beam mode, wavelength range 350–1020 nm, integration time 12 ms, data interval ≈0.8 nm, PMMA cuvettes with 10 mm pathlength from Brand). All spectra were recorded inside of argon glove boxes by connecting the inside cuvette holder to the outside spectrometer via fiber optics. For the high resolution X-ray spectroscopic experiments 200 µl of the solutions were filled in a liquid cell and sealed by 8 µm Kapton foil. An inert gas cell with 8 µm Kapton foil was used as a second containment. HR-XANES spectra of all additional Np compounds can be found in Supplementary Fig. 16.

**[Np$^{VI}$O$_2$]$^{2+}$ (aq).** The Np(VI) aqueous sample ([Np$^{VI}$O$_2$]$^{2+}$) was prepared by diluting 35 µL of an ion-exchange column purified and oxidation state pure 0.118 M Np(VI) stock solution with 165 µL of 1 M HCl to yield a final concentration of 0.02 M, as confirmed by Vis/NIR. The purity of [Np$^{VI}$O$_2$]$^+$ is given as 3.72% Np(V) and 96.28% Np(VI), and was determined by UV-Vis before the X-ray experiments. Another UV-Vis measurement showed that even 6 month after the X-ray experiments mostly [Np$^{VI}$O$_2$]$^+$ was present in the solution while the Np(V) content slightly increased. However, due to the slow kinetics of the conversion and the late UV-Vis measurements the it can be deduced that during the X-ray experiment, the Np(V) impurity was in the low single-digit percent. (Supplementary Fig. 16a).

**[Np$^V$O$_2$]$^+$ (aq).** 540 µL of the 0.118 M Np(VI) stock solution in 1 M HCl and 5.76 mL of 1 M HCl were added to a 15 mL screw-capped PP vessel (Sarstedt, Germany). Instantaneous reduction to Np(V) was achieved by the addition of 120 mg of NaNO$_2$ powder. After 1 h, the addition of 3 ml of 2 M NaOH set a final pH$_{exp}$ of 12.5, resulting in the precipitation of greenish NpO$_2$OH(am). The latter was separated from the supernatant solution, washed three times with 0.01 M NaOH, and dissolved in 2 mL of 1 M HCl. The precipitation-washing procedure was repeated once more. A fraction of the resulting NpO$_2$OH(am) corresponding to about 1.2 mg of Np was digested in 1 M HCl to yield the final sample concentration 0.025 M, as confirmed by Vis/NIR. The purity of [Np$^V$O$_2$]$^+$ was determined by UV-Vis before and after the X-ray experiments to be 100% Np(V) (Supplementary Fig. 16c).

**Na$_2$Np$_2$O$_7$.** Sodium-dineptunate, Na$_2$Np$_2$O$_7$(cr) was synthesized by the following procedure: Powder X-ray diffraction (XRD) was measured on a D8 Advance from Bruker. Combined scanning electron microscopy and energy-dispersive X-Ray spectroscopy (SEM-EDX) were performed on a FEI Quanta 650 FEG ESEM equipped with a UltraDry™ Peltier cooled silicon drift X-ray detector and Pathfinder software (version 2.8). 90 µL of the 0.20 M ion-exchange column purified and oxidation state pure $^{237}$Np(VI) stock solution in 2% HNO$_3$ (4.3 mg of Np), 810 µL of H$_2$O and 4.1 mL of a solution containing 2.0 M CsCl and 0.20 M NaOH were added to a 10 mL screw-capped PTFE vessel (Vitlab), and equilibrated at T = 80 ± 5 °C for 29 days, resulting in the formation of black crystalline solid phase. Sufficient fractions of the latter (about 0.2–0.5 mg of Np) were separated from the main vessel, washed 1 x with 1 mL of H$_2$O and 2 x with 1 mL of absolute ethanol to remove residues from the matrix solution, and placed as wet paste on the corresponding sample holders for SEM-EDX and powder XRD, where the material dried within few minutes. An unwashed fraction of the

material (about 50 µg of Np) was digested in 820 µL of 1 M HCl, and directly analyzed for the Np oxidation state distribution by the solvent extraction procedure using Di-(2-ethylhexyl) phosphoric acid (HDEHP), as detailed in reference[84].

The powder XRD pattern of the synthesized material matches well with the patterns previously reported for sodium-dineptunate, $Na_2Np_2O_7(cr)$[44,85]. XRD and SEM-EDX results confirm a high homogeneity of synthesized material with other solid phases being absent, see Supplementary Figs. 17 and 18. In particular, no cesium-containing compound was detected despite the fact that an excess of cesium was present during the synthesis conditions. The obtained Np(VI) solid phase has a platelet-like morphology as was reported for $Na_2Np_2O_7(cr)$[85], and structural Na, Np and O contents with a ratio 0.96: 1: 3.6, *i.e.*, very close to the expected stoichiometric value in $Na_2Np_2O_7(cr)$. Oxidation state analysis by liquid extraction confirmed that the Np(VI) content is about 97%.

**$[Pu^{VI}O_2]^{2+}$ (aq).** Electrochemically prepared Pu(VI) stock solution (100 mM Pu(VI) in 2 M $HClO_4$) was diluted to form a 50 mM Pu(VI), 1 M $HClO_4$ solution under an Ar atmosphere which was analyzed using Pu $L_3$ edge XANES and Pu $M_4$ edge HR-XANES and RIXS. Pu oxidation state was verified using Pu $L_3$ edge XANES. The sample was diluted to reach a final concentration of 10 mM $Pu^{VI}(aq) = [Pu^{VI}O_2]^{2+}(aq)$ in 1 M $HClO_4$. Ge (422) crystals were used in the Lemonnier-type double-crystal monochromator (DCM). The monochromatic radiation was focused by a Rh-coated toroidal mirror, resulting in a spot size of < 1 mm² at the sample position. The Pu $L_3$ edge spectra were calibrated using an Zr foil, aligning the first inflection point of the Zr K edge to 17,998 eV. Data were recorded at room temperature in fluorescence mode using a 4-element Vortex SDD and a single-element Vortex-60EX SDD (Hitachi USA)). The edge position of the Pu(VI) solution was 18,064.8, acquired from the maximum of the first derivative, which is in good agreement with previously measured Pu(VI)-$HClO_4$ solution, confirming the oxidation state of Pu as +VI (cf. Supplementary Fig. 19 also for HR-XANES)[86].

**$Pu^{IV}$(aq).** A 0.053 M $^{242}Pu$ stock solution in 2 M $HClO_4$ was first electrochemically reduced to Pu(III). After electrochemical oxidation to $Pu^{IV}(aq)$ the purity of the sample was confirmed by UV-Vis spectroscopy. The sample was diluted with 2 M $HClO_4$ to reach a final concentration of 0.025 M $Pu^{IV}(aq)$ in 2 M $HClO_4$. UV-Vis and HR-XANES are depicted in Supplementary Fig. 20.

**$Pu^{III}$(aq).** An ion exchange column purified and oxidation state pure 0.041 M $^{242}Pu(III)$ stock solution in 0.7 M $HClO_4$ (99.4 wt. % $^{242}Pu$, 0.58 wt. % $^{239}Pu$, remaining traces of $^{238}Pu$ and $^{241}Pu$, and $^{241}Am$) was used as primary Pu source. The following procedure was performed to enhance the Pu(III) concentration of the initial Pu(III) stock solution for the HR-XANES experiment 1.4 mL of the 0.041 M Pu(III) stock solution in 0.7 M $HClO_4$ was titrated with 1.06 mL of 1 M NaOH to $pH_{exp} = 2.7$. Addition of 8 mL of a solution containing 0.1 M TRIZMA® HCl, 0.4 M TRIZMA® Base and 0.01 M $Na_2S_2O_4$ resulted in the precipitation of blueish $Pu(OH)_3(am)$, with the final $pH_{exp}$ being 9.7. The dark blue $Pu(OH)_3(am)$ was separated from the supernatant solution by centrifugation, and washed one time with 1 mL of $H_2O$. 159 µL of 1 M $HClO_4$ were added to the wet $Pu(OH)_3(am)$ leading to its quantitative dissolution as deep blue $Pu^{3+}(aq)$, and a final sample volume of 205 µL (the difference of about 46 µL is due to residual solution released from the wet $Pu(OH)_3(am)$ upon dissolution). Vis/NIR spectroscopy of a 1: 100 dilution confirmed Pu(III) as the only Pu oxidation state (> 99%), and a concentration of 0.084 M $Pu^{III}(aq)$. A second Vis/NIR analysis performed after the experiments at the beamline demonstrated practically identical Pu oxidation state distribution (Pu(III) > 99%), and a slightly greater concentration of $[Pu^{3+}] = 109$ mM. The latter is likely due to partial evaporation of solvent over the course of the experiments (cf. Supplementary Fig. 21 for HR-XANES and UV-Vis).

**$Am^{III}VO_3$, $Am^{IV}O_2$.** The synthesis and characterization of the Americium dioxide ($Am^{IV}O_2$) and Am(III)-vanadate ($Am^{III}V^{III}O_3$) solids were reported Vigier et al.[87,88]. Briefly: $Am^{IV}O_2$ was synthesized via thermal decomposition of the corresponding oxalate. $Am^{III}VO_3$ was prepared by thermal treatment of a stochiometric mixture of $Am^{IV}O_2$ and $V_2O_5$ pressed into a pellet under $Ar/H_2$ atmosphere. The Am $M_4$ and Am $M_5$ edge HR-XANES of $Am^{IV}O_2$ and $Am^{III}VO_3$ pellets can be found in Supplementary Fig. 22.

## X-ray methods

**Experiments in 90° π geometry.** The X-ray experiments have been performed at the ACT station for actinide research at the CAT-ACT beamline and the SUL-X beamline at the KIT Light Source (KLS) in Karlsruhe, Germany[37,38]. For the CAT-ACT beamline the beam was collimated by a cylindrically bent Si mirror, monochromatized by a Si(111) double crystal monochromator, and focused by a toroidal Si mirror. Then, if "slits" is noted in column "exp. conditions" in Table 1, a slit was used to confine the beam size to 500 µm × 500 µm onto the sample. A Johann-type X-ray emission spectrometer with a Rowland circle of 1 m was used to detect the emitted fluorescence by the samples[37,89]. The complete spectrometer was enclosed in a He-flushed glovebox to minimize scattering of the tender X-rays[31]. Excitation and emission energies, used crystals, Bragg angles, experimental and calculated energy resolutions as well as core hole lifetime broadenings can be found in Supplementary Table 6 and 7. The experimental conditions and energy ranges for all RIXS maps, HR-XANES or normal emission spectra measured at the CAT-ACT beamline are summarized in Table 1.

**Experiments in 172° σ geometry.** These were performed of the samples $[U^{VI}O_2]^{2+}$-M (M = Pb, Ag or Cd) at the SUL-X Beamline at the at the KIT Light Source, Karlsruhe, Germany[38,39]. The synchrotron radiation generated by a wiggler was passed through the Si(111) double crystal monochromator and collimated using mirrors. Excitation energies were calibrated to the experiments were drift-corrected using the In $L_3$ edge then aligned to the experiments performed at the ACT beamline. The beam was focused to about 100 ×100 µm² on the sample using an intermediate focusing device and KB mirrors. Measurements were performed using a single analyzer crystal X-ray emission spectrometer in focusing horizontal Rowland circle geometry. The crystal, sample and detector are positioned in a horizontal direction in a geometry where 180° are between the incoming beam and the crystal (see Fig. 2b in the main text). In the vertical direction the Roland circle is 8° inclined against the beam direction, allowing the incoming beam to bypass below the crystal to reach the sample position, this results in an overall 172° σ geometry. The spectrometer was set up in a vacuum chamber using the existing Θ−2Θ diffractometer. The horizontal diffractometer consisted of the spectrometer crystal with a 0.5 m bending radius on an xyz-stage on the Θ axis, and an avalanche-photodiode detector on a radially moveable linear stage on the 2Θ arm. The spectrometer covered about 58° to 88° 2Θ Bragg angles in back-scattering geometry. In this work, the spectrometer was aligned for U $M_β$ (U $M_4N_6$ 3337 eV, 75.36° Bragg angle) emission line using Si(220) reflex of a Si(110) spherically bent, horizontally striped analyzer crystal with 0.5 m bending radius (ESRF) for HR-XANES and CC-RIXS experiments. Energy resolution of the spectrometer was determined to be about 2 eV FWHM of elastic scattering measured at 3800 eV excitation energy. A summary of the experimental parameters can be found in Supplementary Table 7.

## Data treatment, calibration, and normalization

**Calibration of spectra.** The spectra of the $U^{IV}O_2$ pellet, the $Np^V$ reference compound described in Vitova et al.[30] and the $Pu^{IV}O_2$ sample were used to calibrate the U, Np and Pu data, respectively. For this calibration, first an emission line was recorded at an excitation energy far

**Table 1 | Experimental conditions and scan parameters for the RIXS maps, HR-XANES and line scans of the studied compounds in 90°π geometry. If not noted otherwise, same conditions were used for RIXS and HR-XANES scans. Excitation and emission energies are denoted as $E_{INC}$ and $E_{EM}$, respectively, please note: The excitation and emission energies are for not calibrated RIXS spectra**

| Sample | $M_4$ edge CC-RIXS map | | | exp. conditions | $M_4$ edge CC-RIXS map used to derive relative shifts | |
|---|---|---|---|---|---|---|
| | $E_{INC}$ (eV) | $E_{EM}$ (eV) | $t_{INT}$ (s) | | $E_{INC}$ (eV) | $E_{EM}$ (eV) |
| $U^{IV}O_2$ | 3719.7 – 3742.7 step size 0.14 | 3322.5 3345.5 step size 0.5 | 2 | RIXS: 4 crystals, no slit HR-XANES:4 crystals, slits | All data was derived from the one large $M_4$ edge CC-RIXS map = Large standard map | |
| $Np^{IV}O_2$ | 3843.11 – 3866.08 step size 0.14 | 3423.01 – 3447.07 step size 0.33 | 2.7 | Crystal 3, no slit | Large standard map | |
| $[Np^{V}O_2]^+$ (aq) | 3840 – 3855.27 step size 0.15 | 3434.31 – 3449.3 step size 0.5 | 4.2 | Crystal 3, slits | Large standard map | |
| $[Np^{VI}O_2]^{2+}$ (aq) | 3840 – 3855.27 step size 0.15 | 3434.31 – 3449.3 step size 0.5 | 4.2 | Crystal 3, slits | Large standard map | |
| $Na_2Np^{VI}_2O_7$ | 3844 – 3855.27 step size 0.11 | 3434.31 3449.3 step size 0.5 | 2 | Crystal 3, slits | Large standard map | |
| $Pu^{IV}O_2$ | 3964.4 – 3987.4 step size 0.14 | 3520.3 – 3987.3 step size 0.5 | 2 | RIXS: 4 crystals, no slit, HR-XANES: crystal 3, slits, mask | Large standard map | |
| $Pu^{III}$(aq) | 3959 – 3977 Step size 0.1 | 3526 – 3546 Step size 0.33 | 1 | RIXS: Crystal 5 in position of crystal 3, HR-XANES: addionally slits | Large standard map | |
| $Pu^{III}$(aq) small map | 3965 – 3977 Step size 0.1 | 3535 – 3549 Step size 0.33 | 2 | Crystal 5 in position of crystal 3 | Large standard map | |
| $[Pu^{VI}O_2]^{2+}$ (aq) | 3968 – 3980 step size 0.1 | 3533 – 3540 step size 0.33 | 2 | RIXS: crystal 3, slits, HR-XANES: crystal 3, slits, mask | Large standard map | |
| $Pu^{IV}$(aq) | 3965 – 3988 step size 0.14 | 3523 – 3547 step size 0.5 | | 3 crystals, no slit | Large standard map | |
| $[U^{VI}O_2]^{2+}$-Pb close Pb-$O_{ax}$ | 3722.21 – 3737.1 Step size 0.4 | 3335.26 – 3352.6 Step size 0.5 | 9 | Crystal 5 in position of crystal 3 | Large standard map | |
| $[U^{VI}O_2]^{2+}$-Cd | 3722.21 – 3737.1 Step size 0.4 | 3335.26 – 3352.6 Step size 0.5 | 9 | Crystal 5 in position of crystal 3 | Large standard map | |
| $[U^{V}O_2]^+$-K | 3723.21 – 3743.71 Step size 0.15 | 3335.26 – 3346.26 step size 0.5 | 2.7 | RIXS: Crystal 3, slits, HR-XANES: 4 crystals, slits | Large standard map | |
| $[U^{V}O_2]^+$-Fe | 3723.21 – 3743.71 Step size 0.15 | 3335.26 – 3346.26 step size 0.5 | 2.7 | RIXS: Crystal 3, slits, HR-XANES: 4 crystals, slits | Large standard map | |
| $[U^{VI}O_2]^{2+}$ | 3723.21 – 3737.1 Step size 0.15 | 3335.26 – 3346.26 Step size 0.5 | 3.6 | RIXS: Crystal 3, slits, HR-XANES: 4 crystals, slits | 3723.66 – 3732.18 step size 0.1 | 3333 – 3340.7 step size 0.45 |
| $U^{VI}O_2(CO_3)_3^{4-}$ | | | | Crystal 1, slits | * | |
| $[Cp''_3U^{II}]^-$ | 3722.2 – 3734 Step size 0.15 | 3335.76 – 3348.12 Step size 0.5 | 2.5 | Crystal 3, slits | Large standard map | |
| $[Cp''_3U^{IV}]^+$ | 3723.21 – 3737.71 Step size 0.15 | 3347.26 – 3335.26 Step size 0.5 | 2.75 | Crystal 3, slits | Large standard map | |
| $Am^{III}V^{III}O_3$ | 4092 – 4103.9 step size 0.3 | 3633.52 – 3649.52 Step size 0.5 | 12 | Crystal 3, slits | Large standard map | |
| $Am^{IV}O_2$ | 4092 – 4103.9 step size 0.3 | 3633.52 – 3649.52 Step size 0.5 | 12 | Crystal 3, slits | Large standard map | |

The integration time in seconds per energy step is denoted as $t_{INT}$. The size of the slits, when used, is 500 µm × 500 µm. The diameter of all crystals used is 100 mm, if mask is denoted the diffracting diameter of the crystal was decreased to 50 mm. All data was derived from the one large $M_4$ edge CC-RIXS map except if denoted in last column.
*Normalization for this sample was performed by scaling the WL of the spectrum to that of $[U^{VI}O_2(Mesaldien)]$.

above the absorption edge for the respective U, Np or Pu reference sample (normal emission). The energy positions of the maximum intensity of these normal emission lines were set to the above-described literature values for the characteristic $M_\beta$ fluorescence lines of the respective elements ($E_{EM}$(U) = 3337 eV, $E_{EM}$(Np) = 3435 eV, $E_{EM}$(Pu) = 3534 eV, $E_{EM}$(Am) = 3635 eV). The obtained energy shifts were used to calibrate the emission energy scale of the RIXS maps. Then the analyzer crystal was fixed at this emission energy and an excitation energy scan (HR-XANES) was recorded. The energy positions of the most intense absorption resonances (white line, WL) were

set to the literature values for the $M_4$ edge of $E_{INC}$(U) = 3725.2 eV, $E_{INC}$(Np) = 3849 eV, $E_{INC}$(Pu) = 3970 eV and $E_{INC}$(Am) = 4096 eV). The obtained energy shifts were used to calibrate the excitation energy scale of the RIXS maps or the HR-XANES.

The spectra of the $Am^{III}VO_3$ sample were used to calibrate the Am data. For this calibration, first an emission line was recorded at an excitation energy far above the absorption edge of $Am^{III}VO_3$ sample (normal emission). The energy positions of the maximum intensity of these normal emission lines were set to the literature value for the characteristic fluorescence lines of the element ($E_{EM}$(Am) = 3444 eV).

Then the analyzer crystal was fixed at this emission energy and an excitation energy scan (HR-XANES) was recorded. The energy position of the most intense absorption resonances (white line, WL) were set to the literature values for the $M_5$ edge of $E_{INC}(Am) = 3890$ eV). The obtained energy shifts were used to calibrate the excitation energy scale of the HR-XANES.

**Relative energy scale of spectra.** To achieve comparability between the RIXS spectra of different elements, relative energy scales were obtained by setting the WL energy positions to 0. In order to achieve this, first the WL energy was determined for each RIXS map individually. The energy position of the maximum intensity of the WL was determined by comparing the maxima of many parallel cuts through the RIXS maps with small emission energy variations around the absorption maximum of the RIXS map. For the $An^{IV}O_2$ samples, this was done using large RIXS maps that include the satellite feature and the WL. For the $U^{VI}$ compound smaller RIXS maps were recorded using the same experimental conditions, i.e., one for the satellite feature and one for the WL (cf. Table 1 and Supplementary Table 7). The relative energy scales were used in Figs. 3 and 6.

The RIXS cross sections presented in Fig. 3g were obtained at the energy positions of the maximum intensities of the satellite peaks. For this, the positions of the satellite peaks were determined individually from the corresponding RIXS maps (cf. Supplementary Table 1).

For the relative energy scale used in Fig. 5 the elemental energy scales were linked via anchored compounds. $[PuO_2]^{2+}$ (aq) ($5f^2$) was matched to $UO_2$ ($5f^2$) and this was marked with the anchor symbol in the figure. $Na_2Np^{VI}_2O_7$ ($5f^1$) was shifted by the average of the shift deduced from $[U^VO_2]^+$-K and $[U^VO_2]^+$-Fe ($5f^1$) denoted by the second anchor, and the $Am^{IV}O_2$ ($5f^5$) was shifted by the $Pu^{III}$ (aq) value ($5f^5$) denoted by the third anchor. The other Np/Pu/Am compounds were shifted relatively to the anchored values of $Na_2Np^{VI}_2O_7$ ($5f^1$), $[PuO_2]^{2+}$ (aq) ($5f^2$) and $Am^{IV}O_2$ ($5f^5$), respectively.

**Normalization of spectra.** Another important step in the data analysis is the normalization of the spectra. Special care was taken to be able to compare spectra of different samples properly. For most samples, large actinide $M_4$ edge CC-RIXS maps were recorded so that all cross sections of the satellite feature originate from the same map as the cross-section of the main resonant peak (WL). Within this map an energy transfer cut (blue line Fig. 1) was taken through the determined energy position of the WL. It was modeled with a Lorentzian function. The whole map was normalized to the maximum of this Lorentzian function. Then the satellite peak cross-section evaluations were performed.

The only two exceptions were: 1st; $[U^{VI}O_2]^{2+}$ where the main and the satellite peak were on different maps and 2nd; $U^{VI}O_2(CO_3)_3^{4-}$ (only used in Supplementary Fig. 4, synthesis and characterization in Supplementary Note 9), for which the white line maxima was then scaled to the normalized $[U^{VI}O_2]^{2+}$ maxima. The simulated spectra contained no noise and therefore were directly normalized to the most intense WL point. Therefore, experimental with theoretical trends as well as relative intensities are comparable.

For the analyses of the satellite peak intensity at 172° σ geometry (Figs. 7 and 8, Supplementary Figs. 25 and 28), the RIXS maps were normalized to the maximum intensity in the post-edge region. Note that this was necessary to obtain the correct trend of the satellite peak intensity. The rate of change of satellite peak intensity and WL intensity can be different when the bond covalency changes and this can lead to misleading results. Therefore, for bond covalency analyses, we consider normalization to the post-edge for a more reliable approach.

**Estimation of uncertainties using Monte Carlo (MC) methods.** The WL and satellite peak positions in the RIXS maps were determined as the maxima of the satellite or most first intense feature called WL

respectively. The WL shifts as they are depicted in Fig. 5 on the x-axis were determined as the difference between two WL maxima. The maps and relative positions can be seen in Supplementary Fig. 6. With that, the error in energy loss ($\sigma_{loss}$) was estimated. Thereafter a Monte Carlo like approach was performed. An energy loss value $x_{loss}$ was randomly taken from a Gaussian distribution centered at $\mu_{loss}$ and with standard deviation $\sigma_{loss}$:

$$p(x) = \frac{1}{\sigma\sqrt{2\pi}} e^{\frac{-(x-\mu)^2}{2\sigma^2}} \tag{2}$$

The diagonal cuts were collected at the drawn $x_{loss}$. The same procedure was applied to the WL and satellite peak and then the relation between the fitted area was computed as presented in the main text and exemplary shown in Supplementary Fig. 7 This procedure was repeated for every sample $N = 1000$ times to get a statistical distribution. Then the mean and the standard deviation of the distribution were taken as expectation value and error, respectively in Figs. 4, 5 and 8.

## Calculations
### Active space Configurations Hamiltonian

$$
\begin{array}{llll}
\text{AT} & \{An_{3d}, An_{4f}, An_{5f}\} & 5f^n & \begin{aligned} H_{AT} &= U_{5f5f} + U_{3d5f} + U_{4f5f} \\ &+ H_{SOC,5f} + H_{SOC,3d} + H_{SOC,4f} \end{aligned}
\end{array} \tag{3}
$$

$$
\text{CFT} \quad \{An_{3d}, An_{4f}, An_{5f}\} \quad 5f^n \quad H_{CFT} = H_{AT} + H_{DFT,5f} \tag{4}
$$

$$
\text{LFT} \quad \{An_{3d}, An_{4f}, L_{5f}\} \quad 5f^n L^{14}, 5f^{n+1}L^{13} \quad H_{LFT} = H_{AT} + H_{DFT,AnO_8} \tag{5}
$$

The theoretical part of this work is based on multiplet calculations by means of the full multiplet Quanty code[90,91]. Since X-Ray excitations for correlated systems are generally localized, an impurity approach is the standard way to model Actinide core X-ray spectra[11,28,32]. In this work, atomic theory (AT) and crystal field theory (CFT), where only $5f^n$ configurations are considered, were applied to An $M_4$ edge CC-RIXS spectra of $An^{IV}O_2$ systems (U, Np, Pu). Calculations on ligand field theory (LFT) levels were also computed, for which configuration interaction with the shell occupation $5f^{n+1} L_f^{13}$ was allowed. The ligand orbitals $L_f$ are linear combination of the $O_{2p}$ like orbitals that interact with the $An_{5f}$ ones and exactly reproduce the DFT eigenstates (cf. Supplementary Figs. 30−33. Equations 3 to 5 summarize the considered ground state configurations and Hamiltonians for the different level of theories. The operators appearing in the Hamiltonian are introduced below in the text. Italic font indicates quantum mechanical operators.

Depending on the mentioned level of theory, different contributions to the total Hamilton operator $H$ were considered. For a configuration with one open core shell $c$, the total coulomb interactions $U$ sum up to

$$U = F_{5f5f} + F_{c-5f} + G_{c-5f} \tag{6}$$

Here, $F_{5f5f}$ is the Coulomb interaction among 5f electrons and $F_{c-5f}$ ($G_{c-5f}$) the direct (exchange) interaction between the 5f and core electrons under conservation of shell occupation. For the initial state configuration, only $F_{5f5f}$ is needed. Together with the SOC operator $H_{SOC,nl} = \xi_{nl}(\vec{l} \cdot \vec{s})$ for the open shells, Coulomb interaction builds the Hamiltonian for the Atomic Theory. The employed atomic parameters are listed in Supplementary Table 2.

In order to account for the presence of an environment around the actinide atom, an approach to build up an ab initio impurity Model was applied[91,92]. Here, atomic like Wannier functions and the associated

tight binding Hamiltonian for bulk $An^{IV}O_2$ were constructed using a full potential local orbital (FPLO)[93,94] density functional theory (DFT) code[95]. PBE(GGA) potential on a scalar relativistic level was chosen for the DFT computation. Being that the obtained FPLO Wannier functions were very close to the FPLO basis set[91], the parameters for atomic interactions were derived from the FPLO basis functions, which are atomic like. The DFT tight binding Hamiltonian was then exploited to include the presence of the environment on a CFT and LFT level; for the latter, the Hamiltonian for a $AnO_8$ ($H_{DFT,AnO_8}$) cluster was built, including only the An $5f$ and 1st Ligand O $2p$ ($L_f$) states. HDFT, $AnO_8$ results in an operator with nine parameters describing the $5f$ and $L_f$ onsite energy ($\epsilon_{5f,\Gamma}$ and $\epsilon_{L,\Gamma}$) and $5f$-$L_f$ hopping ($V_{\Gamma,g}$) for the three irreducible representations of $l = 3$ ($\Gamma=\{a_{2u},t_{1u},t_{2u}\}$) in $O_h$ point group symmetry:

$$H_{DFT,AnO_8} = \sum_{\Gamma,g} \epsilon_{5f,\Gamma}\, a^\dagger_{5f,\Gamma,g} a_{5f,\Gamma,g} + \epsilon_{L,\Gamma}\, a^\dagger_{L,\Gamma,g} a_{L,\Gamma,g} + V_{\Gamma,g}(a^\dagger_{5f,\Gamma,g} a_{L,\Gamma,g} + a^\dagger_{L,\Gamma,g} a_{5f,\Gamma,g}) \quad (7)$$

Where g indexes the degenerated substates for every irreducible representation. The ab initio Tight Binding parameters for the cluster can be found in Supplementary Table 3.

In the simpler CFT case, the antibonding eigenenergies of $H_{DFT,AnO_8}$ were adopted for the effective $5f$ crystal field splitting,

$$H_{DFT,5f} = \sum_{\Gamma,g} \epsilon^\star_\Gamma a^\dagger_{5f,\Gamma,g} a_{5f,\Gamma,g} \quad (8)$$

The number of free parameters in the system is therefore minimal. In AT or CFT for large systems like bulk $An^{IV}O_2$, the ab initio multipole slater coefficients $R^k$ are commonly scaled[28,96,97] by a factor of 0.8-0.9. This factor might decrease in LFT and models beyond, which take in consideration part of the screening mechanism explicitly. Figure 6 displays the need for $G^0_{4f5f}$ to be taken as a free parameter. For LFT, the additional free parameters are $\Delta$, $\Delta_1 = \Delta\text{-}U_{3d5f} + U_{5f5f}$, $\Delta_2 = \Delta\text{-}U_{4f5f} + U_{5f5f}$[98], giving the energy difference between the $5f^n\,L_f^{14}$ and $5f^{n+1}\,L_f^{13}$ center of gravity for initial, intermediate, and final state, respectively. $U_{l1,l2}$ is the average interaction between two electrons in the shells $l_1,l_2$. Supplementary Table 4 displays the value of the free parameters for the figures displayed in the main manuscript.

In the Quanty code, RIXS spectra are simulated by computing a third order response Green's function $G^3(\omega_1,\omega_2)$ of the form:

$$G^3(\omega_1,\omega_2) = \left\langle \psi_{GS} | T_1^\dagger \frac{1}{\omega_1 - H_1^\dagger - \frac{\Gamma_1}{2}} T_2^\dagger \frac{1}{\omega_2 - H_2 + \frac{\Gamma_2}{2}} T_2 \frac{1}{\omega_1 - H_1 + \frac{\Gamma_1}{2}} T_1 | \psi_{GS} \right\rangle \quad (9)$$

where $T_{1(2)}$ is the excitation (emission) operator, $\omega_{1(2)}$ the photon ingoing (outgoing) energy, $H_{1(2)}$ the Hamilton operator for intermediate (final) state and $|\psi_{GS}\rangle$ is the ground state of the system. This expression is equivalent to the Kramers-Heisenberg equations (KHE), which are generally used to model $M_4$ edge CC-RIXS spectra[63]. In such processes, the total spectrum is given by a linear combination of the fundamental ones with coefficients given by the experimental geometry[99]. For the example of a sample with $O_h$ symmetry, the powder spectra for a RIXS 90° π geometry with electric dipole transitions was calculated. Here, 90° is the angle between the incident beam and the analyzer crystal and π means the incoming linear polarization is parallel to the sample-analyzer crystal direction, as displayed in Fig. 2b. Scalar lifetime broadening effects for intermediate and final state were considered as $\Gamma_1$ and $\Gamma_2$ in the Lorentzian like profile of the KHE. In addition to that, 2-d Gaussian filtering of the resulted spectra with finite $\sigma_{exc}$ and $\sigma_{em}$ simulates the effect of experimental broadening. While the values for the lifetime broadening $\Gamma$ is a characteristic of the element and system analyzed, the experimental broadening has to be estimated under comparison with the measured spectra.

## LFDFT Calculations

LFDFT[100] calculations were performed using the Amsterdam Density Functional (ADF in AMS2023) program packages[101]. Similar to the Quanty approach (see above), LFDFT also provides the computational efficiency for calculating multiplet energies and electronic structures with multi-open-shell electron configuration based on DFT using the average of configuration (AOC) type calculation. It has been shown that LFDFT is well suited to deal with actinide coordination compounds in particular[102,103].

The U $M_4$-edge RIXS maps are calculated based on the $[U^{VI}O_2]^{2+}$-Pb (close Pb-O$_{ax}$), $[U^{VI}O_2]^{2+}$-Cd (no Cd-O$_{ax}$ contact) experimental structures (see Supplementary Note 23. LFDFT structures input file). The RIXS maps are modeled using the polarizability tensor for inelastic scattering (KHE). In the RIXS process, the ground-state (GS), intermediate-state (IS) and final-state (FS) are multi-configuration electronic structure with U $3d^{10}4f^{14}5f^0$, $3d^94f^{14}5f^1$ and $3d^{10}4f^{13}5f^1$, respectively. We start with DFT calculations where we force fractional electron occupation on selective molecular orbitals (MOs) in ADF[101] to mimic the electron configuration systems that are involved in the RIXS process. The DFT calculation are performed with the hybrid PBE0 functional[101], whereas the MOs are expanded in terms of all electron Slater-type Orbital basis sets of triple-zeta plus polarization quality (TZ2P for U and TZP for other elements)[101]. For the GS, there is no electron on the U $5f$ orbitals, giving rise to a closed shell electronic structure. For the IS, we removed 1 electron from the core U $3d$ orbitals and smear it in the formally unoccupied MOs that are identified with large fractional parentage coefficients for U $5f$. This results with an average of configuration type electronic structure, where each of the five U $3d$ and each of the seven U $5f$ MOs receive 9/5 and 1/7 electron occupation, respectively. Similarly for the FS, we removed 1 electron from the semi-core U $4f$ orbitals and smear it in the MOs with large fractional parentage coefficients for U $5f$, resulting with an average of configuration type electronic structure, where each of the seven U $4f$ and U $5f$ MOs receive 13/7 and 1/7 electron occupation, respectively. The configuration average energy parameters for GS, IS and FS are calculated as the difference between DFT total energies ($E^{(DFT)}$): i.e. $\triangle(5f,5f)=0$, $\triangle(3d,5f)=E^{(DFT)}_{IS} - E^{(DFT)}_{GS}$ and $\triangle(4f,5f)=E^{(DFT)}_{FS} - E^{(DFT)}_{GS}$. The calculated excitation and emission energies are further shifted in order to align the theoretical maps with the experiments. For $[U^{VI}O_2]^{2+}$-Pb (close Pb-O$_{ax}$) these energy shift values are 57.47 eV and 64.74 eV for the excitation and emission energies, respectively. For $[U^{VI}O_2]^{2+}$-Cd (no Cd-O$_{ax}$ contact) the energy shits values are 57.95 eV and 65.11 eV, respectively. Note that the average of configuration type DFT calculation implies that the electron density belongs to the totally symmetric representation under which the ligand-field Hamiltonian is invariant. Therefore, based on the previous DFT results, we determine the ligand-field parameters. These parameters include the Slater-Condon integrals $F^k(3d,5f)$, with $k = 2, 4$, $F^k(4f,5f)$, with $k = 2, 4, 6$, $G^k(3d,5f)$, with $k = 1, 3, 5$, and $G^k(4f,5f)$, with $k = 0, 2, 4, 6$; the spin-orbit coupling constants $\xi_{3d}$, $\xi_{4f}$ and $\xi_{5f}$, and the 5×5, 7×7 and 7×7 matrices that represent the $3d$, $4f$, and $5f$ ligand-field potential[104]. Supplementary Table 5 lists all the calculated parameter values for the reproduction of the theoretical RIXS maps. The core-hole lifetime and broadening effects in the IS and FS are introduced by using a Lorentzian function with a constant full-width at half-maximum parameters: 3.50 eV and 0.37 eV.

## Data availability

**Supporting Information Available.** Additional figures and details, as well as used parameters, peak positions, lifetime broadenings, experimental resolutions, LFDFT structure input files. The published

experimental data that support the findings (figures in the main text and supporting information) of this study are available in the KIT-open repository (https://www.bibliothek.kit.edu/english/kitopen.php) with the identifier(s) [https://doi.org/10.35097/5qqht0vfqrktvccj] Source data are provided with this paper.

## Code availability

The code Quanty used for this study is released under the CC By license and can be obtained from www.quanty.org. The input files can be found at https://doi.org/10.35097/5qqht0vfqrktvccj.

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

## Acknowledgements

We thank Sven Schenk for the help with experiments. We thank Bayler Barnes (NMSU) and Jean-François Vigier (JRC-KA) for assistance with compound and figure preparation. We acknowledge Steliyana Lehchanska for the graphical design of the featured image. We thank the Institute for Beam Physics and Technology (IBPT, KIT) for the operation of the storage ring, the Karlsruhe Research Accelerator (KARA). B.S., H.K.-H., C.V., A.B., R. S. K. E., T.N., C.R., H.R., T.V. acknowledge funding from the European Research Council (ERC) Consolidator Grant 2020 under the European Union's Horizon 2020 research and innovation program (grant agreement No. 101003292). M.T. and M.W.H. acknowledge support by the Deutsche Forschungsgemeinschaft (DFG, German Research Foundation) through the research unit QUAST, FOR 5249 (Project P7), Project ID No. 449872909. B.S. and J.A.B. acknowledge the support of DOE, Office of Science, Office of Basic Energy Sciences, Chemical Sciences, Geosciences, and Biosciences Division, Heavy Element Chemistry Program at the Lawrence Berkeley National Laboratory (LBNL) under contract DE-AC02-05CH11231 and C.C. and D.B. at The George Washington University under grant number DE-FG02-05ER15736. C.W. acknowledges support by Department of Energy-Early Career Program under grant number DE-SC0024165.

## Author contributions

B. S. designed the study, performed the experiments, analyzed the data, worked on the figures and wrote the manuscript; M. T. performed the calculations, analyzed the data, worked on the figures and wrote the manuscript; M. M., K. P., O. W., D. F., C. R., D. B., C. C., C. W. performed the synthesis of the investigated compounds and materials; C. V., H. K.-H., performed the synthesis of the investigated materials and some of the experiments; T. P., A. B., T. N., J. G., R. S. K. E., J. A. B., T. S., D. S. performed some of the experiments; H. R. performed some of the experiments and some of the calculations, M. H. designed the study, preformed calculations and analyzed the data; T. V. designed the study, analyzed the data, prepared the figures and wrote the manuscript. All authors have revised and approved the submitted version.

## Funding

## Competing interests

The authors declare no competing interests.
