## [Transparent Peer Review file · Nature Communications]

Resonant inelastic X-ray scattering tools to count 5f electrons of actinides and probe bond covalency

Corresponding Author: Dr Tonya Vitova

Version 0:

Reviewer comments:

Reviewer #1

(Remarks to the Author)

The article presents the application of core-to-core Resonance Inelastic X-ray Scattering to investigate the electronic structure of actinide materials. The importance of this method is well recognized in the field and dedicated instruments for actinide research exist at the ESRF and ANKA synchrotrons. Nevertheless, I think that the present article does not have enough novelty required for publication in Nature Communications. The presented work looks to me as a systematic study that can be published in a more specialized journal. When authors consider revision I would recommend to extend the introduction to give more context to the reader. References to many important articles are given, nevertheless, it is not explained in the text which similar things were already investigated and within the context that can be bridged to the present article. The following points give the examples that lead me to the conclusion about the limited novelty of the present article:

1. Experimental RIXS spectra for some systems were reported previously. For example, 3d 4f RIXS of UO₂ shown in Fig 1 is very similar to the RIXS reported in Fig 1 of the review DOI 10.1039/D1CC04851A, Previous article of authors (DOI 10.1038/ncomms16053) also reports 3d-4f RIXS of NpO₂, UO₂, PuO₂.)

2. The instrument that is used to measure RIXS is also not new (classical Johann-type X-ray emission spectrometer with a Rowland circle of 1 m)

3. The theoretical method is also not new and has been previously applied to actinides in a rather systematic manner. For example Fig 12 from Butorin DOI: 10.1021/acs.inorgchem.0c02032 reports simulations of PuO₂ 3d-4f RIXS which are similar to those reported in Figure 5 of the present manuscript.

The discussion section of the manuscript can be also elaborated including more examples with the comparison of the electronic structure parameters ("number of 5f electrons") obtained using the suggested method and other independent techniques (Neutron scattering, measurements of HERFD XAS at other edges, etc). From my point of view, it is not sufficient to repeat that the satellite peak can be used to determine the electronic configuration and stress the complementary to valence-band RIXS. I think it is necessary to explain why disagreement of results is sometimes obtained by comparing different techniques.

Reviewer #2

(Remarks to the Author)

The authors present x-ray spectroscopy fingerprints of the number of 5f electrons in actinides using the information embedded in the M4 edge core-to-core resonant inelastic x-ray scattering (CC-RIXS) satellite line. In addition to the possibility to use of M4 edge CC-RIXS spectra for 5f electron counting, the authors propose also to use the same spectra to benchmark the correct scaling of the monopole part of the Coulomb interaction, i.e., G₀ and F₀ 4f-5f overlap integrals, which usually need ad-hoc scaling to match various kinds of spectroscopy data of the actinides. They also speculate on the sensitivity of the method to the covalent character of the actinide-ligand bonding.

The experiment and theory are definitely state-of-the-art and the methodology is sound. Performing synchrotron RIXS on actinide compounds alone is already a challenge as such and the necessary infrastructure exists in Karlsruhe. The theoretical simulations are based on the most highly developed tools such as Quanty, which is designed to perform exactly these type of calculations.

In general terms, I fully agree with the authors that “experimental and computational results show that the complex electronic structure and binding properties of the actinide elements can be revealed by advanced spectroscopic tools and computations”.

One of the biggest challenges I find is how to prove an universality of the 5f electron “counting” scheme. The proposal to accept this a universal behavior is now based on the two U(V) samples (U-V and U-V-Fe) showing similar satellite intensity, together with the fact that the An-O₂ oxides following the expected trend from theory until 5f⁵ occupation. From the theory side this is culminated in the lines 179-181: is the atomic multiplet theory justified in systems with strong effects of bond covalency, orbital mixing, or spherical symmetry breaking? The usefulness of the universality would arise if someone wishes to use the concept to analyze an entirely “unknown” sample where the 5f occupation and many other electronic structure parameters may be not known. The proposed trend seems nicely followed by the 6 samples here, but are we certain that any other 5f material in the 5f⁰-5f⁴ series will definitely follow this trend regardless of its bond covalency, orbital mixing, or spherical symmetry breaking? This claim would benefit from more evidence than the 6 samples presented here, out of which just two duplicate. Besides, how about the 5f⁶ case, for which theory suggests that the satellite falls nearly back to the energy and intensity of 5f² (fig. 2c).

Also, about “Application of the satellite peak for probing bond covalency of the actinide-ligand bond”: the authors conclude that the satellite intensity is expected to decrease in the 90deg sigma geometry when covalency increases. However, this note is written without presenting quantitative data or more substantial predictive power. It remains unclear how this could be evaluated experimentally in a quantitative manner.

The satellite sits on a tail of the white line peak and its extraction has uncertainty which probably is reflected in the error bars in Fig. 6b, but the uncertainty has not been discussed. Some more quantitativity would be brought by plotting the known 5f electron number as a function of the satellite peak intensity (or vice versa) after its extraction from the data, with error bars, compared to the theory, and the systematic and statistical error bars should be discussed. This would give hint of the accuracy of the method, which would be needed if universality is true.

The authors conclude on lines 322-324: “For example, there is an intensive discussion of the formal oxidation state of low valent actinide compounds, i.e. AnI, II or III.(Refs.29–33) Here, measuring the satellite peak could be used to unambiguously determine the formal oxidation state of the actinide element.” Did the authors try this already? Some results along these lines would make the present case stronger.

More detailed comments about the results and the discussion:

Is it necessary to measure the full RIXS 2D plane, or is there way to determine how to do only the cut in constant emission energy and/or constant excitation energy scans? This would make data acquisition faster, but the possibility to “miss” the peak without full 2D RIXS measurement may be a big risk.

The authors mention that deviating from the 90deg pi scattering geometry diminishes the correspondence between satellite characteristics and 5f occupation, but does not entirely destroy it. Something toward this is shown in Fig.2e, which notes for 5f⁰ case that there will be non-negligible satellite intensity when going away from 90 degree geometry. I should say that this is maybe not clear enough to illustrate the change in the 5f occupancy trend in slightly-out-of-90deg geometry, maybe better would be a figure like Fig. 2c but for analyzer 1. That kind of figure would carry more information on the sensitivity of the trend to the scattering angle than merely the current Fig. 2c.

In Fig. 2f two different samples are compared with two different scattering angles. This does not make directly sense to me, why the comparison of different scattering geometries is not done for one single sample? It is not certain to the reader that the two samples satellite intensities can be directly compared, there may be differences in density and hence normalization, but more importantly in bond covalency, etc may be different. The reason is probably due to some experimental constraint at the time when the measurements were done, but it decreases the systematic character of the reported study.

It is interesting to see the prediction of a second satellite in the low-energy (in the text L257: “high-energy”?) tail measured at constant excitation energy in Fig. 6a. It seems entirely washed out in the experiment. Would higher experimental energy resolution be able to see it or is it buried in the lifetime limited tail because of solid-state effects or similar?

L267-268: Is the experimental broadening for PuO₂ much different than in the other cases? Why? I did not find information on the experimental energy resolutions for different samples.

The Figure 2a is slightly confusing. The scale of the ordinate axis is 1e1, which I understand as meaning 1.0 on the y-scale equaling 10 eV. If the “energy difference between the spin singlet and triplet states is equal to 14 times $G(k=0)(4f5f)$ ” (line 108), why is it in Fig 2a finite at $G=0$ (about 10 eV) and not linearly increasing? (It would be visually clearer if the 1e1 would not be the overall scale, why not label y-axis ticks as they are in eV)

What was the degree of polarization in the horizontal/ vertical polarization geometry and how the polarization was changed between sigma and pi polarization? There was no mention of a possible optical element or such that switches the polarization. Could a deviation from 100% pure polarization cause some effects on the data? Already a finite size of the analyzer crystal creates deviations from pure pi/sigma geometry, which could only be matched at one line on the analyzer surface.

Lines 153-155: "Several levels of theory have been applied to investigate the origin of the intensity of the satellite peak and its relation to ground-state material properties." What does this mean in practice and do the authors refer to the SI or some other references?

Small typographical issue: Lines 62,63,79, subindex 4 is missing from M4 edges.

In order for the others to reproduce the results, Quantum input files and the processed 2D RIXS maps in numeric format, as electronic supplementary information, would be extremely valuable.

Reviewer #3

(Remarks to the Author)

The manuscript describes the assignment of 5f shell electron occupations via the satellite peak observed in M4-edge RIXS maps. Such an experimental tool is urgently needed, and therefore the manuscript is of potential interest to a broad readership. I am not sure whether the manuscript in its current form makes a convincing case, but it may be that I misunderstood some of the text. A revision is recommended.

The manuscript asserts that ground state 5f occupations can be extracted from the RIXS satellite. A series of atomic calculations (labeled AT, CFT) were performed, and in one case a DFT-calibrated ligand-field theory (LFT) calculation was also carried out. The latter appears to be using DFT orbitals and therefore it is potentially impacted (potentially severely) by the Kohn-Sham delocalization error. It has been shown that this error leads to too much covalency in metal complexes. See, for example, DOI 10.1021/ar500171t for an overview. Related to this, in lines 252-278, it is my impression that the attempt here is to calibrate atomic calculations so they are able to reproduce the experimental spectra. One should then be careful not to over-interpret the adjustments in the atomic data as real molecular effects related to bonding. That would have to be based on a thorough comparison with molecular/ligand field calculations that are not impacted by the delocalization error.

Further comments:

(line numbers as in the PDF generated by the publisher)

Line 44: "experimentalgeometry" is missing a blank space

Lines 322-323: "For example, there is an intensive discussion of the formal oxidation state of low valent actinide compounds, i.e. An I, II or III.[29–33]" The citations should perhaps include also the following article: DOI 10.1021/jacs.1c07519

When the authors write in lines 296-298 "Therefore, it could potentially be used as a tool to probe bond covalency, in particular for the case when the formal 5f occupation is already known." , do they consider some of the examples in the citations listed under the previous point as cases for which the formal occupation is not known? If not, please provide other examples.

Version 1:

Reviewer comments:

Reviewer #1

(Remarks to the Author)

The authors adequately addressed my main concerns. I have noticed only a couple of minor issues during the reading:

1. Line 77: "accessible at several synchrotrons all over the world.18" Please check the reference. While the introduction of 18 also contains some information about other beamlines the reference to the review article 19 looks more appropriate.
2. References 18 and 36 are identical

Reviewer #2

(Remarks to the Author)

As the reviewer #3 comments, experimental tools to accurately and reliably determine the actinide 5f shell electron occupation, are highly desirable, and if robust methods can be found, it would have significant impact. The authors present a systematic study of a satellite line in a core-to-core M edge RIXS map that follows a systematic trend as a function of the 5f electron occupancy. My main concern in the first version of the manuscript was that the case was not very convincingly shown to be treated as valuable novel results. This was the impression of the reviewer #1 and #3 as well.

The authors have made significant improvements, and the second version of the manuscript shows clear advancements to the first version. The authors have collected a new and larger data set that confirms the trends shown in the first manuscript smaller dataset. The integral spectral weight of the satellite line has been measured in a variety of compounds with different 5f shell state. For the 5f electron counting purpose the figure 4b-c is now the important one (experimental and theoretical trend of 5f electron number and the integrated relative satellite spectral weight). The new data set shows variations and overlap of the integral values between neighboring formal valence state compounds. It seems that an uncertainty roughly

equal to one 5f electron remains in the spread of the results. Especially 5f1 and 5f2 can be difficult to be discerned from each other, on the other hand 5f3 and 5f2 have similar weights, then 5f4, 5f5 and 5f6 form a group that puts them together in the same ballpark, especially f4 and f6 configurations probably will never be discerned from each other.

Because of this overlap, I still cannot help wondering if this sets the accuracy of the 5f occupancy determination any higher than HR-XANES, see e.g. the HR-XANES spectra in the supplementary information. It is true that other effects also affect the XANES features and renders them indicative of the shell occupancy, but the overall results in Fig 4b indicate similar level of intrinsic uncertainty of the satellite intensity measurement results.

It is entirely true that the more data we have at hand, the better we can understand the system under study, and the fact that both localized and delocalized electrons can influence the HR-XANES, and that the possibility to measure also the satellite to obtain results for more localized electrons is useful. The entire case seems now to rest in this: whether the results in the improved manuscript and larger data set convincingly demonstrate a novel method that is interest to a broad audience. Unfortunately the current manuscript, even being in a much better state than the first version, still does not convince me very strongly. There is a trend that is a useful guide for the estimation of the orbital occupancy, but it is not highly accurate or monotonic, which would be needed to give a unique answer to the original research question. Seems that the original problem still remains, although the presented spectroscopic tool is certainly a useful one in addition to the old ones.

Concerning the discussion of the covalency: Maybe I still misunderstand something, but the results concerning this seem still confusing (most of it is in the SI probably because of length limits, but I found it cumbersome to repeatedly change between reading the SI and the main text). Since there are various factors at play that can alter the interpretation (probably including the 4f-5f exchange interaction parameter), the argument ends up being rather vague: there is likely to be some variation related to the covalency in the satellite intensity at 180 degree scattering angle, but I miss the quantitative result here. I didn't see a point plotting the satellite intensity in the SI as a function of 5f occupancy in case of all data points having this value equal to zero. Some approach to plot it as a function of degree of covalency would make more sense.

Smaller technical remark: I didn't find an explanation to the change in the manuscript's approach to use 90 degree sigma and pi polarization, switching now to 180 degree pi polarization. Looks like there must have been an error in the original manuscript, with the theory in Fig 2d was earlier labeled to come from 90 degree sigma polarization. The old 3-crystal description is still there in the SI, but the 180 degree geometry seems new. I am not sure how the authors could vary the polarization, but now this issue seems resolved with the 90 degree sigma polarization removed. I am also not sure how it is possible to use 180 degree scattering angle, there must be an error; the scattering angle was probably lower but the authors approximate it to 180 degree in the text. Is there a difference between 180 degree sigma and pi polarization case? I would think they must probably become equivalent at that point.

Reviewer #4

(Remarks to the Author)

This work reports novel spectroscopic techniques that can count the localized 5f electrons and derive information on the degree of 5f electron sharing in actinide complexes. Essentially, these techniques explore changes in a satellite feature, which appears some 6-8 eV apart from the WL in the core-to-core CC-RIXS map, as a function of the localized 5f electron count on the actinide ion (i.e., as a function of the actinide formal oxidation state). The appearance of the satellite feature is based on the strength of the 4f-5f exchange interaction causing a net separation between high- and low-spin states resulting from the spin-exchange coupling of the f electrons. The present manuscript provides a novel, alternative approach to rationalize the oxidation state and chemical bonding in actinide complexes, it is definitely urgent and of tremendous interest for the community of Nature Communications. I recommend publishing it.

To be honest, I find the manuscript truly impressive, reporting in detail data from experiments that are far from trivial to handle. Given the complexity of the electronic structure, the calculations reported here, including LFDFT and atomic multiplet approaches, are at the forefront of what can be achieved. This is because fully ab initio methods are inapplicable (due to complexity) to all the electronic structure problems addressed in this work. As an observation, in response to comment R3, comment 2, I would like to point out that RAS is a post-Hartree-Fock method, not a post-DFT method. One can indeed use DFT orbitals in RAS calculations, but this is not a standard recommendation, nor has the theory been formulated within the realm of Kohn-Sham DFT. Similar to CASSCF, the RASSCF method is variational, addressing both the multideterminantal description of the wavefunctions and the ligand field. That is, the ligand orbitals are not "fixed at the DFT level," nor does the whole approach have anything to do with DFT. RASSCF calculations are the desired approach here to calculate the satellite intensities; however, these cannot be used due to the complexity of the electronic structure.

A comment that I have relates to the emergence of intensity under the satellite peak. If I get it correctly, it becomes apparent from the text and Figure 2a that intensity under the WL is due to the pairing of the 5f and 4f electrons in high- and low-spin states. The intensity under the WL comes from the high-spin states (or states of same spin-multiplicity with the ground state?), while the intensity under the satellite comes from the low-spin states. If this is the case, then spin-orbit coupling should also be an important factor (?) in the generation of intensity under the satellite, because it promotes intensity by mixing the optically spin-allowed states with the spin-forbidden states. Thus, the more localized the electrons are in the 5f shell, the more angular momentum they have, and the more they generate states that are involved in better spin-orbit coupling. Charge-transfer 5f electrons describing covalent bonding have diluted angular momentum and do not lead to configurations

that involve significant spin-orbit coupling, hence no improvement in the satellite intensity. Now it becomes clear that the higher the oxidation state of the actinide, the higher the intensity of the satellite peak will be. This aligns well with the experimental observations. It remains unclear why the intensity trend breaks for 5f6 but I guess this should have something to do with the fact that upon promotion of the 3d electron in the 5f6 shell, the 5f shell as a whole becomes half-filled, 8S term, that is hardly involved in spin-orbit coupling. I am not sure how the CF splitting of the 5f orbitals leads to changes in the position of the satellite wrt to the WL. My feeling is that resting the whole electronic structure problem on the monopole part of the exchange interaction isn't too simple. Whatever the answer is here, I do believe that the calculations presented here bring excellent support for the experimental observations and leaves room for future work.

Minor comments:

1. In the abstract, remove "and electron occupation.", I find it redundant with the "spin configuration" which is itself generated by the electron occupation.
2. Line 47: "large number of 5f valence electrons" -> "large number of electrons". E.g., a U(VI) complex formally has 5f0 valence electrons but still shows a tremendous amount of correlation.
3. Line 324: Reference 47 does not seem to report any theoretical data.
4. Line 371: Remove the first sentence. The need for multireference methods to address the electronic structure is well established. In fact, the 4f135fn configurations explored here cannot be correctly calculated without such methods.

Sergentu Dumitru-Claudiu.

Version 2:

Reviewer comments:

Reviewer #2

(Remarks to the Author)

The authors have made significant improvement and answered my concerns in a comprehensive and adequate manner. The new correlation map in figure 5 is very enlightening. I think the manuscript now is in a state where its impact is higher than what it would have been in the first submitted version, and it is also more convincing.

Answers to Reviewers comments on manuscript

‘Counting the 5f electrons of the actinides’

by Bianca Schacherl, Michelangelo Tagliavini, Hanna Kaufmann-Heimeshoff, Jörg Göttlicher, Marinella Mazzanti, Karin Popa, Olaf Walter, Tim Pruessmann, Christian Vollmer, Aaron Beck, Ruwini S. K. Ekanayake, Jacob A. Branson, Thomas Neill, David Fellhauer, Cedric Reitz, Dieter Schild, Dominique Brager, Christopher Cahill, Cory Windorff, Thomas Sittel, Harry Harry Ramanantoanina, Maurits W. Haverkort, Tonya Vitova.

General:

We would like to thank the reviewers for the constructive comments, which greatly helped to improve the manuscript. You will find our point-by-point response below. The comments of the reviewers are in bold followed by our response.

Reviewer #1

Comment (1): The article presents the application of core-to-core Resonance Inelastic X-ray Scattering to investigate the electronic structure of actinide materials. The importance of this method is well recognized in the field and dedicated instruments for actinide research exist at the ESRF and ANKA synchrotrons. Nevertheless, I think that the present article does not have enough novelty required for publication in Nature Communications. The presented work looks to me as a systematic study that can be published in a more specialized journal.

Answer: Our study presents a spectroscopic tool, which is sensitive to the number of localized electrons on the actinide metal and, when the scattering geometry is changed, it is sensitive to the actinide-ligands bond covalency. It is frontier in actinide research and therefore publication in Nature Communications is justified. For more details see the following answers to specific comments and the substantially updated manuscript.

Comment (2): When authors consider revision I would recommend to extend the introduction to give more context to the reader. References to many important articles are given, nevertheless, it is not explained in the text which similar things were already investigated and within the context that can be bridged to the present article.

Answer: We followed the recommendation of the reviewer and extended the introduction giving broader context to the reader and comparing to other experimental techniques applied for actinides from which information about the electron density on the actinide atoms could be obtained. Unfortunately, it is not possible to give a complete literature review due to the limited space. For this a review article on the topic would be more suitable.

The following text is now added to the introduction: “ The ability to apply experimental techniques sensitive to the localized or delocalized 5f electron densities for all actinides, in both model and applied systems, in order to answer fundamental questions as well as those relevant to societal challenges, is much needed. Various advanced experimental techniques, such as Nuclear Magnetic Resonance (NMR),¹⁴ electron paramagnetic resonance (EPR),^{15,16} An M_{4,5} absorption edge (M_{4,5}) X-ray magnetic circular dichroism (XMCD),¹⁷ high energy resolution X-ray absorption near edge structure (HR-XANES) or valence band resonant inelastic X-ray scattering (VB-RIXS) are sensitive to different levels of localization of the 5f electron density on the actinide atoms and provide complementary information.^{18,19} ”

However, these methods have specific limitations. NMR and An M_{4,5} XMCD are limited to specific actinide elements, An M_{4,5} edge HR-XANES is sensitive to changes of 5f electron density near the metal but not strictly to electrons localized on the actinide atom, leading ambiguous determination of local electron configurations.^{19–21} An M_{4,5} edge VB-RIXS can probe both localized and delocalized electron density, but the probability for the electron transitions is low making it very challenging to obtain high quality data, especially with the high energy resolution needed to probe localized electrons.^{19,22} An O_{4,5} edge non-resonant inelastic X-ray scattering (NIXS) at high momentum transfer is powerful for characterizing ground state electron configurations for actinides. However, it requires sample with high concentration of the element of interest, very high photon flux and multi-analyzer crystal spectrometers, making it very challenging and hardly suitable for general applications.^{15,23} Actinide 3d4f core-to-core resonant inelastic X-ray scattering measured at the M₄ absorption edge (An M₄ edge CC-RIXS) was explored by us and others as an advanced tool for probing the electronic structure and bonding properties of the actinide elements.^{13,24–29} It can be applied for all actinide elements and is now accessible at several synchrotrons all over the world.¹⁸ However, there are many aspects of the An M_{4,5} edge CC-RIXS map not yet well understood. ”

Comment (3): The following points give the examples that lead me to the conclusion about the limited novelty of the present article:

1. Experimental RIXS spectra for some systems were reported previously. For example, 3d 4f RIXS of UO₂ shown in Fig 1 is very similar to the RIXS reported in Fig 1 of the review DOI 10.1039/D1CC04851A , Previous article of authors (DOI 10.1038/ncomms16053) also reports 3d-4f RIXS of NpO₂, UO₂, PuO₂.)

Answer: Yes, this is correct that limited number of 3d4f RIXS maps of U, Np and Pu (Not for Am) have been reported previous, however the satellite was experimentally only mentioned in one map before and its potential as a tool was never explored, as the energy region of the limited number of RIXS maps is not measured large enough to catch this feature. As we point out in the introduction and other places in the manuscript, the RIXS maps are very complex, challenging to calculate and not completely understood. We provide a very extended (18 compounds, and additional 3 only for 180° scattering geometry), unique set of data and the most advanced RIXS calculations on actinides up-to-date to give deep understanding of the RIXS maps and their use for electronic structure studies and bonding properties of actinides. We provide tools to probe the localized 5f electron density and bond covalency of the actinides, not available up to date, which will answer questions in chemistry and physics of

the actinides, which have been open for decades. We think that all these points give substantial novelty of our study.

2. The instrument that is used to measure RIXS is also not new (classical Johann-type X-ray emission spectrometer with a Rowland circle of 1 m)

Answer: Yes, this is correct. We do not claim any novelty of the instrumentation. However, for the first time we report actinide 3d4d RIXS data for 180° scattering geometry.

3. The theoretical method is also not new and has been previously applied to actinides in a rather systematic manner. For example Fig 12 from Butorin DOI: 10.1021/acs.inorgchem.0c02032 reports simulations of PuO₂ 3d-4f RIXS which are similar to those reported in Figure 5 of the present manuscript.

Answer: Yes, Butorin has published calculations of 3d4f RIXS maps of actinides using partially similar theoretical approaches and we cite his work. However, he has not done comparison to experimental data and has not discussed the systematics of the intensity of the satellite peak comparing to 18 actinide compounds and its sensitivity to the localized number of 5f electrons on the An atoms, which is one of the main novelty points in our study. Note that the most of the actinide materials are not trivial, i.e., they are difficult to prepare and characterize as well as they are not readily available. It is also not trivial to perform the experiments of liquid Pu samples, which are also air-sensitive, and highly active actinide materials like Am compounds.

Butorin has not discussed the usefulness of the experimental data to benchmark the calculations, specifically 4f-5f exchange strength parameter G^0_{4f5f} could be found from the experiments. Butorin used estimated values.

Sergei Butorin has not shown how the satellite peak changes for different scattering geometries and did not discuss its sensitivity to bond covalency. We have used ligand field DFT. This theoretical approach uses DFT and the average of configuration type calculations that enables the evaluation of ligand-field parameters without scaling factors or empirical corrections. Sergei Butorin has used empirical parameters for the calculations and calculated compounds which do not have substantial An-Ligand bond covalency.

Comment (4): The discussion section of the manuscript can be also elaborated including more examples with the comparison of the electronic structure parameters (“number of 5f electrons”) obtained using the suggested method and other independent techniques (Neutron scattering, measurements of HERFD XAS at other edges, etc). From my point of view, it is not sufficient to repeat that the satellite peak can be used to determine the electronic configuration and stress the complementary to valence-band RIXS. I think it is necessary to explain why disagreement of results is sometimes obtained by comparing different techniques.

Answer: There is now extended introduction discussing different experimental techniques and their limitations. We have now 18 actinide compounds compared in

the results part of the manuscript and we extended the discussion in the results and discussion parts comparing to the absorption edge shift of the HR-XANES spectra (cf. section “Application of the satellite peak for measuring 5f electron configurations of the actinide elements”), commonly used for oxidation states and electronic configuration determination. For some cases, VB-RIXS data, which is very rare, is discussed too. Due to the limited length of the manuscript, a complete review is unfortunately not possible.

The following text was added: “ To verify whether this trend is generally valid, a multitude of actinide compounds with different 5f electron occupations, chemical environments, and physical forms were investigated in 90° π scattering geometry. The An M_4 edge CC-RIXS maps, normalized to maximum of a Lorentzian fitted to the cross section through the WL, displayed in Figure SI6, where the energy positions of the WLs and satellite peaks are marked. This procedure was performed to less affected by noise in the experimental spectra. These positions were determined by comparing cross sections obtained at different constant excitation, emission, or energy transfer values. The energy transfer cross sections (orange and blue traces in Figure 1) through the maxima of the WLs and the satellite peaks were collected and modeled with Lorentzian functions (see Figure 4a). The ratio of the areas (A) ($A_{\text{sat}}/A_{\text{WL}}$) is plotted for different n_{5f} values in Figure 4b. The uncertainties were computed by performing error propagation. The procedure is exemplified for $[\text{Cp}_3\text{U}^{\text{IV}}]^+$ in Figure 4a; for all other compounds, it is shown in Figure SI7. A different approach using the emission energy cross section (pink trace in Figure 1) were also performed, they yielded similar results and can be found in section SI5 and Figure SI8 + SI9.

5f⁰ configuration. We find that the chemical environment does not influence the intensity of the satellite peak for three different U^{VI} compounds, despite differences in their coordination environments. U^{VI} and U^{V} form trans-dioxo bonds often referred to as uranyl ($\text{UO}_2^+ = \text{U}^{\text{V}}\text{-yl}$ and $\text{UO}_2^{2+} = \text{U}^{\text{VI}}\text{-yl}$) with characteristic covalent axial binding of U with O (U-O_{ax}). These axial oxo groups are open to interaction with Lewis acids such as metal cations, which have been shown to affect the $\text{U}=\text{O}$ bond covalency. The average U-O_{ax} bond length is 1.779 Å and 1.773 Å in $[\text{U}^{\text{VI}}\text{O}_2]^{2+}\text{-Pb}$ (Figure S25a, f, h), where Pb has close interaction with O_{ax} , and $[\text{U}^{\text{VI}}\text{O}_2]^{2+}\text{-Cd}$ (Figure S25e, g, e), where Cd has no contact with O_{ax} , respectively. $\text{U}^{\text{VI}}\text{-yl}$ is fivefold coordinated in the equatorial plane for both compounds, but one O is replaced by N in $[\text{U}^{\text{VI}}\text{O}_2]^{2+}\text{-Cd}$. The small difference in bond lengths indicates a similar degree of covalency of the $\text{U}^{\text{VI}}\text{-yl}$ bond for both compounds.³⁹ However, Brager et al. found through quantum theory of atoms in molecules (QTAIM) analyses that there is a small decrease in the bond covalency of the U-O_{ax} bond for $[\text{U}^{\text{VI}}\text{O}_2]^{2+}\text{-Pb}$, corroborated by distinctly different luminescence properties. This is based on the close Pb-O_{ax} bond distance (2.887 Å), which leads to substantial interaction between Pb and O_{ax} not present in the compound with Cd ($\text{Cd-O}_{\text{ax}} = 5.681$ Å). This result is consistent with the U M_4 edge HR-XANES spectra (cf. Figure SI10) where slightly smaller (-0.2 ± 0.05 eV) A-C shift between the first and third spectral peaks, sensitive to the $\text{U}^{\text{VI}}\text{-O}_{\text{ax}}$ bond length and bond covalency, is present for the $[\text{U}^{\text{VI}}\text{O}_2]^{2+}\text{-Pb}$ compound.⁴⁰ The uranium atom is coordinated by 3 N and 2 O donors in the equatorial plane in $[\text{U}^{\text{VI}}\text{O}_2]^{2+}$ and the average $\text{U}^{\text{VI}}\text{-O}_{\text{ax}}$ bond length is 1.782 Å (Figure SI12). The spectrum of the $[\text{U}^{\text{VI}}\text{O}_2]^{2+}$ compound has smaller A-C shift (5.5 ± 0.05 eV) suggesting a smaller $\text{U}^{\text{VI}}\text{-O}_{\text{ax}}$ bond covalency compared to the $[\text{U}^{\text{VI}}\text{O}_2]^{2+}\text{-Pb}$ (6 ± 0.05 eV)/Cd (6.2 ± 0.05 eV) compounds

(Figure SI11b). Despite small or large modifications in the covalent character of the U^{VI} -Oax binding, the satellite peak remains unchanged, suggesting no sensitivity to the delocalized electron density binding the atoms.

5f¹ configuration. For the 5f¹ configuration the satellite peak becomes more intense. It is possible to compare the satellite peak intensity for two compounds $[U^{VI}O_2]^+-K$ and $[U^{VI}O_2]^+-Fe$, that have a uranyl structure with the same equatorial ligand, however, in $[U^{VI}O_2]^+-Fe$, Fe^{II} is axially bound to the U^{VI} -yl.²⁹ We showed recently that the axial U^{VI} -yl bond becomes less covalent, whereas the equatorial bond covalency increases upon Fe^{II} binding to the U^{VI} -yl oxygen atom.²⁹ Despite this change in the U^{VI} binding, well visible in the U M₄ edge HR-XANES spectra in Figure S11a, the satellite peak maintains a similar intensity. Note that the U M₄ edge HR-XANES spectrum of $[U^{VI}O_2]^+-Fe$ is slightly shifted to higher energy compared to the $[U^{VI}O_2]^+-K$ spectrum, whereas there is no intensity differences for the satellite peak. This finding illustrates that the satellite peak is more sensitive to localization of the An 5f electrons compared to the energy shift of the HR-XANES spectrum. Even more impactful is the comparison of compounds with different actinide elements but the same 5f occupation. $[Np^{VI}O_2]^{2+}$ (aq) also has a formal 5f¹ configuration and essentially the same relative satellite peak intensity likely the $[U^{VI}O_2]^+-K$ and $[U^{VI}O_2]^+-Fe$ compounds.

However, the intensity of the satellite peak is slightly higher for the $Na_2Np^{VI}_2O_7$ solid compound. We understand this deviation by carefully comparing the Np^{VI} coordination environment for $[Np^{VI}O_2]^{2+}$ (aq) and $Na_2Np^{VI}_2O_7$. The Np^{VI} -yl is equatorially coordinated by five water molecules and 6 O atoms in $[Np^{VI}O_2]^{2+}$ (aq) and $Na_2Np^{VI}_2O_7$, respectively. The $Np-O_{ax}$ and $Np-O_{eq}$ bond lengths for the two compounds ($[Np^{VI}O_2]^{2+}$ (aq) vs. $Na_2Np^{VI}_2O_7$) are $Np-O_{ax} = 1.75$ vs. 1.91 Å and $Np-O_{eq} = 2.4$ vs. 2.35 Å, respectively. In Figure SI16, we compare the Np M₄ edge HR-XANES spectra for $[Np^{VI}O_2]^{2+}$ (aq) and $Na_2Np^{VI}_2O_7$ (Section SI9, Figures SI17-18). There is a substantial change in the A-C shift from 5.2 eV ($Na_2Np^{VI}_2O_7$) to 6.2 eV ($[Np^{VI}O_2]^{2+}$ (aq)) that suggests increased $Np-O_{ax}$ bond covalency in $[Np^{VI}O_2]^{2+}$ (aq) and agrees with the much shorter Np^{VI} -Oax bond length for $[Np^{VI}O_2]^{2+}$ (aq). The absence of an energy shift of the Np M₄ edge HR-XANES spectra signifies that the electron density on Np does not change notably, likely due to stronger Np^{VI} -O_{eq} interactions in the solid compound, as suggested by the smaller $Np-O_{eq}$ bond length. The small increase of satellite intensity for the solid Np^{VI} compound can be explained by minor increase of localized 5f electron density on Np^{VI} in the solid compound. The HR-XANES and satellite peak probe electron density with different level of localization between the An-ligands bonds, thereby providing an advanced understanding of the bonding situation. The satellite peak allows for comparison of the electron density most localized on Np for the two Np compounds.

5f² configuration. Here, again, the insensitivity of the satellite peak intensity to the chemical environment can be demonstrated by comparing a organometallic $[Cp^*U^{IV}]^+ = Cp^*_3U^{IV}Cl^*Et_2O$ coordination compound to the bulk material UO_2 , which have very different electronic and geometry structures. The drastic 1 eV shift to lower energy in the U M₄ edge HR-XANES WL position (Figure SI13) for the spectrum of $[Cp^*_3U^{IV}]^+$ is confronted with a very similar relative satellite intensity for the two compounds. Combining these two experimental results, we can conclude that there is greater delocalization of 5f electron density for UO_2 . The $[Pu^{VI}O_2]^{2+}$ (aq) (HR-XANES and UV-Vis in Figure SI19) and $[Np^{VI}O_2]^{2+}$ (aq) (HR-XANES and UV-Vis in Figure

SI16) deviate from the values measured for the satellite peak intensity for the U compounds. Both $\text{Np}^{\text{VI}}/\text{Pu}^{\text{VI}}\text{-O}_{\text{ax}}$ bonds are covalent. However, due to the increased nuclear charge, the 5f electrons become more localized for Pu and the overlap driven bond covalency is likely decreasing in $\text{Pu}^{\text{VI}}\text{-yl}$ as previously suggested.⁴⁰ This is reflected in the large satellite peak intensity revealing a larger relative number of 5f electrons localized on Pu^{VI} compared to Np^{VI} .

5f³ configuration. The satellite peak intensity can help to shed light on long standing discussions, such as the heavily discussed formal oxidation state of low valent actinide compounds, i.e. An^{I} , An^{II} or An^{III} .^{41–46} Here, we measured the satellite intensity of $[\text{Cp}''_3\text{U}^{\text{II}}]^-$, which has been theoretically confirmed to be a $5f^3d^1$ compound⁴⁷, and found that it is slightly smaller than that of $\text{Np}^{\text{IV}}\text{O}_2$ ($n_{5f} = 5f^3$). A comparison of the U M₄ edge HR-XANES of $[\text{Cp}''_3\text{U}^{\text{II}}]^-$ to the isostructural U^{III} ($n_{5f} = 5f^3$) compound in Figure SI13 yielded very similar WL positions, corresponding to similar electron density on U. Additional VB-RIXS experiments would need to be performed to identify the 6d portion of the electron occupation, however the NMR of the compound is very similar to the literature spectra (Figure SI14), confirming the U^{II} compound is intact. We find a 1.0 eV energy shift to lower energies of the U M₄ edge HR-XANES spectrum compared to the spectrum of the structurally similar $[\text{Cp}''_3\text{U}^{\text{IV}}]^+$ (cf. Figure S13). Interestingly, the satellite peak intensities are similar for both compounds, suggesting rather delocalized part of the larger 5f electron density on U in $[\text{Cp}''_3\text{U}^{\text{II}}]^-$.

5f⁴ and 5f⁵ configurations. Another important example is the comparison of liquid Pu^{IV} (aq) and solid $\text{Pu}^{\text{IV}}\text{O}_2$. The satellite peak intensity is larger for Pu^{IV} (aq) suggesting more localized 5f electron density on Pu compared to $\text{Pu}^{\text{IV}}\text{O}_2$. The Pu M₄ edge HR-XANES spectra shown in Figure SI20 exhibit nearly no energy shift and a small shift was present for Pu M₅ edge HR-XANES data, revealing that the total 5f electron density on Pu is similar.⁴⁸ Quantum chemical calculations reported that $\text{Pu}^{\text{IV}}\text{O}_2$ is a solid compound with larger Pu-O bond covalency opposed to the more ionic $\text{Pu}^{\text{IV}}\text{-H}_2\text{O}$ chemical bond in Pu^{IV} (aq), which agrees very well with our previous experimental results. For the 5f⁵ case we see the largest discrepancies from the theoretical predictions with $\text{Am}^{\text{IV}}\text{O}_2$ having a higher intensity than the 5f⁴ systems. However, this can be partially rationalized since the non-ideal geometry and conditions of the experiment for this absorption edge introduced larger uncertainties in the determination of the satellite peak area. The satellite peak area for Pu^{III} (aq) has lower intensity than for Pu^{IV} (aq) and thereby, it is consistent with the theoretical prediction in Figure 4c.

5f⁶ configuration. For 5f⁶ electron occupation, the theory predicts a break in the upwards trend and a much lower intensity than the previous 5f occupations. The experimentally determined relative satellite intensity of the $\text{Am}^{\text{III}}\text{VO}_3$ amounts to $5.2 \pm 0.5\%$ of the WL intensity quantitatively matching the theoretically predicted value of $5.2 \pm 0.2\%$ for 5f⁶ and following the same trend. The Pu and Am M₅ and/or M₄ edge HR-XANES data for the 5f⁴, 5f⁵ and 5f⁶ compounds are depicted in Figures SI20, SI21, SI22. It is remarkable that the Am M₄ edge HR-XANES exhibits no energy shift for Am^{III} and Am^{IV} reported here for the first time. A clear energy shift of 1.35 eV is present between the Am M₅ edge spectra of $\text{Am}^{\text{IV}}\text{O}_2$ and $\text{Am}^{\text{III}}\text{VO}_3$. The Pu M₄ edge HR-XANES spectrum of Pu^{III} (aq) is reported here also for the first time.

Our results show that the satellite peak measured at 90° π scattering geometry is sensitive to the number of f electrons localized on the actinide atom, and its integral

intensity (area of the cross section of the peak) can be used to obtain their relative count.

Figure 4. Quantitative evaluation of the satellite intensity in 90° π scattering geometry. **a** Energy transfer cross sections of $\text{Cp}_3\text{U}^{\text{IV}}\text{Cl}^*\text{Et}_2\text{O}$ ($[\text{Cp}_3\text{U}^{\text{IV}}]^+$) through the maxima of the satellite peak and the WL maximum, modelled with Lorentzian functions to showcase the analytical procedure. The intensity of the satellite curve is multiplied by 20. **b** Experimentally determined relative intensities of satellite peaks of An M_4 -edge CC-RIXS maps of 18 actinide compounds sorted by 5f shell occupation. **c** Relative intensities of satellite peaks of An M_4 -edge CC-RIXS maps obtained by performing atomic calculations of U^{VI} to U^0 corresponding to $5f^0$ – $5f^6$ occupancy. The parameters are here the same as for the calculations in Figure 2c–d. The values for the satellite peak intensities in **b** and **c** are given by the ratio between the areas of the curves fitted to the satellite and WL cross sections, respectively, as exemplary shown in **a**.

“

Reviewer #2

Comment (1): The authors present x-ray spectroscopy fingerprints of the number of 5f electrons in actinides using the information embedded in the M_4 edge core-to-core resonant inelastic x-ray scattering (CC-RIXS) satellite line. In addition to the possibility to use of M_4 edge CC-RIXS spectra for 5f electron counting, the authors propose also to use the same spectra to benchmark the correct scaling of the monopole part of the Coulomb interaction, i.e., G_0 and F_0 4f-5f overlap integrals, which usually need ad-hoc scaling to match various kinds of spectroscopy data of the actinides. They also speculate on the sensitivity of the method to the covalent character of the actinide-ligand bonding.

The experiment and theory are definitely state-of-the-art and the methodology is sound. Performing synchrotron RIXS on actinide compounds alone is already a

challenge as such and the necessary infrastructure exists in Karlsruhe. The theoretical simulations are based on the most highly developed tools such as Quanty, which is designed to perform exactly these type of calculations.

Answer: We thank the review for recognizing the high value of our work!

Comment (2): In general terms, I fully agree with the authors that “experimental and computational results show that the complex electronic structure and binding properties of the actinide elements can be revealed by advanced spectroscopic tools and computations”.

One of the biggest challenges I find is how to prove an universality of the 5f electron “counting” scheme. The proposal to accept this a universal behavior is now based on the two U(V) samples (U-V and U-V-Fe) showing similar satellite intensity, together with the fact that the An-O₂ oxides following the expected trend from theory until 5f⁵ occupation. From the theory side this is culminated in the lines 179-181: is the atomic multiplet theory justified in systems with strong effects of bond covalency, orbital mixing, or spherical symmetry breaking? The usefulness of the universality would arise if someone wishes to use the concept to analyze an entirely “unknown” sample where the 5f occupation and many other electronic structure parameters may be not known. The proposed trend seems nicely followed by the 6 samples here, but are we certain that any other 5f material in the 5f⁰-5f⁴ series will definitely follow this trend regardless of its bond covalency, orbital mixing, or spherical symmetry breaking? This claim would benefit from more evidence than the 6 samples presented here, out of which just two duplicate.

Answer: We agree with the comment that the atomic theory shows an indicative trend for the intensity of the satellite peak for different integer 5f electron configurations. This is also pointed out in the manuscript. Variations for the same nominal oxidation state or ground state electron configuration is to be expected. In addition to the U(V), U(V)-Fe example, we studied now several other systems with small and large differences in ligand environment and chemical bonding properties. We compared results for 18 compounds and find that the electron density localized the An elements can vary, which is not surprising and expected, but the general trend is preserved. Within a set of compounds with completely difference chemical environments and bonding characteristics, it is possible to compare the 5f electron density strongly localized on the An atom. We performed quantitative evaluation of the satellite peak intensity and plotted the result as a function of the 5f electron configuration in Figure 4 (b). The theoretical trend for the atomic calculations of U is shown in Figure 4 (c). The changes in intensity of the satellite peak is compared to changes of the An M_{4,5} edge HR-XANES spectra. It is shown that the satellite peak is more sensitive to the localized 5f electron density compared to the absorption edge energy shift.

The following text is now added on page 8ff;

“To verify whether this trend is generally valid, a multitude of actinide compounds with different 5f electron occupations, chemical environments, and physical forms were investigated in 90° π scattering geometry. The An M₄ edge CC-RIXS maps, normalized to maximum of a Lorentzian fitted to the cross section through the WL, displayed in Figure S16, where the energy positions of the WLs and satellite peaks are marked. This procedure was performed to less affected by noise in the experimental spectra. These positions were determined by comparing cross sections obtained at different constant excitation, emission,

or energy transfer values. The energy transfer cross sections (orange and blue traces in Figure 1) through the maxima of the WLs and the satellite peaks were collected and modeled with Lorentzian functions (see Figure 4a). The ratio of the areas (A) ($A_{\text{sat}}/A_{\text{WL}}$) is plotted for different n_{5f} values in Figure 4b. The uncertainties were computed by performing error propagation. The procedure is exemplified for $[\text{Cp}_3\text{U}^{\text{IV}}]^+$ in Figure 4a; for all other compounds, it is shown in Figure SI7. A different approach using the emission energy cross section (pink trace in Figure 1) were also performed, they yielded similar results and can be found in section SI5 and Figure SI8 + SI9.

5f⁰ configuration. We find that the chemical environment does not influence the intensity of the satellite peak for three different U^{VI} compounds, despite differences in their coordination environments. U^{VI} and U^{V} form trans-dioxo bonds often referred to as uranyl ($\text{UO}_2^+ = \text{U}^{\text{V-yl}}$ and $\text{UO}_2^{2+} = \text{U}^{\text{VI-yl}}$) with characteristic covalent axial binding of U with O (U-O_{ax}). These axial oxo groups are open to interaction with Lewis acids such as metal cations, which have been shown to affect the $\text{U}=\text{O}$ bond covalency. The average U-O_{ax} bond length is 1.779 Å and 1.773 Å in $[\text{U}^{\text{VI}}\text{O}_2]^{2+}\text{-Pb}$ (Figure S25a, f, h), where Pb has close interaction with O_{ax} , and $[\text{U}^{\text{VI}}\text{O}_2]^{2+}\text{-Cd}$ (Figure S25e, g, e), where Cd has no contact with O_{ax} , respectively. $\text{U}^{\text{VI-yl}}$ is fivefold coordinated in the equatorial plane for both compounds, but one O is replaced by N in $[\text{U}^{\text{VI}}\text{O}_2]^{2+}\text{-Cd}$. The small difference in bond lengths indicates a similar degree of covalency of the $\text{U}^{\text{VI-yl}}$ bond for both compounds.³⁹ However, Brager et al. found through quantum theory of atoms in molecules (QTAIM) analyses that there is a small decrease in the bond covalency of the U-O_{ax} bond for $[\text{U}^{\text{VI}}\text{O}_2]^{2+}\text{-Pb}$, corroborated by distinctly different luminescence properties. This is based on the close Pb-O_{ax} bond distance (2.887 Å), which leads to substantial interaction between Pb and O_{ax} not present in the compound with Cd ($\text{Cd-O}_{\text{ax}} = 5.681$ Å). This result is consistent with the U M_4 edge HR-XANES spectra (cf. Figure SI10) where slightly smaller (-0.2 ± 0.05 eV) A-C shift between the first and third spectral peaks, sensitive to the $\text{U}^{\text{VI}}\text{-O}_{\text{ax}}$ bond length and bond covalency, is present for the $[\text{U}^{\text{VI}}\text{O}_2]^{2+}\text{-Pb}$ compound.⁴⁰ The uranium atom is coordinated by 3 N and 2 O donors in the equatorial plane in $[\text{U}^{\text{VI}}\text{O}_2]^{2+}$ and the average $\text{U}^{\text{VI}}\text{-O}_{\text{ax}}$ bond length is 1.782 Å (Figure SI12). The spectrum of the $[\text{U}^{\text{VI}}\text{O}_2]^{2+}$ compound has smaller A-C shift (5.5 ± 0.05 eV) suggesting a smaller $\text{U}^{\text{VI}}\text{-O}_{\text{ax}}$ bond covalency compared to the $[\text{U}^{\text{VI}}\text{O}_2]^{2+}\text{-Pb}$ (6 ± 0.05 eV)/Cd (6.2 ± 0.05 eV) compounds (Figure SI11b). Despite small or large modifications in the covalent character of the $\text{U}^{\text{VI}}\text{-O}_{\text{ax}}$ binding, the satellite peak remains unchanged, suggesting no sensitivity to the delocalized electron density binding the atoms.

5f¹ configuration. For the 5f¹ configuration the satellite peak becomes more intense. It is possible to compare the satellite peak intensity for two compounds $[\text{U}^{\text{VO}_2}]^+\text{-K}$ and $[\text{U}^{\text{VO}_2}]^+\text{-Fe}$, that have a uranyl structure with the same equatorial ligand, however, in $[\text{U}^{\text{VO}_2}]^+\text{-Fe}$, Fe^{II} is axially bound to the $\text{U}^{\text{V-yl}}$.²⁹ We showed recently that the axial $\text{U}^{\text{V-yl}}$ bond becomes less covalent, whereas the equatorial bond covalency increases upon Fe^{II} binding to the $\text{U}^{\text{V-yl}}$ oxygen atom.²⁹ Despite this change in the U^{V} binding, well visible in the U M_4 edge HR-XANES spectra in Figure S11a, the satellite peak maintains a similar intensity. Note that the U M_4 edge HR-XANES spectrum of $[\text{U}^{\text{VO}_2}]^+\text{-Fe}$ is slightly shifted to higher energy compared to the $[\text{U}^{\text{VO}_2}]^+\text{-K}$ spectrum, whereas there is no intensity differences for the satellite peak. This finding illustrates that the satellite peak is more sensitive to localization of the An 5f electrons compared to the energy shift of the HR-XANES spectrum. Even more impactful is the comparison of compounds with different actinide elements but the same 5f occupation. $[\text{Np}^{\text{VI}}\text{O}_2]^{2+}$ (aq) also has a formal 5f¹ configuration and essentially the same relative satellite peak intensity likely the $[\text{U}^{\text{VO}_2}]^+\text{-K}$ and $[\text{U}^{\text{VO}_2}]^+\text{-Fe}$ compounds. However, the intensity of the satellite peak is slightly higher for the $\text{Na}_2\text{Np}^{\text{VI}}_2\text{O}_7$ solid compound. We understand this deviation by carefully comparing the Np^{VI} coordination environment for $[\text{Np}^{\text{VI}}\text{O}_2]^{2+}$ (aq) and $\text{Na}_2\text{Np}^{\text{VI}}_2\text{O}_7$. The $\text{Np}^{\text{VI-yl}}$ is equatorially coordinated by five water molecules and 6 O atoms in $[\text{Np}^{\text{VI}}\text{O}_2]^{2+}$ (aq) and $\text{Na}_2\text{Np}^{\text{VI}}_2\text{O}_7$, respectively. The Np-O_{ax} and Np-O_{eq} bond lengths for the two compounds ($[\text{Np}^{\text{VI}}\text{O}_2]^{2+}$ (aq) vs. $\text{Na}_2\text{Np}^{\text{VI}}_2\text{O}_7$) are $\text{Np-O}_{\text{ax}} = 1.75$ vs. 1.91 Å and $\text{Np-O}_{\text{eq}} = 2.4$ vs. 2.35 Å, respectively. In Figure SI16, we compare the Np M_4 edge HR-XANES spectra for $[\text{Np}^{\text{VI}}\text{O}_2]^{2+}$ (aq) and $\text{Na}_2\text{Np}^{\text{VI}}_2\text{O}_7$ (Section SI9, Figures

SI17-18). There is a substantial change in the A-C shift from 5.2 eV ($\text{Na}_2\text{Np}^{\text{VI}}\text{O}_7$) to 6.2 eV ($[\text{Np}^{\text{VI}}\text{O}_2]^{2+}$ (aq)) that suggests increased Np- O_{ax} bond covalency in $[\text{Np}^{\text{VI}}\text{O}_2]^{2+}$ (aq) and agrees with the much shorter Np^{VI}- O_{ax} bond length for $[\text{Np}^{\text{VI}}\text{O}_2]^{2+}$ (aq). The absence of an energy shift of the Np M_4 edge HR-XANES spectra signifies that the electron density on Np does not change notably, likely due to stronger Np^{VI}- O_{eq} interactions in the solid compound, as suggested by the smaller Np- O_{eq} bond length. The small increase of satellite intensity for the solid Np^{VI} compound can be explained by minor increase of localized 5f electron density on Np^{VI} in the solid compound. The HR-XANES and satellite peak probe electron density with different level of localization between the An-ligands bonds, thereby providing an advanced understanding of the bonding situation. The satellite peak allows for comparison of the electron density most localized on Np for the two Np compounds.

5f² configuration. Here, again, the insensitivity of the satellite peak intensity to the chemical environment can be demonstrated by comparing a organometallic $[\text{Cp}''_3\text{U}^{\text{IV}}]^+ = \text{Cp}''_3\text{U}^{\text{IV}}\text{Cl}\cdot\text{Et}_2\text{O}$ coordination compound to the bulk material UO_2 , which have very different electronic and geometry structures. The drastic 1 eV shift to lower energy in the U M_4 edge HR-XANES WL position (Figure SI13) for the spectrum of $[\text{Cp}''_3\text{U}^{\text{IV}}]^+$ is confronted with a very similar relative satellite intensity for the two compounds. Combining these two experimental results, we can conclude that there is greater delocalization of 5f electron density for UO_2 . The $[\text{Pu}^{\text{VI}}\text{O}_2]^{2+}$ (aq) (HR-XANES and UV-Vis in Figure SI19) and $[\text{Np}^{\text{V}}\text{O}_2]^+$ (aq) (HR-XANES and UV-Vis in Figure SI16) deviate from the values measured for the satellite peak intensity for the U compounds. Both Np^{VI}/ Pu^{VI} - O_{ax} bonds are covalent. However, due to the increased nuclear charge, the 5f electrons become more localized for Pu and the overlap driven bond covalency is likely decreasing in Pu^{VI} -yl as previously suggested.⁴⁰ This is reflected in the large satellite peak intensity revealing a larger relative number of 5f electrons localized on Pu^{VI} compared to Np^{VI} .

5f³ configuration. The satellite peak intensity can help to shed light on long standing discussions, such as the heavily discussed formal oxidation state of low valent actinide compounds, i.e. An^I, An^{II} or An^{III}.⁴¹⁻⁴⁶ Here, we measured the satellite intensity of $[\text{Cp}''_3\text{U}^{\text{II}}]$, which has been theoretically confirmed to be a 5f³d¹ compound⁴⁷, and found that it is slightly smaller than that of $\text{Np}^{\text{IV}}\text{O}_2$ ($n_{5f} = 5f^3$). A comparison of the U M_4 edge HR-XANES of $[\text{Cp}''_3\text{U}^{\text{II}}]$ to the isostructural U^{III} ($n_{5f} = 5f^3$) compound in Figure SI13 yielded very similar WL positions, corresponding to similar electron density on U. Additional VB-RIXS experiments would need to be performed to identify the 6d portion of the electron occupation, however the NMR of the compound is very similar to the literature spectra (Figure SI14), confirming the U^{II} compound is intact. We find a 1.0 eV energy shift to lower energies of the U M_4 edge HR-XANES spectrum compared to the spectrum of the structurally similar $[\text{Cp}''_3\text{U}^{\text{IV}}]^+$ (cf. Figure SI13). Interestingly, the satellite peak intensities are similar for both compounds, suggesting rather delocalized part of the larger 5f electron density on U in $[\text{Cp}''_3\text{U}^{\text{II}}]$.

5f⁴ and 5f⁵ configurations. Another important example is the comparison of liquid Pu^{IV} (aq) and solid $\text{Pu}^{\text{IV}}\text{O}_2$. The satellite peak intensity is larger for Pu^{IV} (aq) suggesting more localized 5f electron density on Pu compared to $\text{Pu}^{\text{IV}}\text{O}_2$. The Pu M_4 edge HR-XANES spectra shown in Figure SI20 exhibit nearly no energy shift and a small shift was present for Pu M_5 edge HR-XANES data, revealing that the total 5f electron density on Pu is similar.⁴⁸ Quantum chemical calculations reported that $\text{Pu}^{\text{IV}}\text{O}_2$ is a solid compound with larger Pu-O bond covalency opposed to the more ionic $\text{Pu}^{\text{IV}}\text{-H}_2\text{O}$ chemical bond in Pu^{IV} (aq), which agrees very well with our previous experimental results. For the 5f⁵ case we see the largest discrepancies from the theoretical predictions with $\text{Am}^{\text{IV}}\text{O}_2$ having a higher intensity than the 5f⁴ systems. However, this can be partially rationalized since the non-ideal geometry and conditions of the experiment for this absorption edge introduced larger uncertainties in the determination of the satellite peak area. The satellite peak area for Pu^{III} (aq) has lower intensity than for Pu^{IV} (aq) and thereby, it is consistent with the theoretical prediction in Figure 4c.

5f⁶ configuration. For 5f⁶ electron occupation, the theory predicts a break in the upwards trend and a much lower intensity than the previous 5f occupations. The experimentally determined relative satellite intensity of the $\text{Am}^{\text{III}}\text{VO}_3$ amounts to $5.2 \pm 0.5\%$ of the WL intensity quantitatively matching the theoretically predicted value of $5.2 \pm 0.2\%$ for 5f⁶ and following the same trend. The Pu and Am M_5 and/or M_4 edge HR-XANES data for the 5f⁴, 5f⁵ and 5f⁶

compounds are depicted in Figures SI20, SI21, SI22. It is remarkable that the Am M₄ edge HR-XANES exhibits no energy shift for Am^{III} and Am^{IV} reported here for the first time. A clear energy shift of 1.35 eV is present between the Am M₅ edge spectra of Am^{IV}O₂ and Am^{III}VO₃. The Pu M₄ edge HR-XANES spectrum of Pu^{III} (aq) is reported here also for the first time. Our results show that the satellite peak measured at 90° π scattering geometry is sensitive to the number of f electrons localized on the actinide atom, and its integral intensity (area of the cross section of the peak) can be used to obtain their relative count.

Figure 4. Quantitative evaluation of the satellite intensity in 90° π scattering geometry. **a** Energy transfer cross sections of $Cp^*_3U^{IV}Cl^*Et_2O$ ($[Cp^*_3U^{IV}]^+$) through the maxima of the satellite peak and the WL maximum, modelled with Lorentzian functions to showcase the analytical procedure. The intensity of the satellite curve is multiplied by 20. **b** Experimentally determined relative intensities of satellite peaks of An M₄-edge CC-RIXS maps of 18 actinide compounds sorted by 5f shell occupation. **c** Relative intensities of satellite peaks of An M₄-edge CC-RIXS maps obtained by performing atomic calculations of U^{VI} to U⁰ corresponding to 5f⁰ -5f⁶ occupancy. The parameters are here the same as for the calculations in Figure 2c-d. The values for the satellite peak intensities in **b** and **c** are given by the ratio between the areas of the curves fitted to the satellite and WL cross sections, respectively, as exemplary shown in **a**.”

Comment (3): Besides, how about the 5f6 case, for which theory suggests that the satellite falls nearly back to the energy and intensity of 5f2 (fig. 2c).

Answer: We studied now 5f5 and 5f6, in addition to 5f4, systems (cf. Figure 4) and clearly observe decrease in intensity of the satellite peak for the 5f6 configuration showing that the experimental data follows the theoretical prediction. Please see section cited above.

Comment (4): Also, about “Application of the satellite peak for probing bond covalency of the actinide-ligand bond”: the authors conclude that the satellite intensity is expected to decrease in the 90deg sigma geometry when covalency increases. However, this note is written without presenting quantitative data or more substantial predictive power. It remains unclear how this could be evaluated experimentally in a quantitative manner.

Answer: We also improved upon this part by investigating 6 U(VI)O₂+ (uranyl) compounds, where 5 of them are coordinated by 5 or 6 O atoms. One compound has one O exchanged by N. The strength of U-Oequatorial ligands interaction is modified

by close or long/no interaction of Pb/Ag/Cd with the O_{ax}. This leads to small variations of the U(VI)-O_{ax} bond but stronger U(VI)-O_{eq} bond variations and allows to systematically study the satellite peak intensity at the 180° scattering geometry. Note that this geometry is equivalent to 90° sigma polarization scattering geometry, which was experimentally not possible. We found that the satellite peak is split forming two distinct peaks and the satellite on the bottom of the maps predicted by the theory was observed. Note that this peak is demonstrated here for the first time. We found a clear trend between integral intensities of the two top satellite peaks and variations of U-ligands bond covalency. In addition, we performed ligand field DFT calculations for two of the compounds and observe the same trends as in the experiments. The text discussing the satellite peak as a tool to probe covalency of bonds is on page 16 see below:

“In order to verify this deliberation, we recorded U M₄ edge CC-RIXS maps in 180° π scattering geometry for a series of U^{VI} compounds. In addition to the [U^{VI}O₂]²⁺-Pb and [U^{VI}O₂]²⁺-Cd compounds discussed above, U^{VI}-yl coordinated by 5 or 6 O atoms in the equatorial plane with or without direct interaction of a metal cation (M=Pb, Ag or Cd) with the O_{ax} atom were studied (Figure SI24); details on the coordination structures are given in Figure SI25.^{39,50,51} Furthermore, we performed ligand field density function theory (LF-DFT) calculations of the U M₄ edge CC-RIXS maps (cf. SI section 19). This theoretical approach uses DFT and the average of configuration type calculations that enables the evaluation of ligand-field parameters without scaling factors or empirical corrections. The experimental and calculated U M₄ edge CC-RIXS maps and cross sections show clear variations for the [U^{VI}O₂]²⁺-Pb and [U^{VI}O₂]²⁺-Cd compounds displayed in Figure 6 (cf. Figure SI26, SI27). The RIXS maps and their cross sections at constant emission energies at the maxima of the lower satellite peak or the WL are similar for experiment and theory and reveal two satellites on the top and one satellite on the bottom of the CC-RIXS maps. To decipher the influence of the inter-electron repulsion on the satellite peaks, we performed two different types of calculations (1) omitting both 3d-5f and 4f-5f inter-electron interaction parameters (Figure SI28e and SI28f) or (2) implementing 3d-5f but still not considering 4f-5f interactions (Figure SI28g and h). The results depicted in Figure S28 illustrate that the two satellite peaks equally appear but only when the 4f-5f interaction is considered. For [U^{VI}O₂]²⁺-Cd this 4f-5f interaction is smaller leading to larger intensity of the satellite peaks. This result strongly suggests that the integral intensity of the satellite peaks measured in 180° π scattering geometry is sensitive to the size of the 4f-5f interaction. In turn, the size of the 4f-5f interaction depends on the 5f electron density distribution between U and the ligands. The larger is the U-ligands bond covalency, the smaller will be the inter-electron repulsion interaction as predicted by the Nephelauxetic effect,⁵² which is correlated here with the integral intensity of the satellite peaks (cf. Figure S29). It states that for a free ion, the inter-electron repulsion parameters are the biggest possible. When the metal ion is coordinated with ligands, the value of inter-electron repulsion parameter (4f-5f in our case) ultimately becomes smaller than in the free ion case; the extent of this reduction is related to the electron density by the metal center and varied upon the covalency of the metal-ligand bond.

The integral intensity of the two satellite peaks is larger for [U^{VI}O₂]²⁺-Cd (Figure SI25a, f, h) than [U^{VI}O₂]²⁺-Pb (Figure SI25g, e, i) for both experiment and theory (cf. Figure 6) suggesting smaller U-ligands bond covalency due to less electron density on U in [U^{VI}O₂]²⁺-Cd. As previously discussed, the U-O_{ax} bond covalency is slightly lower for the [U^{VI}O₂]²⁺-Pb due to the Pb-O_{ax} competitive interaction. However, Brager

et al. showed that the equatorial interactions are stronger for close M-O_{ax}. Considering that the integral intensity of the satellite peaks is smaller and distinctly different for the compounds with direct M-O_{ax} interactions [U^{VI}O₂]²⁺-Pb and [U^{VI}O₂]²⁺-Ag (structure Figure SI25 a and c) compared to compounds with no interaction ([U^{VI}O₂]²⁺-Cd), as shown in Figure SI26, we can conclude that the satellite peak recorded in 180° π scattering geometry appears sensitive to the total metal-ligand bond covalency. Note that the A-C shift in the HR-XANES spectra exhibits small variations suggesting only very small U-O_{ax} bond covalency differences for these compounds. The satellite peak has the largest intensity for the [U^{VI}O₂]²⁺ organometallic compound indicating the smallest U-equatorial ligands bond covalency. The clear variations of the intensity of the satellite peak for the 180° π scattering geometry compared to the lack of intensity for the 90° π scattering geometry collaborates the theoretical prediction that the peak at 180° π scattering geometry will be sensitive to An-ligands bond covalency.

Comment (5): The satellite sits on a tail of the white line peak and its extraction has uncertainty which probably is reflected in the error bars in Fig. 6b, but the uncertainty has not been discussed. Some more quantitativity would be brought by plotting the known 5f electron number as a function of the satellite peak intensity (or vice versa) after its extraction from the data, with error bars, compared to the theory, and the systematic and statistical error bars should be discussed. This would give hint of the accuracy of the method, which would be needed if universality is true.

Answer: For the revised version of the manuscript quantitative analyses of the satellite peak intensity were performed including error propagation analyses. The experimental trend is compared to the theoretical trends in Figure 4. The uncertainties are shown in Figure 4. The following text is now added in page 8.

“To verify whether this trend is generally valid, a multitude of actinide compounds with different 5f electron occupations, chemical environments, and physical forms were investigated in 90° π scattering geometry. The An M₄ edge CC-RIXS maps, normalized to maximum of a Lorentzian fitted to the cross section through the WL, displayed in Figure SI6, where the energy positions of the WLs and satellite peaks are marked. This procedure was performed to less affected by noise in the experimental spectra. These positions were determined by comparing cross sections obtained at different constant excitation, emission, or energy transfer values. The energy transfer cross sections (orange and blue traces in Figure 1) through the maxima of the WLs and the satellite peaks were collected and modeled with Lorentzian functions (see Figure 4a). The ratio of the areas (A) ($A_{\text{sat}}/A_{\text{WL}}$) is plotted for different n_{5f} values in Figure 4b. The uncertainties were computed by performing error propagation. The procedure is exemplified for [Cp₃U^{IV}]⁺ in Figure 4a; for all other compounds, it is shown in Figure SI7. A different approach using the emission energy cross section (pink trace in Figure 1) were also performed, they yielded similar results and can be found in section SI5 and Figure SI8 + SI9.”

Comment (6): The authors conclude on lines 322-324: “For example, there is an intensive discussion of the formal oxidation state of low valent actinide compounds, i.e. AnI, II or III.(Refs.29–33) Here, measuring the satellite peak could be used to unambiguously determine the formal oxidation state of the actinide element.” Did the

authors try this already? Some results along these lines would make the present case stronger.

Answer: For the revised version of the manuscript we now studied also a U(II) and U(IV) compounds with very similar coordination environments. The following text is now added on page 10

“5f² configuration. Here, again, the insensitivity of the satellite peak intensity to the chemical environment can be demonstrated by comparing an organometallic [Cp⁺₃U^{IV}]⁺ = Cp⁺₃U^{IV}Cl*Et₂O coordination compound to the bulk material UO₂, which have very different electronic and geometry structures. The drastic 1 eV shift to lower energy in the U M₄ edge HR-XANES WL position (Figure SI13) for the spectrum of [Cp⁺₃U^{IV}]⁺ is confronted with a very similar relative satellite intensity for the two compounds. Combining these two experimental results, we can conclude that there is greater delocalization of 5f electron density for UO₂. “

And

“5f³ configuration. The satellite peak intensity can help to shed light on long standing discussions, such as the heavily discussed formal oxidation state of low valent actinide compounds, i.e. An^I, An^{II} or An^{III}.^{41–46} Here, we measured the satellite intensity of [Cp⁺₃U^{II}]⁻, which has been theoretically confirmed to be a 5f³d¹ compound⁴⁷, and found that it is slightly smaller than that of Np^{IV}O₂ (n_{5f} = 5f³). A comparison of the U M₄ edge HR-XANES of [Cp⁺₃U^{II}]⁻ to the isostructural U^{III} (n_{5f} = 5f³) compound in Figure SI13 yielded very similar WL positions, corresponding to similar electron density on U. Additional VB-RIXS experiments would need to be performed to identify the 6d portion of the electron occupation, however the NMR of the compound is very similar to the literature spectra (Figure SI14), confirming the U^{II} compound is intact. We find a 1.0 eV energy shift to lower energies of the U M₄ edge HR-XANES spectrum compared to the spectrum of the structurally similar [Cp⁺₃U^{IV}]⁺ (cf. Figure S13). Interestingly, the satellite peak intensities are similar for both compounds, suggesting rather delocalized part of the larger 5f electron density on U in [Cp⁺₃U^{II}]⁻.”

Comment (7): Is it necessary to measure the full RIXS 2D plane, or is there way to determine how to do only the cut in constant emission energy and/or constant excitation energy scans? This would make data acquisition faster, but the possibility to "miss" the peak without full 2D RIXS measurement may be a big risk.

Answer: It is possible to measure only a few lines along the excitation energy at constant emission energy position to identify the maximum intensity of the satellite peak. However, it is important to normalize the measured RIXS cross sections. One possibility is to use for normalization the intensity of a normal emission line measured well above the absorption edge. However, we found that the most reliable normalization is to the maximum of the main resonant peak (White line). The ratio of the WL/satellite peak areas are then compared for different compounds (see answer to question X). For this approach, a complete RIXS map needs to be measured. The energy resolution is important to better resolve the satellite peak from the WL. However, even for modest experimental energy resolution it is possible to extract reliable information.

Comment (8): The authors mention that deviating from the 90deg pi scattering geometry diminishes the correspondence between satellite characteristics and 5f occupation, but does not entirely destroy it. Something toward this is shown in Fig.2e, which notes for 5f0 case that there will be non-negligible satellite intensity when going away from 90 degree geometry. I should say that this is maybe not clear enough to illustrate the change in the 5f occupancy trend in slightly-out-of-90deg geometry, maybe better would be a figure like Fig. 2c but for analyzer 1. That kind of figure would carry more information on the sensitivity of the trend to the scattering angle than merely the current Fig. 2c.

Answer: We have now two new figures in Figure 2 (e) and (f). They are measured for the same uranyl compound at 90° and 180° scattering geometry. The appearance of a double satellite peak for the 180° scattering geometry is clearly visible. The comparison for cr. 1 and cr. 3 is now in Figure S14. The theoretical trend for analyzer crystal 1 is now also shown in Figure S14+5.

Comment (9): In Fig. 2f two different samples are compared with two different scattering angles. This does not make directly sense to me, why the comparison of different scattering geometries is not done for one single sample? It is not certain to the reader that the two samples satellite intensities can be directly compared, there may be differences in density and hence normalization, but more importantly in bond covalency, etc may be different. The reason is probably due to some experimental constraint at the time when the measurements were done, but it decreases the systematic character of the reported study.

Answer: We agree with the reviewer that this comparison was not ideal and it was related to lack of appropriate data for the same compound. We did study now three different 5f0 uranyl(VI) compounds for 90° and 180° scattering geometry and clearly demonstrate the differences in satellite peak intensity. This is shown in Figure 2 (e) and (f), Figure 4 (b) and Figure S24+S126.

Comment (10): It is interesting to see the prediction of a second satellite in the low-energy (in the text L257: "high-energy"?) tail measured at constant excitation energy in Fig. 6a. It seems entirely washed out in the experiment. Would higher experimental energy resolution be able to see it or is it buried in the lifetime limited tail because of solid-state effects or similar?

Answer: We observed now for the first time the satellite peak on the bottom of the RIXS map for 180° scattering geometry. It is marked with an ellipse in the experimental and theoretical data in Figure 6. We assume that it is not observed at 90° likely since it has weak intensity.

Figure 6. **M₄ edge CC-RIXS maps at 180° π scattering geometry for U^{VI} samples with different covalency** a-d M₄ edge CC-RIXS intensity maps plotted on a logarithmic scale and with traced isointensity curves at fixed values. The ellipsoids evidence emerging satellites at lower emission energy -a-b Experimental spectra for

$[\text{U}^{\text{VI}}\text{O}_2]^{2+}\text{-Pb}$ (a) and $[\text{U}^{\text{VI}}\text{O}_2]^{2+}\text{-Cd}$ (b). c-d corresponding computed spectra on a LF-DFT level. e-f constant emission energy cross sections at the maximum intensity of the satellite and WL peaks. Lines with corresponding linestyle and color are traced in a-d to show the path along which the section are taken. Magenta and grey are chosen for the satellite section of $[\text{U}^{\text{VI}}\text{O}_2]^{2+}\text{-Pb}$ and $[\text{U}^{\text{VI}}\text{O}_2]^{2+}\text{-Cd}$, respectively, while cyan and black for the WL ones. Dashed lines are used for the experiment, continuous ones for the theory.

Comment (11): L267-268: Is the experimental broadening for PuO₂ much different than in the other cases? Why? I did not find information on the experimental energy resolutions for different samples.

Answer: We added now a table with calculated and for some cases measured experimental energy broadenings (experimental energy resolution), core-hole lifetime broadenings and total broadening. The data presented in the manuscript is measured over several years using different experimental conditions. All details are now summarized in table 1 and table S17.

Comment (12): The Figure 2a is slightly confusing. The scale of the ordinate axis is $1e1$, which I understand as meaning 1.0 on the y-scale equaling 10 eV. If the “energy difference between the spin singlet and triplet states is equal to 14 times $G(k=0)(4f5f)$ ” (line 108), why is it in Fig 2a finite at $G=0$ (about 10 eV) and not linearly increasing? (It would be visually clearer if the $1e1$ would not be the overall scale, why not label y-axis ticks as they are in eV)

Answer: Figure 2a shows the possible energies of a $4f^{13} 5f^1$ configuration. There are 14 possibilities for the 4f hole and 14 possibilities for the 5f electron. In total there are $14 \times 14 = 196$ microstate in the a $4f^{13} 5f^1$ configuration. At $G_{04f5f}=0$ one finds that the states are split by the spin-orbit coupling and multipolar coulomb interactions. Figure 2a shows these states labeled by $5f_{5/2}$ hole and $4f_{7/2}$ hole with an energy difference of about 10 eV on the left side where $G_0=0$. Moving along the x axis shows how the states evolve as a function of G_0 . One can see that from the multiplets with a $j=5/2$ hole the 1S_0 state splits and the energy is nearly linear. For small G_0 one should include also the spin-orbit interactions and multipolar Coulomb interactions to understand the full energies and the behavior slightly deviates from a linear curve.

Comment (13): What was the degree of polarization in the horizontal/ vertical polarization geometry and how the polarization was changed between sigma and pi polarization? There was no mention of a possible optical element or such that switches the polarization. Could a deviation from 100% pure polarization cause some effects on the data? Already a finite size of the analyzer crystal creates deviations from pure pi/sigma geometry, which could only be matched at one line on the analyzer surface.

Answer: We did not change the polarization of the incoming X-rays between pi and sigma. All data presented is for pi polarization. However, in the revised manuscript, we performed RIXS studies for 180 pi scattering geometry, which leads to the same results as 90° sigma geometry.

Comment (14): Lines 153-155: “Several levels of theory have been applied to investigate the origin of the intensity of the satellite peak and its relation to ground-

state material properties.” What does this mean in practice and do the authors refer to the SI or some other references

Answer: We would like to thank the referee for this comment: We have improved the wording in the paper and now write: “We applied atomic multiplet theory, crystal field theory, multiplet ligand field theory and DFT based ligand field theory (cf. Calculations in Methods) to investigate the origin of the intensity of the satellite peak and its relation to ground-state material properties. These levels of theory all include the full local coulomb interaction and add an increasing realistic description of the solid environment and chemical bonding.”

Comment (15): Small typographical issue: Lines 62,63,79, subindex 4 is missing from M4 edges.

Answer: Thanks for catching this, we improved it.

Comment (16): In order for the others to reproduce the results, Quantity input files and the processed 2D RIXS maps in numeric format, as electronic supplementary information, would be extremely valuable.

Answer: We will upload them to the quantity webpage as a tutorial before the paper is published. We have included a code availability statement: “The code Quantity used for this study is released under the CC By license and can be obtained from www.quantity.org. The input files are accessible as part of Quantity tutorials under (xxxx)”

Reviewer #3

Comment (1): The manuscript describes the assignment of 5f shell electron occupations via the satellite peak observed in M4-edge RIXS maps. Such an experimental tool is urgently needed, and therefore the manuscript is of potential interest to a broad readership. I am not sure whether the manuscript in its current form makes a convincing case, but it may be that I misunderstood some of the text. A revision is recommended.

Answer: We thank the review for recognizing the value of our work.

Comment (2): The manuscript asserts that ground state 5f occupations can be extracted from the RIXS satellite. A series of atomic calculations (labeled AT, CFT) were performed, and in one case a DFT-calibrated ligand-field theory (LFT) calculation was also carried out. The latter appears to be using DFT orbitals and therefore it is potentially impacted (potentially severely) by the Kohn-Sham delocalization error. It has been shown that this error leads to too much covalency in metal complexes. See, for example, DOI 10.1021/ar500171t for an overview. Related to this, in lines 252-278, it is my impression that the attempt here is to calibrate atomic calculations so they are able to reproduce the experimental spectra. One should then be careful not to over-interpret the adjustments in the atomic data as real molecular effects related to bonding. That would have to be based on a thorough comparison with molecular/ligand field calculations that are not impacted by the delocalization error.

Answer: It is indeed correct that DFT over estimates covalency. In order to circumvent this problem, we use multi reference (post Hartree-Fock or DFT) methods that can capture both the covalence and the local atomic multiplets of open shell materials. Multi reference methods generally build on a Hartree Fock or DFT calculation on top of which a multi-Slater determinant calculation is performed. The underlying idea is to use a large basis for a not so expensive (Hartree-Fock or DFT) level of theory and then a smaller basis for the more expensive multi Slater determinant method that includes the correlations and covalence better. If one expands the basis for the many body part one converges to exact result independent if one starts on top of the one particle basis obtained from Hartree-Fock, DFT or a set of Gaussian basis functions. For charged single atoms or small molecules convergence has been shows (by us in for example <https://doi.org/10.1038/s41567-024-02461-9>). In general, we have the working hypothesis that also starting from Hartree-Fock orbitals yields a lower total energy, starting from DFT orbitals converges excitation energies faster as one makes a similar error for the ground and excited states.

We furthermore would like to compare Multiplet ligand field theory to restrictive active space calculations. Both are post Hartree-Fock or post DFT methods that include many body effects. In RAS calculations the active space is formed by the orbitals around the chemical potential. In MLFT the local U 5f and core electrons are treated with all many body interactions but still interact with the ligands. The difference is that within RAS calculations the amount of covalent mixing between the U and Ligand orbitals is fixed on the DFT level. In MLFT calculations the amount of covalency can change due to the interactions and dynamically changes when calculating spectroscopy. This allows one to obtain charge transfer satellites on this level of theory, especially visible in core level photo-emission (see for example Multiplet ligand-field theory using Wannier orbitals, Physical Review B 85 (16), 165113 by us, or the textbook Core Level Spectroscopy of Solids by de Groot and Kotani.

As a last comment we would like to stress that on a MLFT level the charge fluctuations are greatly reduced with respect to DFT or Hartree-Fock. On a DFT or Hartree-Fock level the energy of different f^n configurations scales linearly with n. On a many body level it scales quadratic in n thereby reducing the charge fluctuations and thus the bonding covalence. Core level spectroscopy can generally also be used to measure the amount of charge fluctuations and thus give a measure for the correlations in the material, but that goes beyond the scope of the present paper.

**(3) Further comments:
(line numbers as in the PDF generated by the publisher)**

Line 44: "experimentalgeometry" is missing a blank space

Answer: Thanks, we improved it.

Lines 322-323: "For example, there is an intensive discussion of the formal oxidation state of low valent actinide compounds, i.e. An I, II or III.[29–33]" The citations should perhaps include also the following article: DOI 10.1021/jacs.1c07519

Answer: We added this reference.

Comment (4): When the authors write in lines 296-298 "Therefore, it could potentially be used as a tool to probe bond covalency, in particular for the case when the formal 5f occupation is already known." , do they consider some of the examples in the citations listed under the previous point as cases for which the formal occupation is not known? If not, please provide other examples.

Answer: For the revised paper we now investigated 6 U(VI)O₂⁺ (uranyl) compounds, where 5 of them are coordinated by 5 or 6 O atoms. One compound has one O exchanged by N. The strength of U-O_{equatorial} ligands interaction is modified by close or long/no interaction of Pb/Ag/Cd with the O_{axial}. This leads to small variations of the U(VI)-O_{ax} bond but stronger U(VI)-O_{eq} bond variations and allows to systematically study the satellite peak intensity at the 180° scattering geometry. Note that this geometry is equivalent to 90° sigma polarization scattering geometry, which was experimentally not possible. We found that the satellite peak is split forming two distinct peaks and the satellite on the bottom of the maps predicted by the theory was observed. Note that this peak is demonstrated here for the first time. We found a clear trend between integral intensities of the two top satellite peaks and variations of U-ligands bond covalency. In addition, we performed ligand field DFT calculations for two of the compounds and observe the same trends as in the experiments. The text discussing the satellite peak as a tool to probe covalency of bonds is on page 16 see below:

"In order to verify this deliberation, we recorded U M₄ edge CC-RIXS maps in 180° π scattering geometry for a series of U^{VI} compounds. In addition to the [U^{VI}O₂]²⁺-Pb and [U^{VI}O₂]²⁺-Cd compounds discussed above, U^{VI}-yl coordinated by 5 or 6 O atoms in the equatorial plane with or without direct interaction of a metal cation (M=Pb, Ag or Cd) with the O_{ax} atom were studied (Figure SI24); details on the coordination structures are given in Figure SI25.^{39,50,51} Furthermore, we performed ligand field density function theory (LFDFT) calculations of the U M₄ edge CC-RIXS maps (cf. SI section 19). This theoretical approach uses DFT and the average of configuration type calculations that enables the evaluation of ligand-field parameters without scaling factors or empirical corrections. The experimental and calculated U M₄ edge CC-RIXS maps and cross sections show clear variations for the [U^{VI}O₂]²⁺-Pb and [U^{VI}O₂]²⁺-Cd compounds displayed in Figure 6 (cf. Figure SI26, SI27). The RIXS maps and their cross sections at constant emission energies at the maxima of the lower satellite peak or the WL are similar for experiment and theory and reveal two satellites on the top and one satellite on the bottom of the CC-RIXS maps. To decipher the influence of the inter-electron repulsion on the satellite peaks, we performed two different types of calculations (1) omitting both 3d-5f and 4f-5f inter-electron interaction parameters (Figure SI28e and SI28f) or (2) implementing 3d-5f but still not considering 4f-5f interactions (Figure SI28g and h). The results depicted in Figure S28 illustrate that the two satellite peaks equally appear but only when the 4f-5f interaction is considered. For [U^{VI}O₂]²⁺-Cd this 4f-5f interaction is smaller leading to larger intensity of the satellite peaks. This result strongly suggests that the integral intensity of the satellite peaks measured in 180 π scattering geometry is sensitive to the size of the 4f-5f interaction. In turn, the size of the 4f-5f interaction depends on the 5f electron density distribution between U and the ligands. The larger is the U-ligands bond covalency, the smaller will be the inter-electron repulsion interaction as predicted by the Nephelauxetic effect,⁵² which is correlated here with the integral intensity of the satellite peaks (cf. Figure S29). It states that for a free ion, the inter-electron repulsion parameters are the biggest possible. When the metal

ion is coordinated with ligands, the value of inter-electron repulsion parameter (4f-5f in our case) ultimately becomes smaller than in the free ion case; the extent of this reduction is related to the electron density by the metal center and varied upon the covalency of the metal-ligand bond.

The integral intensity of the two satellite peaks is larger for $[\text{U}^{\text{VI}}\text{O}_2]^{2+}\text{-Cd}$ (Figure SI25a, f, h) than $[\text{U}^{\text{VI}}\text{O}_2]^{2+}\text{-Pb}$ (Figure SI25g, e, i) for both experiment and theory (cf. Figure 6) suggesting smaller U-ligands bond covalency due to less electron density on U in $[\text{U}^{\text{VI}}\text{O}_2]^{2+}\text{-Cd}$. As previously discussed, the U- O_{ax} bond covalency is slightly lower for the $[\text{U}^{\text{VI}}\text{O}_2]^{2+}\text{-Pb}$ due to the Pb- O_{ax} competitive interaction. However, Brager et al. showed that the equatorial interactions are stronger for close M- O_{ax} .

Considering that the integral intensity of the satellite peaks is smaller and distinctly different for the compounds with direct M- O_{ax} interactions $[\text{U}^{\text{VI}}\text{O}_2]^{2+}\text{-Pb}$ and $[\text{U}^{\text{VI}}\text{O}_2]^{2+}\text{-Ag}$ (structure Figure SI25 a and c) compared to compounds with no interaction ($[\text{U}^{\text{VI}}\text{O}_2]^{2+}\text{-Cd}$), as shown in Figure SI26, we can conclude that the satellite peak recorded in $180^\circ \pi$ scattering geometry appears sensitive to the total metal-ligand bond covalency. Note that the A-C shift in the HR-XANES spectra exhibits small variations suggesting only very small U- O_{ax} bond covalency differences for these compounds. The satellite peak has the largest intensity for the $[\text{U}^{\text{VI}}\text{O}_2]^{2+}$ organometallic compound indicating the smallest U-equatorial ligands bond covalency. The clear variations of the intensity of the satellite peak for the $180^\circ \pi$ scattering geometry compared to the lack of intensity for the $90^\circ \pi$ scattering geometry collaborates the theoretical prediction that the peak at $180^\circ \pi$ scattering geometry will be sensitive to An-ligands bond covalency.

“

Answers to Reviewers comments on manuscript

‘Counting the 5f electrons of the actinides’

by Bianca Schacherl, Michelangelo Tagliavini, Hanna Kaufmann-Heimeshoff, Jörg Göttlicher, Marinella Mazzanti, Karin Popa, Olaf Walter, Tim Pruessmann, Christian Vollmer, Aaron Beck, Ruwini S. K. Ekanayake, Jacob A. Branson, Thomas Neill, David Fellhauer, Cedric Reitz, Dieter Schild, Dominique Brager, Christopher Cahill, Cory Windorff, Thomas Sittel, Harry Harry Ramanantoanina, Maurits W. Haverkort, Tonya Vitova.

Reviewer #1:

The authors adequately addressed my main concerns. I have noticed only a couple of minor issues during the reading:

1. Line 77: “accessible at several synchrotrons all over the world.18“ Please check the reference. While the introduction of 18 also contains some information about other beamlines the reference to the review article 19 looks more appropriate.
2. References 18 and 36 are identical

Answer: We thank the reviewer for his/her time and the positive feedback. We have corrected the remaining issues about the references.

Reviewer #1 (Remarks on code availability):

Quanty is well known code in the field. Some of my colleagues use it.

Reviewer #2:

As the reviewer #3 comments, experimental tools to accurately and reliably determine the actinide 5f shell electron occupation, are highly desirable, and if robust methods can be found, it would have significant impact. The authors present a systematic study of a satellite line in a core-to-core M edge RIXS map that follows a systematic trend as a function of the 5f electron occupancy. My main concern in the first version of the manuscript was that the case was not very convincingly shown to be treated as valuable novel results. This was the impression of the reviewer #1 and #3 as well.

The authors have made significant improvements, and the second version of the manuscript shows clear advancements to the first version. The authors have collected a new and larger data set that confirms the trends shown in the first manuscript smaller dataset. The integral spectral weight of the satellite line has been measured in a variety of compounds with different 5f shell state. For the 5f electron counting purpose the figure 4b-c is now the important one (experimental and theoretical trend of 5f electron number and the integrated relative satellite spectral weight). The new data set shows variations and overlap of the integral values between neighboring formal valence state compounds. It seems that an uncertainty roughly equal to one 5f electron remains in the spread of the results. Especially 5f1 and 5f2 can be difficult to be discerned from each other, on the other hand 5f3 and 5f2 have similar weights, then 5f4, 5f5 and 5f6 form a group that puts them together in the same ballpark, especially f4 and f6 configurations probably will never be discerned from each other.

Answer: We agree that the relative intensity of the satellite peak does not seem to change very strongly for some of the configurations. The satellite-peak tool needs to be used for a set of materials with different 5f configurations in order for reliable conclusions to be drawn for unknown compound(s). We have now added the correlation of the satellite peak intensity to the relative An M4 edge HR-XANES shift, which is also a useful tool. This will aid the characterization of the 5f electron configuration since the absorption edge shift is sensitive to more delocalized 5f electron density (see the answers to questions below). For example, for similar satellite peak intensities, one could compare the energy shift of the absorption edge. One example is U(IV)Cp₃ (5f²) and U(II)Cp₃ (5f³). The intensities of the satellite peaks are very similar, but there is a strong shift of the U(II)Cp₃ (5f³) HR-XANES to lower energy. This implies that the additional electron density on U(II) is likely not strongly localized on the U atom but rather delocalized and participating in covalent bonds. The reduction of the satellite peak intensity is also likely partially influenced by the Nephelauxetic effect in U(II)Cp₃ (5f³). Note that the theory predicts that the Nephelauxetic effect has a much stronger influence on the satellite peak intensity at 180° σ scattering geometry.

Because of this overlap, I still cannot help wondering if this sets the accuracy of the 5f occupancy determination any higher than HR-XANES, see e.g. the HR-XANES spectra in the supplementary information. It is true that other effects also affect the XANES features and renders them indicative of the shell occupancy, but the overall results in Fig 4b indicate similar level of intrinsic uncertainty of the satellite intensity measurement results.

It is entirely true that the more data we have at hand, the better we can understand the system under study, and the fact that both localized and delocalized electrons can influence the HR-XANES, and that the possibility to measure also the satellite to obtain results for more localized electrons is useful. The entire case seems now to rest in this: whether the results in the improved manuscript and larger data set convincingly demonstrate a novel method that is interest to a broad audience. Unfortunately the current manuscript, even being in a much better state than the first version, still does not convince me very strongly. There is a trend that is a useful guide for the estimation of the orbital occupancy, but it is not highly accurate or monotonic, which would be needed to give a unique answer to the original research question. Seems that the original problem still remains, although the presented spectroscopic tool is certainly a useful one in addition to the old ones.

Answer: We are very grateful for the involvement of the reviewer and the very critical and useful comments. We felt extremely motivated to further develop and improve the manuscript. Figure 5 now shows the correlation between the relative satellite peak intensity and the relative absorption edge shift of the An M4 edge HR-XANES. The dashed linear function is a fit to the U values, excluding the values for U(II)Cp₃ (5f³) and U(IV)Cp₃ (5f²), which deviate. We observe values both below and above the line. For example, Pu has increased localization of the 5f electrons, leading to a large satellite peak intensity but a smaller relative absorption edge shift. In contrast, U(II)Cp₃ (5f³) and U(IV)Cp₃ (5f²) are below the linear function and strongly shifted to lower energies compared to UO₂ (5f²). This suggests that the 5f electron density on U is largely delocalized in both compounds.

This correlation graph can be extremely useful in drawing final conclusions about electron configurations and the covalency of actinide bonds, leading to more accurate analyses. We believe that the manuscript is significantly improved by including this Figure 5.

Concerning the discussion of the covalency: Maybe I still misunderstand something, but the results concerning this seem still confusing (most of it is in the SI probably because of length

limits, but I found it cumbersome to repeatedly change between reading the SI and the main text). Since there are various factors at play that can alter the interpretation (probably including the 4f-5f exchange interaction parameter), the argument ends up being rather vague: there is likely to be some variation related to the covalency in the satellite intensity at 180 degree scattering angle, but I miss the quantitative result here. I didn't see a point plotting the satellite intensity in the SI as a function of 5f occupancy in case of all data points having this value equal to zero. Some approach to plot it as a function of degree of covalency would make more sense.

Answer: We added Figure 8 (cf. discussion in page 18f) showing the relation between the satellite peak intensity in the backscattering geometry and the Quantum Theory of Atoms in Molecules (QTAIM) metrics for the U^{VI} system. These include the total electron density (ρ), and delocalization index (δ) for the U- O_{ax} and U-equatorial ligands bonding interactions at their respective bond critical points. The graph suggests that for larger U-ligands bond covalency, the satellite peak intensity decreases. Since the differences in bond covalency are small, this is mainly clear for the $[U^{VI}O_2]^{2+}$ -Pb (close Pb- O_{ax}) and $[U^{VI}O_2]^{2+}$ -Cd (no Cd- O_{ax} contact) compounds which present the most similar coordination environments but differences in bond covalency. The 4f-5f electron interactions decrease due to the larger 5f electron density on U for $[U^{VI}O_2]^{2+}$ -Cd (no Cd- O_{ax} contact), as a result of the Nephelauxetic effect, and the satellite peak intensity decreases.

“To decipher the influence of the inter-electron repulsion on the satellite peaks, we performed two different types of calculations (1) omitting both 3d-5f and 4f-5f inter-electron interaction parameters (Figure SI29e and SI29f) or (2) implementing 3d-5f but still not considering 4f-5f interactions (Figure SI29g and h). The results depicted in Figure S30 illustrate that the two satellite peaks equally appear but only when the 4f-5f interaction is considered. This result strongly suggests that the integral intensity of the satellite peaks measured in 172° σ scattering geometry is sensitive to the size of the 4f-5f inter-electron repulsion interaction. In turn, the size of the 4f-5f interaction depends on the 5f electron density distribution between U and the ligands. The larger the U-ligand bond covalency, the smaller the 4f-5f repulsion interaction as predicted by the Nephelauxetic effect,⁵³ which is correlated here with smaller integral intensity of the satellite peaks (cf. Figure S30). It states that for a free ion, the inter-electron repulsion parameters are the largest possible. When the metal ion is coordinated by ligands, the value of inter-electron repulsion parameter (4f-5f in our case) ultimately becomes smaller than in the free ion case since the 4f-5f repulsion decreases; the extent of this reduction is related to the electron density of the metal center and varies depending on the covalency of the metal-ligand bond(s).

The integral intensity of the two satellite peaks is smaller for $[U^{VI}O_2]^{2+}$ -Cd (no Cd- O_{ax} contact) (Figure SI26a, f, h) than $[U^{VI}O_2]^{2+}$ -Pb (close Pb- O_{ax} contact) (Figure SI26g, e, i) for both experiment and theory (cf. Figure 7), which suggests larger U-ligand bond covalency due to more electron density on U in $[U^{VI}O_2]^{2+}$ -Cd. This result is confirmed by the Quantum Theory of Atoms in Molecules (QTAIM)⁵⁴ metrics displayed in Figure 8. Figure 8 shows correlation graphs between the relative intensity of the satellite peaks for the 172° σ scattering geometry and reported QTAIM metric values for the $[U^{VI}O_2]^{2+}$ systems (see Table SI8). The electron density (ρ) and delocalization index (δ) metrics were chosen for the analysis. They were determined for the U- O_{ax} and U-equatorial ligands bonding interactions at their respective bond critical points. The bonding interaction is essentially dominated by the U- O_{ax} interaction (larger ρ and δ values for the five $[U^{VI}O_2]^{2+}$ systems). But the contribution of the U-equatorial ligand interaction is also not negligible and it presents the biggest disparity between the different systems. Compounds with five-coordinated equatorial ligands ($[U^{VI}O_2]^{2+}$ -Cd (no Cd- O_{ax} contact) and $[U^{VI}O_2]^{2+}$ -Pb close Pb- O_{ax}) have strong U-equatorial ligand bonds and at the same time slightly weaker U-O axial bonds. In turn, these compounds are mainly associated with smaller satellite peak intensity. On the other hand, compounds with six-coordinated equatorial ligands ($[U^{VI}O_2]^{2+}$ -Ag close Ag- O_{ax} , $[U^{VI}O_2]^{2+}$ -Ag long Ag- O_{ax} and $[U^{VI}O_2]^{2+}$ -Pb no Pb- O_{ax}) have weak U-equatorial ligand bonds and slightly stronger U-O axial bonds, which are mainly associated with larger satellite peak intensity. Note that $[U^{VI}O_2]^{2+}$ -Ag close Ag- O_{ax} slightly deviates from this trend, which probably arises from the effect of the Ag^+ ion on the overall electronic structure. The clear variations of the intensity of the satellite peak for

the 172° σ scattering geometry compared to the lack of intensity for the 90° π scattering geometry corroborates the theoretical prediction that the peak at backscattering geometry will be sensitive to An-ligand bond covalency”

Smaller technical remark: I didn't find an explanation to the change in the manuscript's approach to use 90 degree sigma and pi polarization, switching now to 180 degree pi polarization. Looks like there must have been an error in the original manuscript, with the theory in Fig 2d was earlier labeled to come from 90 degree sigma polarization. The old 3-crystal description is still there in the SI, but the 180 degree geometry seems new. I am not sure how the authors could vary the polarization, but now this issue seems resolved with the 90 degree sigma polarization removed.

Answer: In the first version of the manuscript we had only calculations for the 90 degrees sigma polarization. We wanted to demonstrate that the polarization play a role. We also revised the polarization for the near backscattering geometry (172° scattering angle). The polarization is defined with respect to the scattering plane, which is defined by the incident beam, the sample and the analyzer crystal. Since the polarization of the beam in linear in the plane of the synchrotron, it is linear σ with respect to the scattering plane.

I am also not sure how it is possible to use 180 degree scattering angle, there must be an error; the scattering angle was probably lower but the authors approximate it to 180 degree in the text. Is there a difference between 180 degree sigma and pi polarization case? I would think they must probably become equivalent at that point.

Answer: • We improved the definition of 180° scattering geometry by giving the precise scattering angle of 172° in Figure 2b. The polarisation for this scattering geometry was also clarified.

Reviewer #2 (Remarks on code availability):

Quanty is a state-the-art code, used by many in the field, and I have no doubts about its ability to perform the calculations; I haven't tried to reproduce the results but there I have absolute confidence in the code and the authors behind the computations.

Answer: We are very thankful and pleased to read this positive comment of the reviewer!

Reviewer #4:

This work reports novel spectroscopic techniques that can count the localized 5f electrons and derive information on the degree of 5f electron sharing in actinide complexes. Essentially, these techniques explore changes in a satellite feature, which appears some 6-8 eV apart from the WL in the core-to-core CC-RIXS map, as a function of the localized 5f electron count on the actinide ion (i.e., as a function of the actinide formal oxidation state). The appearance of the satellite feature is based on the strength of the 4f-5f exchange interaction causing a net separation between high- and low-spin states resulting from the spin-exchange coupling of the f electrons. The present manuscript provides a novel, alternative approach to rationalize the oxidation state and chemical bonding in actinide complexes, it is definitely urgent and of tremendous interest for the community of Nature Communications. I recommend publishing it.

To be honest, I find the manuscript truly impressive, reporting in detail data from experiments that are far from trivial to handle. Given the complexity of the electronic structure, the

calculations reported here, including LFDFT and atomic multiplet approaches, are at the forefront of what can be achieved. This is because fully ab initio methods are inapplicable (due to complexity) to all the electronic structure problems addressed in this work. As an observation, in response to comment R3, comment 2, I would like to point out that RAS is a post-Hartree-Fock method, not a post-DFT method. One can indeed use DFT orbitals in RAS calculations, but this is not a standard recommendation, nor has the theory been formulated within the realm of Kohn-Sham DFT. Similar to CASSCF, the RASSCF method is variational, addressing both the multideterminantal description of the wavefunctions and the ligand field. That is, the ligand orbitals are not "fixed at the DFT level," nor does the whole approach have anything to do with DFT. RASSCF calculations are the desired approach here to calculate the satellite intensities; however, these cannot be used due to the complexity of the electronic structure.

Answer: We are very thankful for the positive remarks and the appreciation of Dr. Sergentu for our work!

A comment that I have relates to the emergence of intensity under the satellite peak. If I get it correctly, it becomes apparent from the text and Figure 2a that intensity under the WL is due to the pairing of the 5f and 4f electrons in high- and low-spin states. The intensity under the WL comes from the high-spin states (or states of same spin-multiplicity with the ground state?), while the intensity under the satellite comes from the low-spin states. If this is the case, then spin-orbit coupling should also be an important factor (?) in the generation of intensity under the satellite, because it promotes intensity by mixing the optically spin-allowed states with the spin-forbidden states. Thus, the more localized the electrons are in the 5f shell, the more angular momentum they have, and the more they generate states that are involved in better spin-orbit coupling. Charge-transfer 5f electrons describing covalent bonding have diluted angular momentum and do not lead to configurations that involve significant spin-orbit coupling, hence no improvement in the satellite intensity. Now it becomes clear that the higher the oxidation state of the actinide, the higher the intensity of the satellite peak will be. This aligns well with the experimental observations. It remains unclear why the intensity trend breaks for 5f6 but I guess this should have something to do with the fact that upon promotion of the 3d electron in the 5f6 shell, the 5f shell as a whole becomes half-filled, 8S term, that is hardly involved in spin-orbit coupling. I am not sure how the CF splitting of the 5f orbitals leads to changes in the position of the satellite wrt to the WL. My feeling is that resting the whole electronic structure problem on the monopole part of the exchange interaction isn't too simple. Whatever the answer is here, I do believe that the calculations presented here bring excellent support for the experimental observations and leaves room for future work.

Answer: The underlying mechanism sketched by the referee is indeed correct. We would like to add that the 4f states are core states that are not directly influenced by covalence or crystal fields. The 4f state on its own (single hole) is thus well described on a jj coupling basis. The interaction of the 4f core states with electrons in the 5f shell is mediated by Coulomb interaction. This interaction favors an LS coupling. The competition between jj or LS coupled states is purely given by the exchange part of the Coulomb interaction and leads to the non-statistical branching ratios in XAS as for example discussed by Thole and Van der Laan (PRB 38, 3158 1988). The competition between core-valence exchange interaction and core spin-orbit coupling also leads to additional multiplet peaks. These multiplets broaden the XAS spectra while in RIXS these multiplets show as clear satellites. The intensity of the satellite peak is given by Clebsch Gordon coefficients related to the angular momentum coupling of the valence and core states. The values of $\langle L^2 \rangle$ and $\langle S^2 \rangle$ for the core states are "fixed" as it is a single hole in the 4f shell. As a result, the intensity depends on the value of $\langle S^2 \rangle$ and $\langle L^2 \rangle$ in the 5f shell, which scales with the number of electrons. At half filling the expectation value of $\langle L^2 \rangle$ becomes zero and

the angle between L and S changes this leads to a change in the trend that one start to see for $5f^6$ compounds.

Naturally, albeit in higher order, the satellite intensity also depend on crystal field sizes. (To first order crystal fields do not act on the 4f core states) Actually, the crystal field splitting is often induced due to orbital dependent covalency that lead to different energies for the anti-bonding orbitals. RIXS spectra are sensitive to orbital dependent covalence and bonding, but that would require a more detailed polarization dependent analysis of the spectra on single crystals. Something that is interesting for future research.

Minor comments:

1. In the abstract, remove “and electron occupation.”, I find it redundant with the “spin configuration” which is itself generated by the electron occupation.

Answer: Thanks for your comment we removed it.

2. Line 47: “large number of 5f valence electrons” -> “large number of electrons”. E.g., a U(VI) complex formally has 5f0 valence electrons but still shows a tremendous amount of correlation.

Answer: Yes we agree, we removed the 5f valence electrons from the sentence.

3. Line 324: Reference 47 does not seem to report any theoretical data.

Answer: We added citation to a publication with the theoretical data and also changed the sentence to:

“Here, we measured the satellite intensity of $[\text{Cp}''_3\text{U}^{\text{II}}]^-$, which has been assigned a $5f^36d^1$ configuration based on the UV-visible spectroscopy, magnetic data, single crystal XRD and their analogy to the lanthanide compounds.⁵⁰ DFT calculations confirmed this assignment.⁵¹”

4. Line 371: Remove the first sentence. The need for multireference methods to address the electronic structure is well established. In fact, the 4f135fn configurations explored here cannot be correctly calculated without such methods.

Answer: We thank the referee for this comment, which is naturally correct. The interesting question is the competition between local moment formation and covalence or band formation. We removed the first sentence of this paragraph.

Sergentu Dumitru-Claudiu.

Reviewer #4 (Remarks on code availability):

Quanty has become a very popular code, especially for modeling the electronic multiplets involved in X-ray absorption spectroscopy for strongly correlated systems. The software is straightforward to install and run, allowing users to script and calculate matrix elements easily

among angular momentum states and an operator of choice. The authors provide Quantity input samples, making it easy to reproduce the data in the manuscript.